# HyU: Hybrid Unmixing for longitudinal in vivo imaging of low signal-to-noise fluorescence

Hsiao Ju Chiang[1,2,5], Daniel E. S. Koo[1,2,5], Masahiro Kitano[1,3], Sean Burkitt [2], Jay R. Unruh[4], Cristina Zavaleta[2], Le A. Trinh[1,3], Scott E. Fraser[1,2,3] & Francesco Cutrale [1,2] ✉

The expansion of fluorescence bioimaging toward more complex systems and geometries requires analytical tools capable of spanning widely varying timescales and length scales, cleanly separating multiple fluorescent labels and distinguishing these labels from background autofluorescence. Here we meet these challenging objectives for multispectral fluorescence microscopy, combining hyperspectral phasors and linear unmixing to create Hybrid Unmixing (HyU). HyU is efficient and robust, capable of quantitative signal separation even at low illumination levels. In dynamic imaging of developing zebrafish embryos and in mouse tissue, HyU was able to cleanly and efficiently unmix multiple fluorescent labels, even in demanding volumetric timelapse imaging settings. HyU permits high dynamic range imaging, allowing simultaneous imaging of bright exogenous labels and dim endogenous labels. This enables coincident studies of tagged components, cellular behaviors and cellular metabolism within the same specimen, providing more accurate insights into the orchestrated complexity of biological systems.

In recent years, high-content imaging approaches have been refined for decoding the complex and dynamic orchestration of biological processes[1–3]. Fluorescence, with its high contrast, high specificity and multiple parameters, has become the reference technique for imaging[4,5]. Continuous improvements in fluorescence microscopes[6–9] and the ever-expanding palette of genetically encoded and synthesized fluorophores have enabled the labeling and observation of a large number of molecular species[10,11]. This offers the potential of using multiplexed imaging to follow multiple labels simultaneously in the same specimen, but the technologies for this have fallen short of their fully imagined capabilities. Standard fluorescence microscopes collect multiple images sequentially, employing different excitation and detection bandpass filters for each label. Recently developed techniques allow for massive multiplexing by utilizing sequential labeling

of fixed samples but are not suitable for in vivo imaging[12,13]. Unfortunately, these approaches are ill-suited to separating overlapping fluorescence emission signals and the narrow bandpass optical filters used to increase selectivity decrease the photon efficiency of the imaging (Supplementary Figs. 1 and 2) These limitations have restricted the number of imaged fluorophores per sample (usually 3–4) and risk exposing the specimen to damaging levels of exciting light. This has been a substantial obstacle for dynamic imaging, preventing in vivo and intravital imaging from reaching its full potential with a broader impact on research, from developmental biology[14], cancer research[15] and immunology[2] to neuroimaging[16].

Hyperspectral fluorescent imaging (HFI) potentially overcomes the limitations of overlapping emissions by expanding signal detection into the spectral domain[17]. HFI captures a spectral profile from each

[1]Translational Imaging Center, University of Southern California, Los Angeles, CA, USA. [2]Department of Biomedical Engineering, University of Southern California, Los Angeles, CA, USA. [3]Molecular and Computational Biology, University of Southern California, Los Angeles, CA, USA. [4]Stowers Institute for Medical Research, Kansas City, MO, USA. [5]These authors contributed equally: Hsiao Ju Chiang, Daniel E.S. Koo. ✉e-mail: cutrale@usc.edu

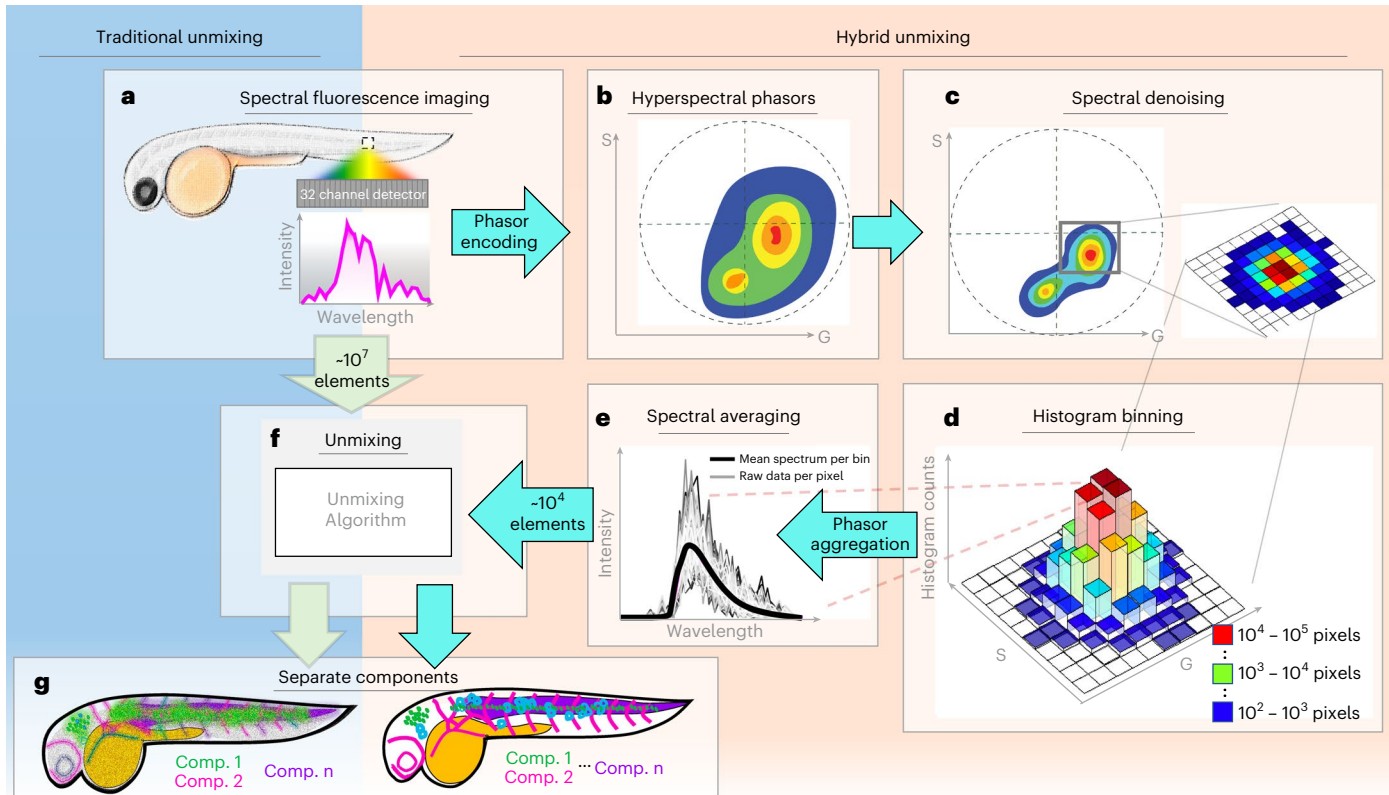

**Fig. 1 | Schematic illustrating how HyU enhances analysis of multiplexed hyperspectral fluorescent signals in vivo. a**, Multicolor fluorescent biological sample (here a zebrafish embryo) is imaged in hyperspectral mode, collecting the fluorescence spectrum of each voxel in the specimen. **b**, HyU represents spectral data as a phasor plot, a 2D histogram of the real and imaginary Fourier components (at a single harmonic). **c**, Spectral denoising filters reduce the Poisson and instrumental noise on the phasor histogram, providing the first signal improvement. **d–f**, The phasor acts as an encoder, where each histogram bin corresponds to a number *n* of pixels, each with a relatively similar spectrum (**e**). Summing these spectra effectively averages the spectra for that phasor position. This denoising results in cleaner average spectrum for this set of pixels, which are ideally suited for analytical decomposition through unmixing algorithms (**f**). **g**, Unmixing results in images that separated into spectral components. Here, LU is used for unmixing, but HyU is compatible with any unmixing algorithm.

pixel, resulting in a hyperspectral cube (*x,y* and wavelength) of data, which can be processed to deduce the labels present in that pixel. Linear unmixing (LU) has been widely utilized to analyze HFI data and has performed well with bright samples emitting strong signals from fully characterized, extrinsic fluorophores, such as fluorescent proteins and dyes[18–20]. However, in vivo fluorescence microscopy is almost always limited in the number of photons collected per pixel (owing to expression levels, biophysical fluorescent properties and sensitivity of the detection system), which reduces the quality of the spectra acquired. While solutions beyond the standard LU have been proposed[21], the challenge of analyzing low-intensity spectral signals remains.

A further challenge that affects the quality of spectra is the presence of multiple forms of noise in the imaging of the sample. Two examples of instrumental noise are photon noise and read noise[22–25]. Photon noise, also known as Poisson noise, is an inherent property related to the statistical variation of photons emission from a source and of detection. Poisson noise is inevitable when imaging fluorescent dyes and is more pronounced in the low-photon regime. It poses challenges, especially in live and timelapse imaging, where the power of the exciting laser is reduced to avoid photo-damage to the sample, decreasing the amount of fluorescent signal. Read noise arises from voltage fluctuations in microscopes operating in analog mode during the conversion from photon to digital levels intensity and commonly affects fluorescence imaging acquisition. Most biological samples used for in vivo microscopy are labeled using extrinsic signals from fluorescent proteins or probes but often include intrinsic signals (autofluorescence). Autofluorescence contributes to photons that are undesired,

difficult to identify and to account for in LU. The cumulative presence of noise inevitably leads to a degradation of acquired spectra during imaging. As a result, the spectral separation by LU is often compromised and the signal-to-noise ratio (SNR) of the final unmixing is often reduced by the weakest of the signals detected[19]. Increasing the amount of laser excitation can partially overcome these challenges, but the higher energy deposition in the sample causes photo-bleaching and photo-damage, affecting both the integrity of the live sample and the duration of the observation. Traditional unmixing strategies such as LU are computationally demanding, requiring long analyses and often slowing the experiment. Combined, these compromises have reduced both the overall multiplexing capability and the adoption of HFI multiplexing technologies.

We have developed HyU as an answer to the challenges that have limited the wider acceptance of HFI for in vivo imaging. HyU combines our previous phasor hyperspectral approach[26] merged with traditional unmixing algorithms to untangle the fluorescent signals more rapidly and more accurately from multiple exogenous and endogenous labels. The phasor approach[26] is a popular dimensionality reduction approach for the analysis of both fluorescence lifetime and spectral image analysis[27–29]. In HyU, the phasor approach provides spectral compression, denoising and computational reduction that simplifies both pre-processing[30] and unmixing[31–33] of HFI datasets. Standard phasor analysis[26,27,34–41] is fully supervised and requires a manual selection of regions or points on the phasor plot, a graphical representation of the transformed spectra. In contrast, HyU utilizes phasor processing as an encoder to aggregate similar spectra onto the phasor plot, reducing

even the largest volumetric datasets so that unmixing algorithms, such as LU, can be applied on a far smaller number of elements (the number of pixels on the phasor plot). Furthermore, HyU provides unsupervised analysis of the HFI data, simplifying data processing and removing user subjectivity. Our results show that HyU offers three key advantages: (1) improved unmixing over conventional LU, especially for low-intensity images, down to five photons per spectra; (2) simplified identification of independent spectral components; and (3) dramatically faster processing of large datasets, overcoming the typical unmixing bottleneck for in vivo fluorescence microscopy.

## Results

### Method overview

HyU combines the best features of hyperspectral phasor analysis and LU, resulting in faster computation speeds and more reliable results, especially at low light levels. Phasor approaches reduce the computational load because they are compressive, reducing the 32 channels of an HFI spectral plot into a position on a two-dimensional (2D) histogram, representing the real and imaginary Fourier components of the spectrum (Fig. 1a,b). Different 32-channel spectra are represented as different positions on the 2D phasor plot and mixtures of the two spectra will be rendered at a position along a line connecting the pure spectra. Because the spectral content of an entire 2D or three-dimensional (3D) image set is rendered on a single phasor plot, there is a dramatic data compression, from a spectrum for each voxel in an image set (up to or even beyond gigavoxels) to a histogram value on the phasor plot (megapixels). In addition, because each 'bin' on the phasor plot histogram corresponds to multiple voxels with highly similar spectral profiles, the binning itself represents spectral averaging, which reduces the Poisson and instrumental noise (Fig. 1c–e). Poisson noise in the collected light is unavoidable in HFI unless the excitation is turned so high that the statistics of collected fluorescence creates hundreds or thousands of photons per spectral bin. The clear separation of the spectral phasor plot and its referenced imaging data, permits denoising algorithms to be applied to phasor plots with minimal degradation of the image resolution. LU or other unmixing approaches can be applied to the spectra on the phasor plot, offering a dramatic reduction in computational burden for large image datasets (Fig. 1d). To understand this saving, consider the conventional approach of LU applied to image data at the voxel level (Fig. 1a,f). A timelapse volumetric dataset of 512 × 768 × 17 (*x*, *y* and *z*) pixels, over six time points, (Supplementary Table 1), would require 40 million operations. HyU requires only ~18,000 operations to unmix each bin on the phasor plot, representing more than a thousand-fold saving (Fig. 1f,g).

### Performance evaluation

To quantitatively assess the relative performance of LU and HyU, we analyzed them on synthetic hyperspectral fluorescent datasets, created by computationally modeling the biophysics of fluorescence spectral emission and microscope performance (Fig. 2a,b and Supplementary Figs. 3–5). We used this synthetic dataset to evaluate LU

and HyU algorithm performance quantitatively by using metrics such as mean squared error (MSE) and unmixing residual (Supplementary Fig. 6 and Methods; for both metrics, a lower value indicates better performance). In addition to the computational efficiency mentioned above, HyU analysis shows better ability to capture spatial features over a wide dynamic range of intensities, when compared to standard LU, in large part due to the denoising created by processing in phasor space (Fig. 2a,b). The improved accuracy is demonstrated by a lower MSE, in comparing the results of LU and HyU to the image ground truth. The absolute MSE for HyU is ~2× lower than that of LU, especially at low and ultralow fluorescence levels (five denoising filters and 16 photons per spectrum) (Fig. 2c). MSE can be further decreased using denoising filters on the phasor plot, resulting in superiority of HyU relative to LU for HFI at low (5–20 photons per spectrum) and ultralow (2–5 photons per spectrum) levels (Fig. 2d). To better characterize the performance in the experimental data without ground truth, we also define the unmixing residual as the difference between the original multichannel hyperspectral images and their unmixed results. Residuals provide a measure of how closely the unmixed results reconstruct the original signal (Supplementary Fig. 3 and Methods). Unmixing residuals are inversely proportional to the performance of the algorithm, with low residuals indicating high similarity between the unmixed and the original signals. Analysis of unmixing residuals in the synthetic data highlights an improved interpretation of the spectral information in HyU with an average unmixing residual reduction of 21% compared to the standard (Supplementary Fig. 5c). The reduction in both MSE and average unmixing residual for synthetic data demonstrates the superior performance of HyU and provides a baseline comparison when demonstrating performance improvements for experimental data.

We support the enhanced performance of HyU with analysis of experimental data, which reveals comparatively lower unmixing residuals and a higher dynamic range as compared to LU. Data were acquired from a quadra-transgenic zebrafish embryo *Tg(ubiq:Lifeact-mRuby); Gt(cltca-citrine);Tg(ubiq:lyn-tdTomato);Tg(fli1:mKO2)*, labeling actin, clathrin, plasma membrane and pan-endothelial cells, respectively (Figs. 2e–l and 3, Supplementary Figs. 7–9 and Supplementary Video 1). HyU unmixing of the data shows minimal signal cross-talk between channels, whereas LU presents noticeable bleed-through (Fig. 2m–p). Consistent with synthetic data, we utilize the unmixing residual as the main indicator for quality of the analysis in experimental data, owing to the absence of a ground truth. The residual images (Fig. 2f,g) depict a striking difference in performance between HyU and LU. The average relative residual of HyU denotes a sevenfold improvement compared to LU (Fig. 2h) in disentangling the fluorescent spectra. We visualize the unmixed channels independently (Fig. 2i–l), zooming in on details (Fig. 2i–p) to highlight areas affected by bleed-through and which are difficult to unmix. HyU, with contrast twofold higher than standard LU, reduces bleed-through effects and produces images with sharper spatial features, leading to better interpretation of the experimental data (Fig. 2k,l, Supplementary Fig. 7 and Methods).

---

**Fig. 2 | HyU outperforms standard LU in both synthetic and live spectral fluorescence imaging. a,b**, HyU (**a**) and LU (**b**) tested using a hyperspectral fluorescence simulation that was generated from four fluorescent signatures (emission spectra; Supplementary Fig. 5e). **c**, Absolute MSE shows that HyU offers a consistent reduction in error across a broad range of photons per spectra (no. photons per independent spectral components, here resulting from four reference spectra combined). Shaded regions for line plots denote the 95% confidence interval around the mean. **d**, The performance differences in the MSE of HyU relative to LU persists when applying multiple phasor denoising filters (0–5 median filters). Shaded regions for line plots denote the 95% confidence interval around the mean. **e**, Unmixing of experimental data from a four-color zebrafish shows increased contrast for HyU (left) compared to LU (right). Scale bar, 50 μm. **f,g**, The increased accuracy is revealed by residual images of HyU

and LU, showing the spatial distribution of unassigned signals after the analysis of data in **e**. **h**, Box plots of the residuals (shown in **f,g**) presents values of 11% for HyU compared to 77% for LU with *($P < 10^{-10}$) with an independent two-sample *t*-test and $n = 1.05 \times 10^6$ pixels. Center shows median; box represents first and third quartiles; whiskers represent 1.5× first and third quartiles; min/max are not shown. **i–p**, Enlarged rendering of HyU results (**e**, white box) clearly shows low levels of bleed-through between labels (**m–p**). Similar enlargement of LU results show noticeably worse performance. Note that regions with bright signals (membrane **j,n**, white arrow) bleed through other channels (**m,o**). Scale bar, 20 μm. Tetra-labeled specimen used here was *Gt(cltca-citrine);Tg(ubiq:lyn-tdTomato; ubiq:Lifeact-mRuby;fli1:mKO2)*. The sample depicted in **e** is representative of 28 experimental sessions each with three to five biological replicates, yielding similar results.

## Decreased signal cross-talk

Applying HyU to another HFI dataset further highlights HyU's improvements in noise reduction (less bleed-through) and reconstitution of spatial features (increased spatial resolution) for low-photon unmixing (Fig. 3 and Supplementary Fig. 8). In the zoomed-in image of a single slice of the embryo skin surface, acquired in the trunk region, the HyU image correctly does not display pan-endothelial (magenta) signal in the periderm, an area which should be devoid of endothelial cells and mKO2 signal (Fig. 3c). In contrast, the result from LU shows a visually

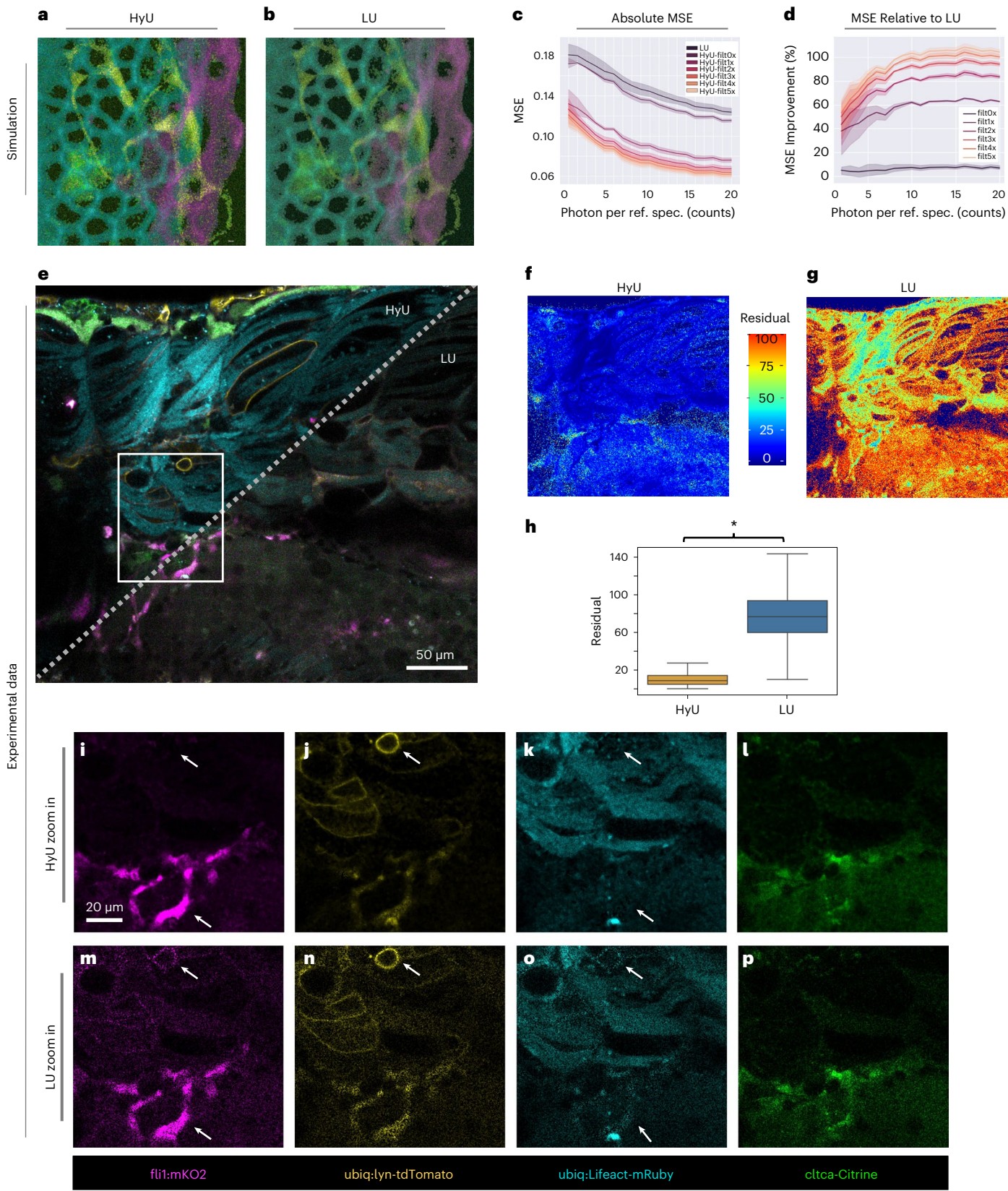

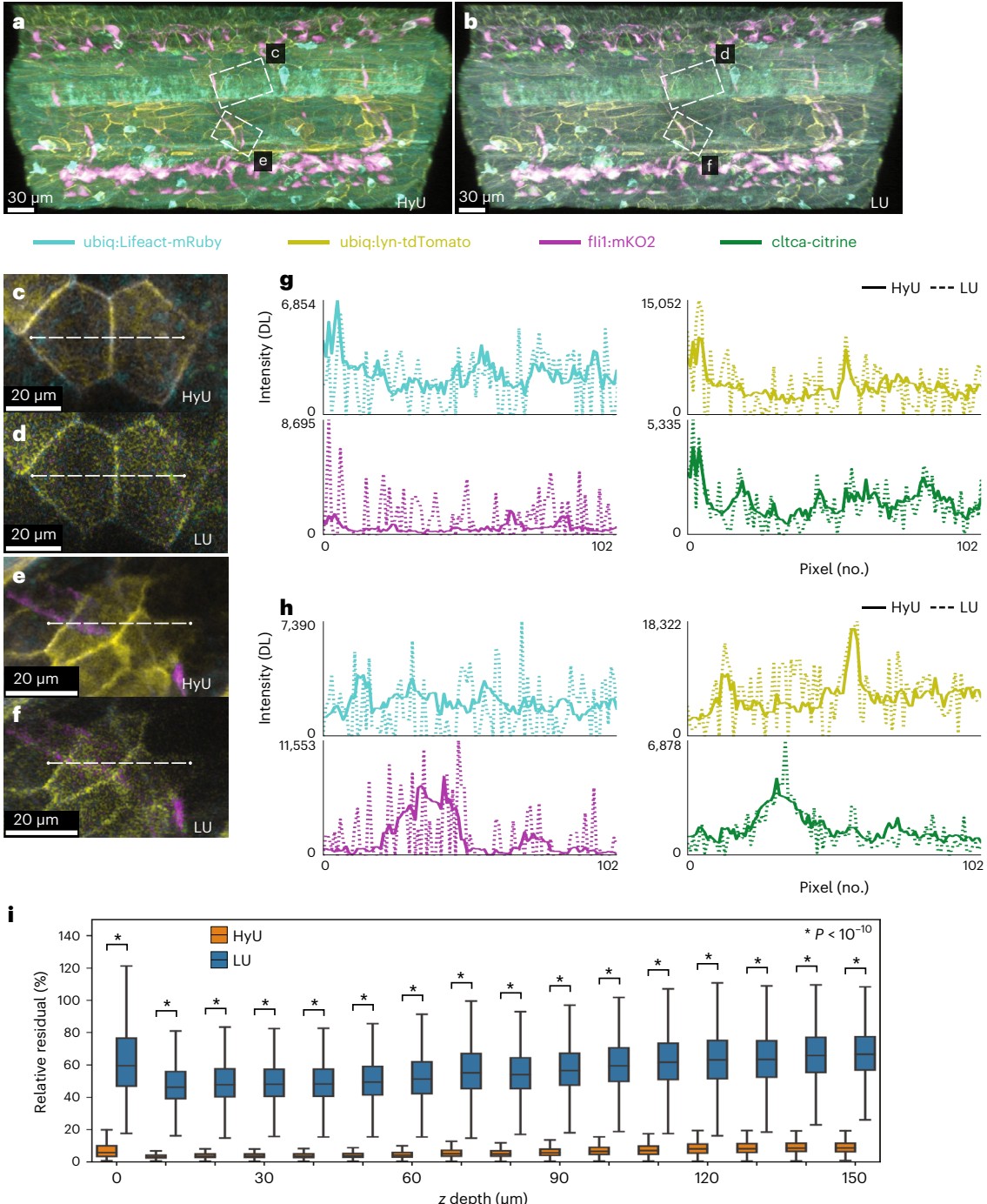

**Fig. 3 | HyU enhances unmixing for low-signal in vivo multiplexing and achieves deeper volumetric imaging. a,b,** HyU volumetric renderings (**a**) compared to those of LU (**b**) for the trunk portion in a four-color zebrafish. The four labels in the fish are *Gt(cltca-citrine);Tg(ubiq:lyn-tdTomato;ubiq:Lifeact-mRuby;fli1:mKO2)*, respectively labeling clathrin-coated pits (green), membrane (yellow), actin (cyan) and endothelial (magenta). **c–f,** Zoom-in views of insets from **a** and **b**. Scale bar, 20 µm. **g,h,** Intensity line plots of each of the four results signals for HyU (solid) and LU (dashed) demonstrate the improved profiles with greatly reduced noise peaks in HyU as compared to LU. Intensities are scaled by the maximum of each unmixed channel. DL, digital level. **i,** Box plots

of the relative residual values as a function of *z* depth for HyU and LU highlight the improvements in the unmixing results. HyU has an unmixing residual of 6.6 ± 5.3% compared to that of LU at 58 ± 17%. The average amount of residual is ninefold lower in HyU with narrower variance of residual. An independent two-sample *t*-test was used with $n = 5.2 \times 10^5$ pixels for each *z* slice. Center shows median; box shows first and third quartile; whiskers show 1.5× first and third quartiles; min/max are not shown. The sample depicted is representative of 28 experimental sessions each with three to five biological replicates, yielding similar results.

distinctive pan-endothelial signal throughout the tissue plane (Fig. 3d). This incorrect estimation of the relative contribution of mKO2 fluorescence for LU is possibly due to the presence of noise, corrupting the spectral profiles. This is further delineated in the intensity

profiles of the mKO2 signal between HyU and LU with much higher individual peaks from noise demonstrated for LU (Fig. 3g, bottom left). Intensity profiles for both magnified cross-sections of the volume (Fig. 3c–f) provide visualization of the improvements of HyU.

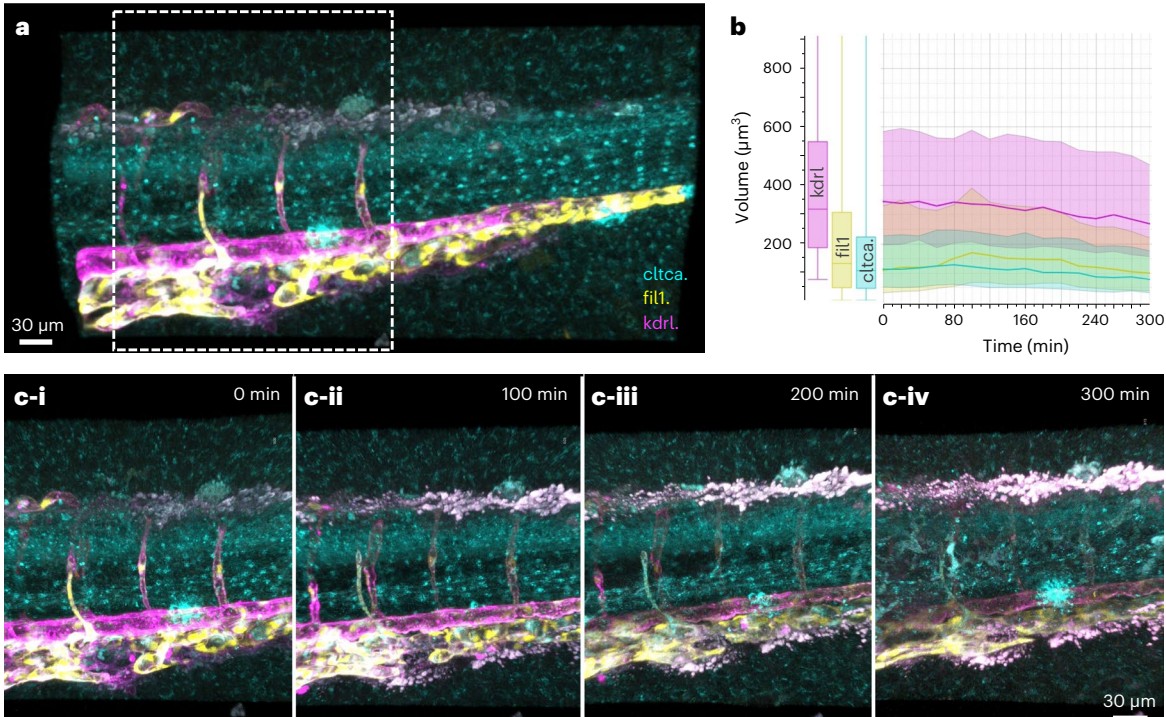

**Fig. 4 | HyU reveals the dynamics of developing vasculature by enabling multiplexed volumetric timelapse. a**, HyU rendering for the trunk portion of a three-color zebrafish *Gt(cltca-citrine);Tg(kdrl:mCherry;fli1:mKO2)* at time point 0. **b**, HyU unmixed results allow for quantitative analysis and segmentation, here an example representing the time evolution of the segmented volumes of mCherry (vasculature, magenta), mKO2 (endothelial lymphatics, yellow) and citrine (clathrin-coated pits, cyan). Box and line plots were generated using ImarisVantage as described in Methods. $n = 8.38 \times 10^4$ surface objects.

Center shows median; box represents the first and third quartiles; and whiskers represent the minimum and maximum. Shaded regions for line plots denote the interquartile range around the median. **c**, Timelapse imaging of the formation of the vasculature over 300 min (zoomed-in rendering of the box in **a**) at 0 (i), 100 (ii), 200 (iii) and 300 min (iv). The sample depicted is representative of ten experimental sessions each with two to three biological replicates, yielding similar results.

The line intensity profiles in HyU present reduced noise and represent more closely the expected distribution of signals (Fig. 3g,h). The visible micro-patterns of actin on the membrane of the periderm suggest that the improvements quantified with synthetic data are maintained in live samples' signals and geometrical patterns of microridges[42]. In contrast, noise corruption and the presence of misplaced signals are characterized in the results from LU, with high-frequency intensity variations that mismatch both the labeling and biological patterns.

HyU is more accurate and results in more reliable unmixing results across the depth of sample with greatly reduced unmixing residuals. The average residual for HyU is ninefold lower than that of LU with a threefold narrower variance. (Fig. 3i and Supplementary Fig. 8). This reduction in the residual is consistent with increasing *z* depth, where HyU unmixing results stably maintain both lower residuals and variance on average. These reduced residuals correspond both to a mathematically more precise and more uniform decomposition of signals as illustrated by the distribution of residuals versus photons (Supplementary Figs. 8e,f and 14).

We utilized HyU's increased sensitivity to overcome common challenges of multiplexed imaging, such as poor photon yield and spectral cross-talk and were able to visualize dynamics in a developing zebrafish embryo. We used a triple-transgenic zebrafish embryo with labeled pan-endothelial cells, vasculature and clathrin-coated pits (*Tg(fli1:mKO2)*; *Tg(kdrl:mCherry)*; *Gt(cltca-citrine)*). Multiplexing these spectrally close fluorescent proteins is enabled by HyU's increased sensitivity at lower photon counts. The increased performance at lower SNR allowed us to maintain high quality results (Fig. 4 and Supplementary Video 2) while performing faster acquisitions and reducing photo-damage through lower excitation laser power and pixel dwell

time. Decreased experimental requirements allow for tiling of larger volumes and extending the field of view, while still providing enough time resolution for developmental events, even with a high number of multiplexed fluorescent signals. The timelapses include the simultaneous acquisition of clathrin, kdrl and fli1, enabling visualization of the formation of ventral vasculo-endothelial protrusions, while tracking the development of vesicles and vasculature. HyU enables comparative quantifications of spatiotemporal features, allowing for the determination of volumetric changes over lengthy timelapses, in this case, over the course of 300 min (Fig. 4b)[43,44].

### Analysis of both Intrinsic and extrinsic fluorescent signals

HyU provides the ability to combine the information from intrinsic and extrinsic signals during live imaging of samples, at both single (Fig. 5) and multiple time points (Fig. 6). The graphical representation of phasors allows identification of unexpected intrinsic fluorescence signatures in a quadra-transgenic zebrafish embryo *Gt(cltca-citrine);Tg (ubiq:lyn-tdTomato;ubiq:Lifeact-mRuby;fli1:mKO2)*, imaged with single photon (488 and 561 nm excitation) (Fig. 5a–d). The elongated distribution on the phasor (Fig. 5c) highlights the presence of an additional, unexpected spectral signature, related to strong sample autofluorescence (Fig. 5d, blue). HyU analysis of the sample, inclusive of this additional signal, provides separation of the contributions of five different fluorescent spectra with residual 3.9 ± 0.3%. HyU allows for reduced energy load, tiled imaging of the entire embryo without perturbing its development or depleting its fluorescence signal (Fig. 5a). The higher speed, lower power imaging allows for subsequent re-imaging of the same sample, as we report in the zoomed high-resolution acquisitions of the head section (Fig. 5b,e).

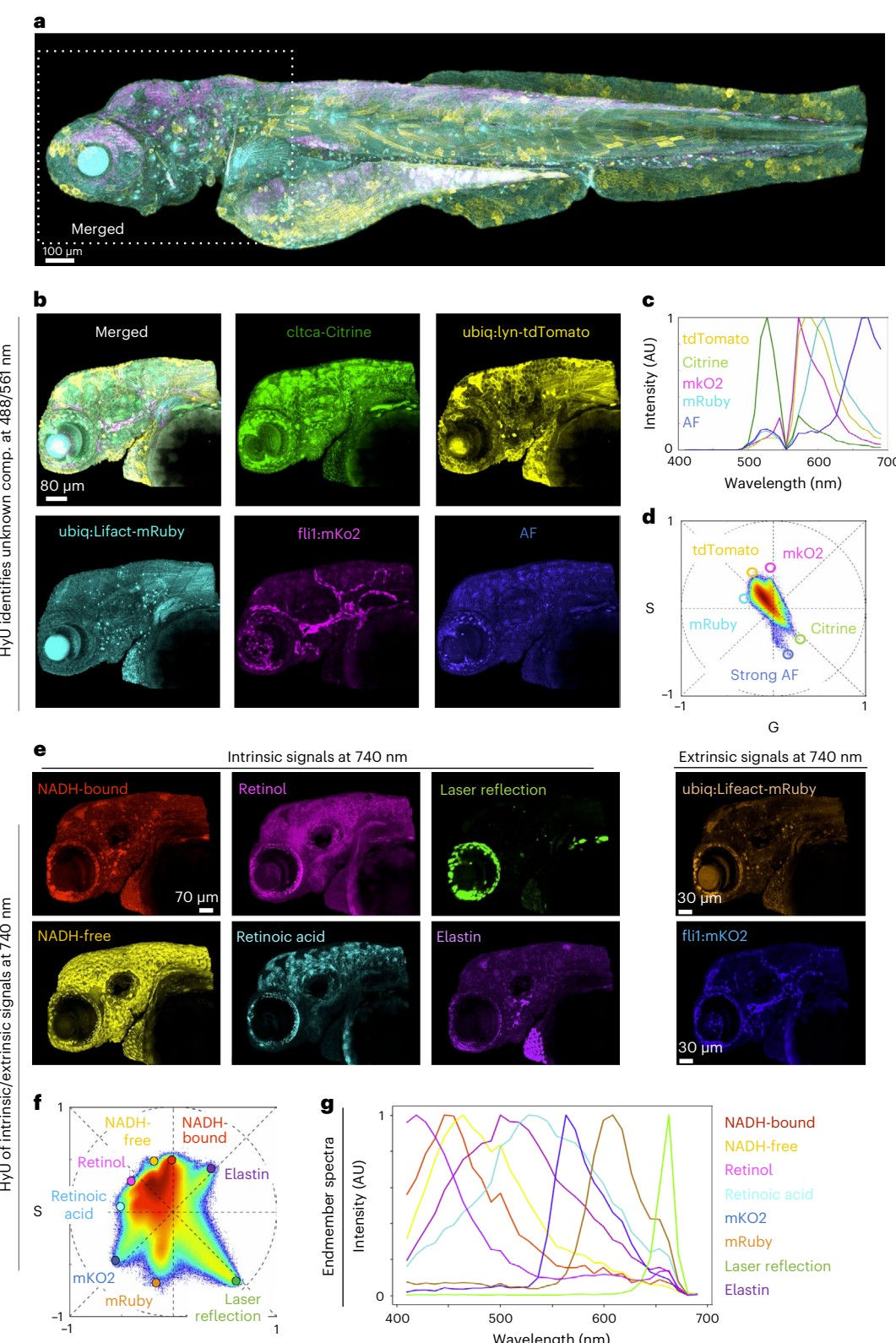

**Fig. 5 | HyU enables identification and unmixing of low-photon intrinsic signals in conjunction with extrinsic signals. a**, HyU results of a quadra-transgenic zebrafish *Gt(cltca-citrine);Tg(ubiq:lyn-tdTomato;ubiq:Lifeact-mRuby;fli1:mKO2)* imaged over multiple tiles. **b**, HyU results of the head region (box in **a**). Scale bar, 80 µm. AF, autofluorescence. **c,d**, The input spectra (**c**) required to perform the unmixing are easily identified on the phasor plot (**d**) when visualizing each spectrum as a spatial location. AU, arbitrary units. **e**, The zoomed-in acquisition of the head region of the embryo (box in **a**) after application of HyU. Scale bar, 70 µm. **f**, Phasor plot representation of eight

independent fluorescent fingerprint locations. **g**, The spectra corresponding to each of the eight independent spectral components are also provided as reference. Colors in **f** match renderings in **e** and **g**: NADH-bound (red), NADH-free (yellow), retinoid (magenta), retinoic acid (cyan), reflection (green), elastin (purple) and extrinsic signals, mKO2 (blue) and mRuby (orange). All signals were excited with a single-photon laser (**a**–**d**) at both 488 nm and 561 nm or a two-photon laser (**e**–**g**) at 740 nm. The sample depicted is representative of 28 (for **a**,**b**) and 24 (for **e**) experimental sessions each with three to five biological replicates, yielding similar results.

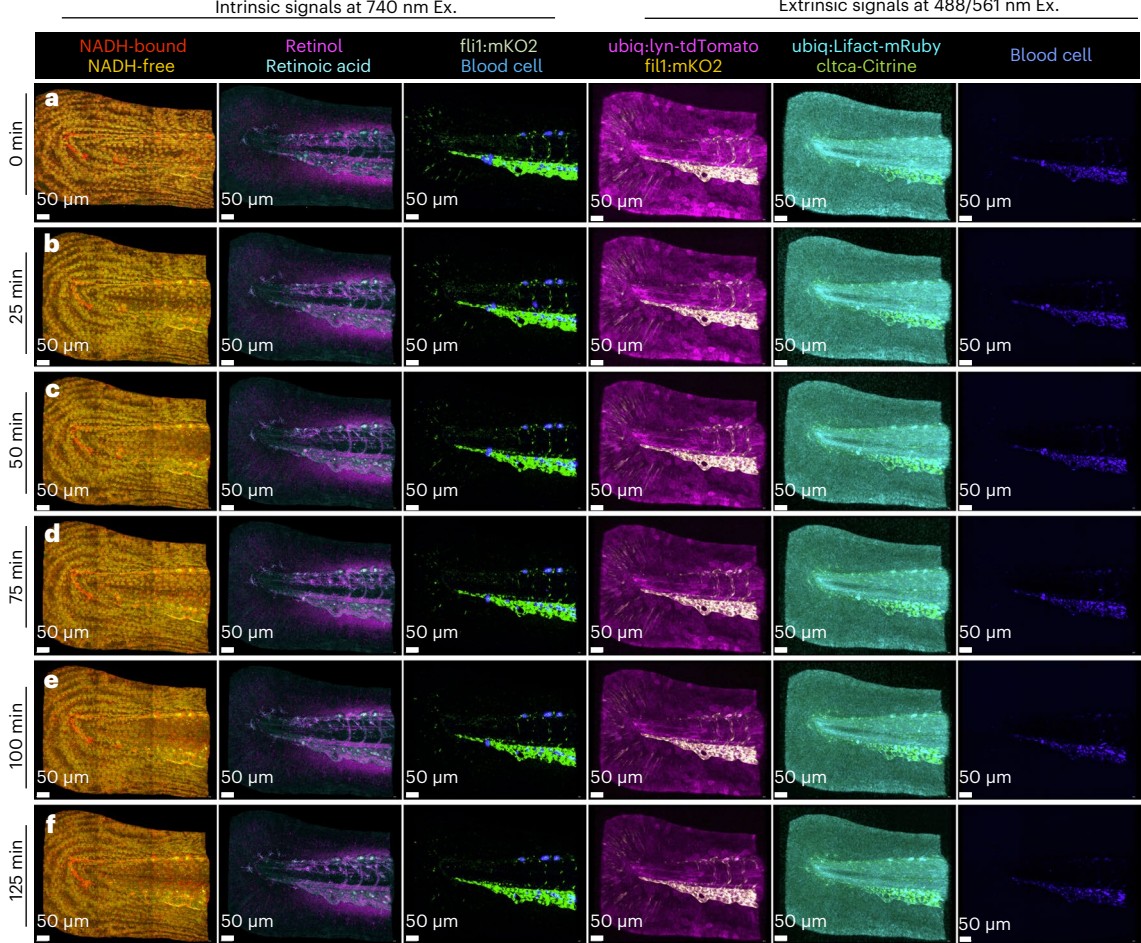

**Fig. 6 | HyU pushes the upper limits of live multiplexed volumetric timelapse imaging of intrinsic and extrinsic signals.** HyU's increased sensitivity provides a simple solution for the challenging task of imaging timelapse data at six time points (125 min) for both intrinsic signals and extrinsic signals of a quadra-transgenic zebrafish: *Gt(cltca-citrine);Tg(ubiq:lyn-tdTomato;ubiq:Lifeact-mRuby;fli1:mKO2)*. **a–f**, Volumetric renderings of HyU results for time points acquired at 25 min intervals reveal the high-contrast and high-multiplexed labels of NADH-bound (red), NADH-free (yellow), retinoid (magenta), retinoic acid

(cyan), mKO2 (green) and AF from blood cells (blue) when excited at 740 nm. Further extrinsic signals for mKO2 (yellow), tdTomato (magenta), mRuby (cyan), citrine (green) and blood cells AF (blue) are also readily unmixed using HyU when exciting the sample at 488/561 nm. HyU provides the capacity to simultaneously multiplex nine signals in a live sample over long periods of time, a previously unexplored task. Scale bars, 50 μm. The sample depicted is representative of 28 experimental sessions each with three to five biological replicates, yielding similar results.

With the ability to unmix low-photon signals, HyU enables imaging and decoding of intrinsic signals, which are inherently low light. Two-photon lasers are ideal for exciting and imaging blue-shifted intrinsic fluorescence from samples[45–48]. Here, the same quadra-transgenic sample is imaged using 740 nm excitation to access both intrinsic and extrinsic signals (Fig. 5e–g, Supplementary Note 2 and Supplementary Fig. 21). HyU enables unmixing of at least nine intrinsic and transgenic fluorescent signals (Fig. 5), recovering fluorescent intensities from labels illuminated at a suboptimal excitation wavelength (Fig. 5e). The spectra for intrinsic fluorescence were obtained from in vitro measurements and values reported in literature (Methods). For this sample the intrinsic signals arise from events related mainly with metabolic activity (nicotinamide adenine dinucleotide (NADH) and retinoids)[49–53], tissue structure (elastin)[54] and illumination (laser reflection) (Fig. 5e, Supplementary Figs. 22 and 26 and Supplementary Note 2). These results confirm our conclusion that HyU is a powerful tool for allowing the imaging and analysis of endogenous labels.

### Multiplexed timelapse acquisition
Finally, we exploited the HyU capabilities to multiplex volumetric timelapse of extrinsic and intrinsic signals by imaging the tail region

of the same quadra-transgenic zebrafish embryo. We excited extrinsic labels at 488/561 nm and the intrinsic signals with 740 nm two-photon microscopy, collecting six tiled volumes over 125 min (Fig. 6, Supplementary Figs. 9–11, 15, Supplementary Video 3 and Supplementary Note 2). HyU unmixing in this sample allows for distinction of nine signals, separating their contributions with sufficiently low requirements to allow repeated imaging of notoriously low SNR intrinsic fluorescence.

## Discussion
Our results reveal the advantages of HyU over more-conventional LU in performing complex multiplexing experiments. HyU overcomes the considerable challenges of separating multiple fluorescent and autofluorescent labels with overlapping spectra while minimizing sample exposure to excitation light.

The chief advantage of HyU is its multiplexing capability when imaging in the presence of biological and instrumental noise, especially at low signal levels. HyU increased sensitivity improves multiplexing in photon limited applications (Fig. 2f–l), in deeper volumetric acquisitions (Fig. 3i and Supplementary Fig. 23) and in signal-starved imaging of autofluorescence (Figs. 5e and 6). This improvement is demonstrated through increased contrast and reduced residuals. Our simulation

results (Fig. 2) demonstrate that HyU improves unmixing of spatially and spectrally overlapping fluorophores excited simultaneously. The increased robustness at low-photon imaging conditions reduces the imaging requirements for excitation levels and detector integration time, allowing for imaging with reduced photo-toxicity. Live imaging on multicolor samples performed at high sampling frequency enables improved tiling to increase the field of view (Figs. 3 and 4) while maximizing the usage of the finite fluorescent signals over time. Two-photon imaging of intrinsic and extrinsic signals suggests the ability of HyU to multiplex signals with large dynamic range differences (Fig. 5) extending multiplexed volumetric imaging into the time dimension (Fig. 6). Although improved, images with particularly low signals still present corruption (Supplementary Fig. 4), setting a reasonable range of utilization above eight photons per spectrum.

Simplicity of use and versatility are other key advantages of HyU, inherited from both the phasor approach[35] and traditional unmixing algorithms. Phasors here operate as a spectral encoder, reducing computational load and integrating similar spectral signatures in histogram bins of the phasor plot. This representation simplifies identification of independent spectral signatures (Fig. 5 and Supplementary Note 1) through both phasor plot selection and phasor residual mapping (Supplementary Fig. 11), accounting for unexpected intrinsic signals (Figs. 5 and 6, Supplementary Fig. 12 and Supplementary Note 2) in a semi-automated manner, while still allowing fully automated analysis by means of spectral libraries.

The simplicity of this approach is especially helpful in live imaging, where identifying independent spectral components remains an open challenge, owing to the presence of intrinsic signals (Supplementary Fig. 12 and Supplementary Note 1). Intrinsic signals are notoriously low in emitted photons, leading to an inability to unmix using traditional unmixing algorithms. High-SNR reference spectra can be derived from other experimental data or identified directly on the phasor. Selection of portions on the phasor plot allows for visualization of the corresponding spectra in the wavelength domain (Fig. 5c,d,f,g and Supplementary Fig. 27). This intuitive versatility allows for identification of both the number of unexpected signatures and their spectra, a task previously difficult to perform due to noise and lack of global visualization tools. In single-photon imaging (Fig. 5a–d), the HyU phasor allowed identification of a fifth distinct spectral component arising from a general autofluorescent background, thereby improving the unmixed results. In two-photon imaging, HyU enabled identification and multiplexing of eight highly overlapping signals possessing a wide dynamic range of intensities, between intrinsic and extrinsic markers (Fig. 5f,g). Combination of single- and two-photon imaging increased the number of multiplexed fluorophores to nine (Fig. 6), considering some of the extrinsic labels being excited with two-photon microscopy. Multiplexing of signals may be further improved by implementing HyU on fluorescent dyes.

HyU performs better than standard algorithms both in the presence and absence of phasor noise reduction filters[35]. Compared to LU, the unmixing enhancement when such filters[35] are applied is demonstrated by a decrease of the MSE of up to 21% (Fig. 2c), with a reduction of the average amount of residuals by sevenfold. Even in the absence of phasor denoising filters, HyU performs up to 7.3% better than the standard (Fig. 2d) based on MSE of synthetic data unmixing. This base improvement is due to the averaging of similarly shaped spectra in each phasor histogram bin, which reduces the statistical variability within the spectra used for the unmixing calculations (Fig. 1e). This averaging strategy works well for general fluorescence spectra owing to their broad and mostly unique spectral shape.

In the absence of noise, for example in the ground-truth simulations, LU produces an MSE that is sixfold lower than HyU (Supplementary Figs. 5b,c and 6g). In these noiseless conditions, the binning and averaging of spectra in the phasor histogram, without denoising, provides statistically indifferent values of error respect to LU, suggesting results of similar quality.

HyU can interface with different unmixing algorithms, adapting to existing experimental pipelines. We successfully tested hybridization with iterative approaches such as non-negative matrix factorization[55], fully constrained and non-negative least squares[56] (Methods). Speed tests with iterative fitting unmixing algorithms demonstrate a speed increase of up to 500-fold when the HyU compressive strategy is applied. (Supplementary Fig. 13 and Supplementary Note 3). Due to the initial computational overhead for encoding spectra in phasors, there is a twofold speed reduction for HyU in comparison to standard LU; however, this may be improved with further optimizations of HyU implementation or by implementing different types of encoding.

One restriction of HyU derives from the mathematics of LU, where linear equations representing the unmixed channels need to be solved for the unknown contributions of each analyzed fluorophore. To obtain a unique solution from these equations and to avoid an underdetermined equation system, the maximum number of spectra for unmixing may not exceed the number of channels acquired[57], which is generally 32 for commercial microscopes. This number could be increased; however, due to the broad and photon-starved nature of fluorescence spectra, acquisition of a larger number of channels could negatively affect the sample, imaging time and intensities. Depending on the number of labels in the specimen of interest, extending the number of labels to simultaneously unmix beyond 32 will likely require spectral resolution upsampling strategies.

HyU improvement is related to the presence of various types of signal disruption and noise in microscopy images, such as stochastic emission, Gaussian, Poisson and digital, as well as unidentified sources of spectral signatures that affect SNR in a variety of ways (Supplementary Fig. 5b,c and Supplementary Figs. 6g and 28). In the multiplexing of fluorescent signals, HyU offers improved performance, quality and speed in the low-signal regime. HyU is an improvement compared to previously published phasor analysis (Supplementary Figs. 24 and 25 and Supplementary Note 4) and the current gold standard LU under multiple experimental conditions of low SNR (Supplementary Figs. 16 and 17) reduced number of channels (Supplementary Figs. 18 and 19) in the case of fluorescent signals as well as combination of multiple fluorescent and autofluorescent signals (Supplementary Fig. 20). HyU is poised to be used in the context of in vivo imaging, collecting information from samples labeled at an endogenous level even in scattering mammal samples (Supplementary Figs. 29 and 30).

HyU is fully compatible with any commercial and common microscopes capable of spectral detection, facilitating access to the technology. Our analysis demonstrates HyU's robustness, simplicity and improvement in identifying both new and known spectral signatures and vastly improved unmixing outputs, providing a much-needed tool for delving into the many questions still surrounding studies with live imaging.

## Online content

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

## Methods

### Zebrafish lines

Adult fish were raised and maintained as described[58] in strict accordance with the recommendations in the Guide for the Care and Use of Laboratory Animals by the University of Southern California, where the protocol was approved by the Institutional Animal Care and Use Committee (permit no. 12007 USC). Upon crossing appropriate adult lines, the embryos obtained were raised in Egg Water (60 µg ml$^{-1}$ Instant Ocean and 75 µg ml$^{-1}$ CaSO$_4$ in Milli-Q water) at 28.5 °C with addition of 0.003% (w/v) 1-phenyl-2-thiourea around 18 hours post fertilization (h.p.f.) to reduce pigment formation.

Transgenic *Gt(cltca-citrine)*$^{ct116a}$ line is a genetrap of clathrin, heavy polypeptide a, labeling transport vesicles with heightened expression in the vasculature[59]. *Tg(kdrl:mCherry)* labels the vasculature and was a kind gift from C.-L. Lien (Children's Hospital Los Angeles). *Tg(fli1:mKO2)*$^{ct641ca}$ labels pan-endothelial cells in both blood vessels and lymphatics as previously reported[30]. *Tg(ubiq:lyn-tdTomato)* labels all cell membrane by expression of *lyn-tdTomato* from the ubiquitin promoter, whereas *Tg(ubiq:Lifeact-mRuby)* labels actin by expression of *LifeAct-mRuby* fusion from the ubiquitin promoter.

The mpv17a9/a9;mitfaw2/w2 (casper) line was purchased from the Zebrafish International Resource Center and csf1rj4e1/j4e1 (panther) line[60] was a kind gift from D. Parichy (University of Virginia). We crossed casper with panther to produce triple heterozygote mpv17a9/+;mitfaw2/+;csf1rj4e1/+ F1 generation fish, which were subsequently incrossed to produce F2 generation with 27 combinations of mutational state of these genes. As csf1rj4e1 phenotype was not clear in F2 adult with casper phenotype, we outcrossed these fish with panther fish to determine the zygosity of csf1rj4e1 mutation based on the frequency of larva with xanthophores (the heterozygote and homozygote produced a 50% and 0% fraction of xanthophore-positive larva, respectively) by fluorescence microscopy. The casper;csf1rj4e1/j4e1 line is viable and reproducible; we outcrossed either casper;csf1rj4e1/j4e1 line or casper;csf1rj4e1/+ line with other fluorescent transgenic lines over several generations to obtain fish harboring multiple transgenes on the casper background either in the presence or absence of xanthophores.

### Mouse lines

Mice imaging was approved by the Institutional Animal Care and Use Committee of University of Southern California, protocol no. 20847. Experimental research on vertebrates complied with institutional, national and international ethical guidelines. Animals were kept on a 12-h light–dark cycle. Animals were breathing double filtered air, temperature in the room was kept at 70–73 °F, with humidity at 50% and cage bedding was changed biweekly. All these factors contributed to minimize intra- and inter-experiment variability. Adult Balb-c mice (Charles River Laboratories) were killed via overdose of isoflurane followed by cervical dislocation. Organs were quickly collected from the mice, washed in phosphate-buffered saline (PBS) and cut longitudinally alongside the mid-section to expose the inner part of the organ. The two halves of the organ were arranged onto a microscope slide for imaging.

### Zebrafish sample preparation

Transgenic zebrafish lines were intercrossed over multiple generations to obtain embryos with multiple combinations of the transgenes. All lines were maintained as heterozygous for each transgene. Embryos were screened using a fluorescence stereo microscope (Axio Zoom, Carl Zeiss) for expression patterns of individual fluorescence proteins before imaging experiments. A confocal microscope (LSM 780, Carl Zeiss) was used to isolate *Tg(ubiq:Lifeact-mRuby)* lines from *Tg(ubiq:lyn-tdTomato)* lines by distinguishing spatially and spectrally overlapping signals.

For in vivo imaging, 5–6 zebrafish embryos at 18–72 h.p.f. were immobilized and placed into 1% UltraPure low-melting-point agarose (catalog no. 16520-050, Invitrogen) solution prepared in 30% Danieau (17.4 mM NaCl, 210 M KCl, 120 M MgSO$_4$7H$_2$O, 180 M Ca(NO$_3$)$_2$ and 1.5 mM HEPES buffer in water, pH 7.6) with 0.003% 1-phenyl-2-thiourea and 0.01% tricaine in an imaging dish with no. 1.5 coverglass bottom (catalog no. D5040P, WillCo Wells). Following solidification of agarose at room temperature (1–2 min), the imaging dish was filled with 30% Danieau solution and 0.01% tricaine at 28.5 °C.

### Fluorescent silica bead characterization

One fluorescent silica beads solution (Nanocs) labeled with Cy3 (Si500-S3-1, 0.5 ml, 0.5 µm, 1% solid, lot no. 1608BRX5) was characterized in its spectral fluorescence emission and physical size.

A 10× dilution in PBS of the beads was placed on a no. 1.5 imaging coverglass and spectrally characterized using spectral mode on a Zeiss LSM 780 laser confocal scanning microscope equipped with a 32-channel detector using 40×/1.1 W LD C-Apochromat Korr UV-VIS-IR lens utilizing a two-photon laser at 740 nm to excite fluorescence from the beads, using a 690 nm lowpass filter to separate excitation and fluorescence. Spectra obtained from multiple beads with the same label were averaged, producing the reference spectrum reported in Supplementary Fig. 30g (dashed line). Fluorescent silica bead size and concentration were determined via nanoparticle tracking analysis on the NanoSight NS300 (Malvern Panalytical). Samples were run five times and results averaged for final size and concentration values reported.

### Mouse sample preparation

For autofluorescent measurements, mouse organ samples were collected from Balb-c mice. Following euthanasia, organs were resected and washed in PBS to remove residual blood and kept in PBS until imaging preparation. Organs were sectioned to image the internal architecture and mounted on a glass imaging dish with sufficient PBS to avoid dehydration of the sample. Following imaging, all samples were fixed in a 10% neutral buffered formalin solution at 4 °C.

For ex vivo bead characterization in tissue, mouse organ samples were collected from Balb-c mice. Following euthanasia, organs were resected and washed in PBS followed by incubation for at least 24 h in 10% buffered formalin. The kidney was then removed from the fixative and sectioned into smaller ~5 × 5 × 5 mm pieces for imaging. A fluorescent silica beads working solution (Nanocs) labeled with Cy3 (Si500-S3-1, 0.5 ml, 0.5 µm, 1% solid, lot no. 1608BRX5) and previously characterized was prepared using a 10× dilution of the fluorescent beads from their stock concentration. Beads were injected in the sample using 50 µl of the solution loaded into a 0.5 ml syringe with a 28 g needle. The kidney sections were then placed in imaging dishes with a small volume of PBS to keep the samples hydrated before imaging.

### Image acquisition

Images were acquired on a Zeiss LSM 780 laser confocal scanning microscope equipped with a 32-channel detector using 40×/1.1 W LD C-Apochromat Korr UV-VIS-IR lens at 28 °C.

Samples of *Gt(cltca-citrine)*, *Tg(ubiq:lyn-tdTomato)*, *Tg(fli1:mKO2)* and *Tg(ubiq:Lifeact-mRuby)* were simultaneously imaged with 488 nm and 561 nm laser excitation, for citrine, tdTomato, mKO2 and mRuby. A narrow 488 nm/561 nm dichroic mirror was used to separate excitation and fluorescence emission. Samples were imaged with a two-photon laser at 740 nm to excite autofluorescence, using a 690 nm lowpass filter to separate excitation and fluorescence.

Samples of mouse kidney tissue were imaged with two-photon excitation at 740 nm or 850 nm with a 690+ nm lowpass filter, at 37 °C incubation.

For all samples, detection was performed at the full available range (410.5–694.9 nm) with 8.9 nm spectral binning.

Supplementary Table 1 provides the detailed description of the imaging parameters used for all images presented in this work.

## Hyperspectral fluorescence image simulation

The model simulates spectral fluorescent emission by generating a stochastic distribution of photons with profile equivalent to the pure reference spectra (as described in Supplementary Note 1). The effect of photon starvation, commonly observed on microscopes, is synthetically obtained by manually reducing the number of photons in this stochastic distribution. Detection, Poisson and signal transfer noises are then added to produce 32-channel fluorescence emission spectra that closely resemble those acquired on microscopes. The simulations include accurate integration of dichroic mirrors and imaging settings.

## Simulation types

**Biologically comparable simulations.** To quantify the performance of HyU versus LU for microscopy data acquired experimentally, we generated synthetic data where each input spectra were organized with intensity distributions taken from experimental data of fluorescently labeled biological samples. We calibrated the analog-(DLs)-to-photon counting rate based on existing literature[61,62]. Experimental data were discretized to photons to produce biologically relevant photon masks with distributions of signals highly resembling those of the samples. This provided intensities and ratios that closely resemble those acquired from a confocal microscope, while allowing control over the effects of photon starvation.

**Spatially and spectrally overlapping simulations.** We also included simulations to quantify the performance of HyU versus LU with respect to the number of spectral combinations and of end-members. The results are summarized in Supplementary Figs. 16–19 in matrices of spectral overlap (0–100%, steps of 10%, x axis) by number of end-members (2–8 end-members, y axis) representing the relative MSE (Supplementary Methods; Performance quantification). Each relative MSE value reported in a matrix is the average of analysis of a 1,024 × 1,024 pixel image simulation with a spectral dimension of 32 channels matching the spectral range and bandwidth of the detectors in commercial confocal microscopes (LSM 780, Carl Zeiss). These simulations were created with artificial intensity distributions so that a simulation with x% overlap and n fluorophores would have x% of pixels with a randomized ratio of n input spectra. As an example, for a simulation with six fluorophores and 50% overlap, the simulated dataset would have 50% of the pixels contain a randomized combination of the six fluorophores, while the remaining pixels contain a single fluorophore. This allowed us to investigate the effects of an increasing number of spectral combinations on the compressive nature of the phasor method for HyU.

## Image analysis

**Independent spectral signatures.** Independent spectral fingerprints can be obtained through samples, solutions, literature or spectral viewer websites (Thermo Fisher, BD Spectral Viewer and Spectra Analyzer). Fluorescent signals used in this paper were obtained by imaging single-labeled samples in areas morphologically and physiologically known to express the specific fluorescence (Supplementary Fig. 21). For each dataset a phasor plot was computed. The 32-channel spectral fingerprint was extracted from the phasor bin at the count-weighted average position of the phasor cluster. Those fingerprints were compared to literature fingerprints and manually corrected to reduce noise. Further descriptions for how to identify new components can be found in Supplementary Note 1 and Supplementary Figs. 11 and 27.

For AF signals, the spectrum for elastin was obtained experimentally and compared to the literature[54]. Spectra for NADH-free, NADH-bound, retinoic acid, retinol and flavin adenine dinucleotide (FAD) were acquired from in vitro solutions using the microscope. NADH-free was B-NAD (Sigma-Aldrich, 43420) in PBS solution. NADH-bound was B-NAD and L-lactic dehydrogenase (Sigma-Aldrich, 43420, L3916) in PBS. Retinoic acid was a solution of retinoic acid

(Sigma-Aldrich, R2625) in dimethylsulfoxide. Retinol was a solution of synthetic retinol (Sigma-Aldrich, R7632) in dimethylsulfoxide. FAD was FAD disodium salt hydrate (Sigma-Aldrich, F6625) in PBS.

**Phasor analysis.** For each pixel in a dataset, the Fourier coefficients of its normalized spectra define the coordinates $(G(n), S(n))$ in the phasor plane, where:

$$G(n) = \frac{\sum_{\lambda_s}^{\lambda_f} I(\lambda) \cos(n\omega\lambda)\Delta\lambda}{\sum_{\lambda_s}^{\lambda_f} I(\lambda) \Delta\lambda} \quad (1)$$

$$S(n) = \frac{\sum_{\lambda_s}^{\lambda_f} I(\lambda) \sin(n\omega\lambda)\Delta\lambda}{\sum_{\lambda_s}^{\lambda_f} I(\lambda) \Delta\lambda} \quad (2)$$

$$\omega = \frac{2\pi}{c} \quad (3)$$

Where $\lambda_s$ and $\lambda_f$ are starting and ending wavelengths respectively; $I$ is the measured intensity; $c$ is the number of spectral channels (32 in our case) and $n$ is the harmonic number[63]. In this work, we utilized the first harmonic ($n = 1$) for the autofluorescent signals and the second harmonic ($n = 2$) for fluorescent signals based on the sparsity of independent spectral components. A 2D histogram with dimensions ($S$ and $G$) is applied to the phasor coordinates to group pixels with similar spectra within a single square bin. We define this process as phasor encoding.

**LU.** The hypothesis for LU in this work is that given $i$ independent spectral fingerprints ($fp$), each collected spectrum ($I(\lambda)$) is a linear combination of $fp$ and the sum of each $fp$ contribution ($R$) is 1:

$$I(\lambda) = W_1 R_1 fp_1 + W_2 R_2 fp_2 + \ldots + W_i R_i fp_i + N \quad (4)$$

$$\Sigma R_i = 1 \quad (5)$$

where $R_i$ is the ratio, $W_i$ the weight and $N$ the noise. The acquired spectra are collected in the original spectral cube with shape ($t,z,c,y,x$), with $t$ as time, $c$ as channel and $x,y,z$ spatial dimensions.

$i$ spectral vectors, $fp_i$, need to be provided to the unmixing function. It is assumed that there are identical weights for all $fp$ and a low value for noise $N$. Under these conditions, we obtain $R_i$ by applying a Jacobian matrix inversion:[64]

$$\begin{bmatrix} \sum_x w(x) \frac{\partial f^0}{\partial \alpha_1} \frac{\partial f^0}{\partial \alpha_1} & \sum_x w(x) \frac{\partial f^0}{\partial \alpha_1} \frac{\partial f^0}{\partial \alpha_2} & \cdots \\ \sum_x w(x) \frac{\partial f^0}{\partial \alpha_2} \frac{\partial f^0}{\partial \alpha_1} & \sum_x w(x) \frac{\partial f^0}{\partial \alpha_2} \frac{\partial f^0}{\partial \alpha_2} & \cdots \\ \vdots & \vdots & \ddots \end{bmatrix} \begin{bmatrix} \alpha_1 - \alpha_1^0 \\ \alpha_2 - \alpha_2^0 \\ \vdots \end{bmatrix}$$

$$= \begin{bmatrix} \sum_x w(x) [y(x) - f^0(x)] \frac{\partial f^0}{\partial \alpha_1} \\ \sum_x w(x) [y(x) - f^0(x)] \frac{\partial f^0}{\partial \alpha_1} \\ \sum_x w(x) [y(x) - f^0(x)] \frac{\partial f^0}{\partial \alpha_1} \end{bmatrix} \quad (6)$$

In the pixel-by-pixel LU implementation in this work, Jacobian matrix inversion is applied on the acquired spectrum in each pixel with dimensions ($t,z,c,y,x$). Resulting ratios for each spectral vector are assembled in the form of a ratio cube with shape ($t,z,i,y,x$) where $x,y,z,t$ are the original image spatial and time dimensions, respectively and $i$ is the number of input spectral vectors. The ratio cube ($t,z,i,y,x$) is multiplied with the integral of intensity over channel dimension of the original spectral cube, with shape ($t,z,y,x$), to obtain the final resulting dataset with shape ($t,z,i,y,x$).

**Hybrid unmixing: LU.** In the HyU implementation, Jacobian matrix inversion is applied on the average spectrum of each phasor bin with dimensions $(c,s,g)$ where $g$ and $s$ are the phasor histogram sizes and $c$ is the number of spectral channels acquired. The average spectrum in each bin is calculated by using the phasor as an encoder, to reference each original pixel spectra to a bin. Resulting ratios for each component channel are assembled in the form of a phasor bin-ratio cube with shape $(i,s,g)$ where $i$ is the number of input independent spectra $fp$ (LU section). This phasor bin-ratio cube is then referenced to the original image shape, forming a ratio cube with shape $(t,z,i,y,x)$ where $x,y,z,t$ are the original image dimensions. We multiply the ratio cube with the integral of intensity over channel dimension of the original spectral cube, with shape $(t,z,y,x)$, obtaining a final result dataset with shape $(t,z,i,x,y)$.

**HyU algorithm.** The pseudo-code utilized for the HyU algorithm is as follows:

> Input: *I(x,y,c,z,t) (5D hyperspectral image)*
>> *U(i,c) (reference spectra (n spectra))*
> Output: *I_U(x,y,i,z,t) (multichannel unmixed image)*
> Procedure:
>> *HYU(I(x,y,c,z,t), U(n, c))*
>> *// Single harmonic Fourier transform*
>> *G(x,y,z,t), S(x,y,z,t) = phasor_transform(I(x,y,c,z,t))*
>> *// 2D histogram of G and S values*
>> *H(g,s) = histogram2d(G(x,y,z,t) S(x,y,z,t))*
>> *// Averaging of hyperspectral image over phasor histogram*
>> *I_H(g,s,c) = phasor_average(I(x,y,c,z,t), H(g,s))*
>> *// Linear Unmixing of averaged hyperspectral image*
>> *I_U(g,s,i) = LU(I_H(g,s,c), U(i,c))*
>> *// Reference unmixed phasor image back to original image dimensions*
>> *I_U(x,y,i,z,t) = reverse_phasor_reference(I_U(g,s,i)*
>> *return I_U(x,y,i,z,t)*

**Other unmixing algorithms.** Unmixing algorithms utilized for speed comparisons with the HyU algorithm (Supplementary Fig. 13) were plugged in the unmixing step of the analysis pipeline and sourced as follows. Non-negative constrained least squares and fully constrained least squares from pysptools.abundance_maps (https://pysptools.sourceforge.io/abundance_maps.html). Robust non-negative matrix factorization[55] Python implementation was obtained from (https://github.com/neel-dey/robust-nmf)

**Data visualization.** Rendering of final result datasets were performed using Imaris v.9.5–9.7. In Figs. 2 and 3, contrast settings (minimum, maximum and gamma) for each channel were set to be equal to provide reasonable comparison between HyU and LU results. Gamma was set to 1, no minimum threshold was applied and the maximum for each channel was set to one-third of the maximum intensity. The images were rendered using maximum intensity projection and for improving display, they were digitally resampled in the $z$ direction, maintaining a fixed $xy$ ratio to attenuate the gap generated from sparse sampling $z$-wise on the microscope.

**Box plot generation.** All box plots were generated using standard plotting methods. The center line corresponds to the median, the lower box border corresponds to the first quartile and the upper box border corresponds to the third quartile. The lower and upper whiskers correspond to 1.5× the interquartile range below and above the first and third quartiles, respectively.

**Timelapse registration.** A customized Python script (Supplementary Code) was first utilized to pad the number of z slices across multiple time points, obtaining equally sized volumes. The 'Correct 3D drift' plugin[65] (https://imagej.net/Correct_3D_Drift) in FIJI[66] (https://imagej.net/Fiji) was used to register the data.

**Timelapse statistics.** Box plots and line plots for timelapses were generated using ImarisVantage in Imaris v.9.5–9.7. Box plot elements follow the same guidelines as described above. Line plots are connected box plots for each time point with the solid line denoting the median values and the shaded region denoting the first and third quartiles.

### Reporting summary

Further information on research design is available in the Nature Portfolio Reporting Summary linked to this article.

### Data availability

All the relevant data are available from the corresponding author upon reasonable request. Datasets for Figs. 1–6 and simulations are available for download at http://bioimaging.usc.edu/software.html#sampledatasets in the samples section.

### Code availability

All the relevant code is available from the corresponding author upon reasonable request. Software and instructions can be downloaded from http://bioimaging.usc.edu/software.html.

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

### Acknowledgements

The authors thank F. Schneider, S. Restrepo (Translational Imaging Center, University of Southern California), C.-L. Chiu (Chan Zuckerberg Initiative) and S. Ojosnegros (Institute for Bioengineering of Catalonia) for helpful discussions. This material is based upon work supported by the National Science Foundation Graduate Research Fellowship under grant DGE-1842487, Department of Defense PR150666 and University of Southern California.

### Author contributions

H.J.C., D.E.S.K. and F.C. analyzed the results and wrote the software. H.J.C., D.E.S.K., J.R.U. and F.C. provided conceptualization. H.J.C., D.E.S.K., M.K., F.C. and L.A.T. helped in the experimental design and data analysis. M.K. generated the inducible and mKO2 zebrafish

lines. H.J.C. and F.C. acquired data. S.E.F. provided supervision. H.J.C., D.E.S.K. and F.C. wrote the paper. L.A.T., J.R.U. and M.K. supported review and editing. S.B. and C.Z. provided mouse samples and helped in mouse experimental design.

## Competing interests

The University of Southern California has filed a provisional patent application covering this method listing H.J.C., D.E.S.K., J.R.U., S.E.F. and F.C. as inventors.

## Additional information

**Correspondence and requests for materials** should be addressed to Francesco Cutrale.

# nature research

# Reporting Summary

Nature Research wishes to improve the reproducibility of the work that we publish. This form provides structure for consistency and transparency in reporting. For further information on Nature Research policies, see our Editorial Policies and the Editorial Policy Checklist.

## Statistics

For all statistical analyses, confirm that the following items are present in the figure legend, table legend, main text, or Methods section.

| n/a | Confirmed | |
|---|---|---|
| ☐ | ☒ | The exact sample size (*n*) for each experimental group/condition, given as a discrete number and unit of measurement |
| ☐ | ☒ | A statement on whether measurements were taken from distinct samples or whether the same sample was measured repeatedly |
| ☐ | ☒ | The statistical test(s) used AND whether they are one- or two-sided<br>*Only common tests should be described solely by name; describe more complex techniques in the Methods section.* |
| ☒ | ☐ | A description of all covariates tested |
| ☒ | ☐ | A description of any assumptions or corrections, such as tests of normality and adjustment for multiple comparisons |
| ☐ | ☒ | A full description of the statistical parameters including central tendency (e.g. means) or other basic estimates (e.g. regression coefficient) AND variation (e.g. standard deviation) or associated estimates of uncertainty (e.g. confidence intervals) |
| ☐ | ☒ | For null hypothesis testing, the test statistic (e.g. *F*, *t*, *r*) with confidence intervals, effect sizes, degrees of freedom and *P* value noted<br>*Give P values as exact values whenever suitable.* |
| ☒ | ☐ | For Bayesian analysis, information on the choice of priors and Markov chain Monte Carlo settings |
| ☒ | ☐ | For hierarchical and complex designs, identification of the appropriate level for tests and full reporting of outcomes |
| ☒ | ☐ | Estimates of effect sizes (e.g. Cohen's *d*, Pearson's *r*), indicating how they were calculated |

*Our web collection on statistics for biologists contains articles on many of the points above.*

## Software and code

Policy information about availability of computer code

| | |
|---|---|
| Data collection | Commercial software Zen Black 2.3 SP1 FP3, Carl Zeiss, for controlling Zeiss 780 Microscope. |
| Data analysis | Data was preprocessed utilizing custom python algorithms now integrated in HySP (http://bioimaging.usc.edu/software.html#HySP), data was rendered using commercial software Imaris 9.5-9.7. 3D registration was done using the Correct 3D Drift plugin in FIJI (Java 8 Life-Line version). Simulations were made using custom Python code, starting from real data.<br>All the relevant code is available from the corresponding author upon reasonable request. Software and instructions can be downloaded from http://bioimaging.usc.edu/software.html. |

For manuscripts utilizing custom algorithms or software that are central to the research but not yet described in published literature, software must be made available to editors and reviewers. We strongly encourage code deposition in a community repository (e.g. GitHub). See the Nature Research guidelines for submitting code & software for further information.

## Data

Policy information about availability of data

All manuscripts must include a data availability statement. This statement should provide the following information, where applicable:
- Accession codes, unique identifiers, or web links for publicly available datasets
- A list of figures that have associated raw data
- A description of any restrictions on data availability

Data and materials availability: All the relevant data are available from the corresponding author upon reasonable request. Datasets for Figs. 1–6 and simulations are available for download at http://bioimaging.usc.edu/software.html#sampledatasets in the samples section.

# Field-specific reporting

Please select the one below that is the best fit for your research. If you are not sure, read the appropriate sections before making your selection.

☒ Life sciences ☐ Behavioural & social sciences ☐ Ecological, evolutionary & environmental sciences

For a reference copy of the document with all sections, see nature.com/documents/nr-reporting-summary-flat.pdf

# Life sciences study design

All studies must disclose on these points even when the disclosure is negative.

| Sample size | No sample size was predetermined. The work here presented describes an analysis method which is run against different fluorescent samples including autofluorescence and multiple-fluorescent labels. Samples were acquired over multiple experimental sessions until a clear performance trend was evident. Performance was then validated across multiple scenarios with simulated data. |
|---|---|
| Data exclusions | No data was excluded from analysis |
| Replication | Reproducibility was verified across at least 5 different samples in different days. Datasets are available for download at http://bioimaging.usc.edu/software.html#sampledatasets in the samples section while software can be downloaded from http://bioimaging.usc.edu/software.html#HySP. |
| Randomization | Non-pertinent, no clinical trial involved. Sample order was not considered in this work as this is an analysis method which runs on one sample per time. Order does not affect how the method works. |
| Blinding | Blinding was not relevant to this study as this method returns multiple images as output which require no interpretation from the user beside the capability of distinguishing colors on a screen. |

# Reporting for specific materials, systems and methods

We require information from authors about some types of materials, experimental systems and methods used in many studies. Here, indicate whether each material, system or method listed is relevant to your study. If you are not sure if a list item applies to your research, read the appropriate section before selecting a response.

## Materials & experimental systems

| n/a | Involved in the study |
|---|---|
| ☒ | ☐ Antibodies |
| ☒ | ☐ Eukaryotic cell lines |
| ☒ | ☐ Palaeontology and archaeology |
| ☐ | ☒ Animals and other organisms |
| ☒ | ☐ Human research participants |
| ☒ | ☐ Clinical data |
| ☒ | ☐ Dual use research of concern |

## Methods

| n/a | Involved in the study |
|---|---|
| ☒ | ☐ ChIP-seq |
| ☒ | ☐ Flow cytometry |
| ☒ | ☐ MRI-based neuroimaging |

## Animals and other organisms

Policy information about studies involving animals; ARRIVE guidelines recommended for reporting animal research

| Laboratory animals | 18 to 72 hours-post-fertilization zebrafish embryos:<br>1. Gt(cltca-Citrine); Tg(ubiq:lyn-tdTomato; ubiq:Lifeact-mRuby; fli1:mKO2)<br>2. Gt(cltca-Citrine); Tg(kdrl:mCherry;fli1:mKO2)<br>3. Gt(cltca-Citrine); Tg(ubiq:lyn-tdTomato; ubiq:Lifeact-mRuby)<br>4. mpv17a9/a9;mitfaw2/w2 (casper) /csf1rj4e1/j4e1 (panther)<br>Detailed reporting of lines is in "Zebrafish lines" and Table 1 Supplementary Materials. All zebrafish lines are available from the authors.<br><br>7 month old female Balb-C mice Details are reported in "Mouse sample preparation" section and Table 1 in Supplementary Material. |
|---|---|
| Wild animals | This study did not involve wild animals |
| Field-collected samples | This study did not involve field-collected samples |
| Ethics oversight | For Zebrafish lines: IACUC of University of Southern California (permit number: 12007); For Mouse lines: IACUC of USC (permit number: 21311) |

Note that full information on the approval of the study protocol must also be provided in the manuscript.

