## [Peer Review File · Nature Methods]

HyU: Hybrid Unmixing for longitudinal in vivo imaging of low signal to noise fluorescenceREVIEWER COMMENTS

Reviewer #1 (Remarks to the Author: Overall significance):

The paper by Chiang et al titled “HyU: Hybrid Unmixing for longitudinal in vivo imaging of low signal to noise fluorescence” has a lot of details that are really attractive. This includes the possibility of multiplexing using a combination of Phasor and linear unmixing and potential for understanding the distribution pattern for multiple fluorophores, both endogenous and exogenous ones in the live system. The authors provide software based on this principle and detailed instruction to run the system. This is really attractive for the biological community as this does not require a specialized instrument and can use a hyperspectral detector with a linear array which is more commonly available to the researchers.

However, there is a major flaw that concerns me the most. The authors use only one harmonics to do phasor transformation and then linear unmixing, as stated in their mathematical arguments. This should be fine up to three components. For two components a pixel which has contribution of those components, it's phasor position is along the line joining those two components. In case of three it will be inside the triangle created by the three individual components. This isn't true for four components or higher. In four components, the position of an image phasor point can be either contributed to all four components or just three – as it will always be within a triangle. This has been shown previously by Dr. Enrico Gratton's group (PMID: 32235070). In that case another harmonic needs to be calculated to identify and explain if there are three or four components. The examples and analysis provided in this paper only involve a single harmonic – and that makes it impossible to specify the difference between image phasor points in the middle of the phasor cloud with multiple species (Fig 5F). My main concern is that use of a single harmonic assume that all of the components are present in an image pixel whose phasor coordinate is within the pentagon with the vertices occupied by the five components. It may be a very small contribution based on the fractional intensity – but it is there. This may or may not be true depending on pixel size, presence of components and basically the type of the sample. This argument needs to be resolved as I feel this is a fundamental approach to phasor and its limitation when only one harmonic is calculated.

Other comments and concerns follow below:

1. The authors provide a spectra.txt files for the unmixing. How was that obtained. How to calculate and save the data from samples having only one fluorophores?
2. I did not see any mention of collagen fluorescence when excited at 740 nm. That should be a component in the autofluorescence category.
3. What happens to FAD? That can be excited at 740 nm (PMID: 11964266).
4. About Bound NADH – does the spectra change based on the proteins it binds to?
5. What is the distance of separation of the pure components in the phasor space that allow for successful linear unmixing. I presume at some point the S/N will make it difficult if the spectra of the components are too close.
6. What about when the linear unmixing won't work? For example, a case where the individual components lie in line in the phasor space.
7. Coming back to Q1. – Prior knowledge of the components – How are they calculated? Where are the coordinates stored? Are only the center of the phasor cloud used or the whole distribution?
8. I do like the point mentioned in lines 10-20 in page 2. Spectral imaging and deconvolution is absolutely necessary.
9. Lines 30-40, page 2 – missing references about the different noise.
10. The linear deconvolution of the phasor space involves fractional intensity and not the actual fraction – something that I found missing in the discussion.
11. How do 5 photons/spectra work with Poisson statistics and associated uncertainty?
12. Does the Elastin spectrum change on crosslinking in a tissue compared to the solution?
13. Td-Tomato (PMID 19127988) and mRuby (PMID 23459413) can be excited with a 740 nm two-photon excitation. I am curious how the authors did not observe that in Figure 6.
14. What determines how much spatial denoising needs to be used? Spatial denoising indeed doesn't affect the intensity image – but it does affect the phasor mapped image – something that hasn't been discussed at all.

15. Figure 1 D-E – this is strictly not true. Once you transfer to phasor – the information remaining for the spectra is the FWHM and the peak/center – so how does the proper spectra being calculated in figure E?
16. The reduction of data from 10^7 to 10^4 . How much of that is related to spectral denoising and how much is related to the transformation to phasor?
17. Page 4 line 19 – after two-components – what happens with three/four and their possible combinations?
18. One of the uses of HyU is for low light level and long term imaging. What happens to the deconvolution if there is bleaching? This is a minor concern.
19. How to create the spectral libraries in the software provided by the authors (page 15, line 29-30)?
20. I do feel the references can be expanded for the phasor analysis of the multicomponent systems from other labs.

Reviewer #1 (Remarks to the Author: Impact):

The paper will influence the community - but the discrepancies need to be cleared and explained.

Reviewer #1 (Remarks to the Author: Strength of the claims):

The main concern is the linear additivity of phasor space and their implementation in this paper. Use of a single harmonic should not be enough for anything more than three components.

Reviewer #1 (Remarks to the Author: Reproducibility):

I do think the data is reproducible as the imaging is done using a commercial microscope and the authors provide an software to do so. There are details that is missing that need to be provided for the use. This includes calculation and storage of single components for the analysis.

Reviewer #2 (Remarks to the Author: Overall significance):

The manuscript of Hsiao Ju Chang et al (from the lab of Prof Cultrale) deals with dynamic (time-lapse) multiplexed imaging and offers a global-based solution for spectral unmixing of hyperspectral imaging data. Therefore, the authors improve the previously published algorithm HySP (Cultrale et al, Nat. Meth. 2017), which uses dimensionality reduction via the phasor approach (normalized discrete Fourier transformation of the hyperspectral 4D fluorescence data). They achieve this improvement by integrating in HySP a linear unmixing of the expected spectral signatures in the phase domain (HyU) - including both extrinsic signals (fluorescent proteins) and intrinsic signals (NAD(P)H, retinol, elastin, etc.). The dimensionality reduction of the phasor approach implies also a global analysis of the spectra (i.e. appreciates similarities of the spectra per voxel) and by that better deals with low signals. A thorough characterization of the laser, detector, read background noise and of their distribution type (Poisson, Gaussian, etc.) and implementation for denoising and additional reference-based preprocessing (SEER, Shi et al, Nat. Commun. 2020) improves not only the image quality but also the success of the hyperspectral unmixing of 8 or 9 emission (intrinsic and extrinsic) signals, at high computation speeds, as impressively demonstrated on simulated data and on time-lapse imaging data of multiple-reporter zebra fish larvae. While being of great interest for the live imaging community, in my opinion, the manuscript needs additional experimental, algorithmic and background (citation of previous work) information to unfold the full potential, as described in detail in the following.

Reviewer #2 (Remarks to the Author: Impact):

The relevance of the question/need for simultaneous spectrally multiplexed fluorescent microscopy to allow dynamic (time-lapse) multi-color imaging is tremendous, however, certainly going far beyond the field of developmental biology and zebra fish larvae imaging. This need has been previously recognized in the frame of

intravital multi-photon imaging (not hyperspectral), with impact for cancer research (Entenberg et al, 2011), immunology and neurosciences/neuroimaging, just to mention a few examples. Specifically, there have been solutions proposed and demonstrated for dynamic in vivo fluorescence imaging, including unmixing algorithms apart of the state-of-the-art linear unmixing (Rakhymzhan et al, Sci Rep 2017), in which up to 8 extrinsic and intrinsic signals are simultaneously distinguished, while dealing with low SNRs of multi-photon microscopy still remained a challenge. Including this information in the introduction is key, in order to demonstrate the potential general relevance of the present work and to awake a real interest for a broad readership.

In line with this, it is crucial to demonstrate the power of the presented algorithm for unmixing also intravital multi-photon imaging data in optically more challenging tissues and organisms, which need to deal with much lower signals and SNR values, especially due to massive scattering and wave-front distortions in mammal tissue.

Referring to the algorithm itself and to its characterization, the evolution from hyperspectral multiplexed imaging using the phasor approach HySP (Cultrale et al, 2017, Nat Meth), enhanced by preprocessing the data to account for various experimental noise via SEER (2020, Nat Commun) and finally by applying linear unmixing in the hyperspectral phase space, bringing additional significant accuracy to the unmixing capacity of the data is currently not clear in the manuscript and needs to be elaborated in the introduction, to emphasize the novelty of the present work.

Reviewer #2 (Remarks to the Author: Strength of the claims):

A. Referring to the broad applicability of the algorithm and the interest for a large community:

A.1. As previously mentioned, in order to prove the value of the approach presented in this manuscript, multiplexed time-lapse imaging in a mammal (adult mouse or rat or human) tissue is key and experimental data on this need to be added to the manuscript. I believe, one 4D (3D + time) imaging example showing 8-9 distinct emission signals would be absolutely convincing.

B. Referring to the unmixing approach:

B.1. In order to judge the added value of the integration of linear unmixing and of reference extrinsic and intrinsic spectral signatures on the performance of unmixing, a thorough comparison with the previously available HySP (Cultrale et al, 2017) needs to be provided, additionally to the comparison to state-of-the-art linear unmixing algorithms already included in the manuscript.

B.2. A central advantage of the here presented approach is the capacity of dealing even with low signals, i.e. unmixing even low endogenous signals, such as NAD(P)H, even free and bound – having extremely similar emission spectra (one reason why their fluorescence lifetime has been used to resolve the two states). The authors show the improvement referring to number of photons per spectrum, however, in order to judge the true improvement brought by the algorithm for real imaging data (which includes background with diverse types of noise distributions), the unmixing quality needs to be related to the signal-to-noise (SNR) ratio per voxel. While mentioning SNR in the text, no values or comparison are provided in this sense – it is important to mention how the SNR as such (not only the number of photons per spectrum) impacts on the spectral resolution, i.e. how similar can be two spectra at a certain SNR to be able to still resolve them?

B.3. A corner stone in acquiring better unmixing is the availability of appropriate reference spectral signatures. Whereas the current software provides the spectra necessary for the data shown in the manuscript and gives the opportunity for the users to identify signatures in their own data, the manuscript remains elusive of how the user can differentiate between a real spectral signature and different types of optical or electronical background and interferences – as well known from the use of the phasor approach in fluorescence lifetime imaging, a major challenge when dealing with experimental noisy imaging data in the frequency (phase) domain. The manuscript would benefit from including such a guide to validate the capacity for external use of the algorithm.

B.4. Finally, fully agreeing with the authors that the number of detectors may be varied, depending on the imaged sample type and on the excitation strategy, in order to acquire an emission signal at all, an analysis of how the number of detectors (channels) impacts on the resolution between different signatures (spectra) is needed also for less than 32 detectors (4 to 6 channels being the reality in many labs due to truly low fluorescence signals in deep tissue, e.g. of mice or of humans).

Reviewer #2 (Remarks to the Author: Reproducibility):

The current version of the HySP platform was easy to use and the provided sample data delivered similar results as those shown in the manuscript.

Reviewer #3 (Remarks to the Author: Overall significance):

In this report, Chiang and co-workers presented the Hybrid Unmixing (HyU) method for the efficient and robust analysis of multiple fluorescent signals. The authors employ the spectral phasor method for reducing spectral data dimension and denoising noises in the imaging system. The superiority of the proposed method has been demonstrated compared to the conventional linear unmixing method by exploiting computer simulation and experimental results. This article seems to be timely the report as increasing the biomedical applications using hyperspectral imaging methods. However, I found that there are some confusing points to be addressed clearly to publish this manuscript in Nature Portfolio.

Comments:

- 1) Hyperspectral phasor compresses spectral dimension by exploiting real and imaginary parts of Fourier transformation. Moreover, there were reports that hyperspectral phasor could be applied for multiplexed fluorescence imaging. If there are any advantages of combining phasor and spectral unmixing methods, please describe them clearly in the Introduction.
- 2) If I understood correctly, numbers of photons (For instance, 5 photons per spectral in the last paragraph in Introduction) were calculated from the computer simulation. If so, this quantitative value is significantly affected by the noise levels used in the simulation. Therefore, it would be good to add these values were obtained from the simulation for clarity.
- 3) The authors addressed that the HyU method is more computationally efficient than the linear spectral unmixing method. This is true as the spectral dimension was reduced in Hyperspectral Phasors and histogram binning. However, these spectral compression and denoising also require computational power. Does the proposed method is more efficient when the entire process is considered?
- 4) For spectral unmixing, it seems to use the reference signals obtained from pure fluorophores. What happens if there are unknown fluorescence signals? Can the proposed method be applied for blind spectral separation?
- 5) Following the previous question, I wonder about the effect of light scattering on the accuracy of the proposed method. In fig4, the proposed method can be applied for volumetric imaging. I wonder there are consistent fluorescence signals over the depth of tissue. Fluorescence signals occurred in deep tissue regions experience more light scattering, which might occur in spectral distortions.

Reviewer #3 (Remarks to the Author: Strength of the claims):

This work demonstrates the superiority of the proposed method using computer simulation and experimental data. The authors clearly claim that the proposed method is more efficient and robust than conventional linear spectral unmixing methods.

Reviewer #3 (Remarks to the Author: Reproducibility):

The authors provide the code and data used in the manuscript. This allows other people to reproduce these

results. And the dataset used in this work is appropriate for the purpose of the study.

Response to Reviewers' Comments

We appreciated the positive response to our manuscript submission. We would like to thank the reviewers and editors for constructive comments and suggestions that helped improve the manuscript. We have addressed the comments, performed additional quantifications and experiments and revised the re-submission accordingly.

Reviewer #1 (Remarks to the Author: Overall significance):

The paper by Chiang et al titled "HyU: Hybrid Unmixing for longitudinal in vivo imaging of low signal to noise fluorescence" has a lot of details that are really attractive. This includes the possibility of multiplexing using a combination of Phasor and linear unmixing and potential for understanding the distribution pattern for multiple fluorophores, both endogenous and exogenous ones in the live system. The authors provide software based on this principle and detailed instruction to run the system. This is really attractive for the biological community as this does not require a specialized instrument and can use a hyperspectral detector with a linear array which is more commonly available to the researchers.

However, there is a major flaw that concerns me the most. The authors use only one harmonics to do phasor transformation and then linear unmixing, as stated in their mathematical arguments. This should be fine up to three components. For two components a pixel which has contribution of those components, it's phasor position is along the line joining those two components. In case of three it will be inside the triangle created by the three individual components. This isn't true for four components or higher. In four components, the position of an image phasor point can be either contributed to all four components or just three – as it will always be within a triangle. This has been shown previously by Dr. Enrico Gratton's group (PMID: 32235070). In that case another harmonic needs to be calculated to identify and explain if there are three or four components. The examples and analysis provided in this paper only involve a single harmonic – and that makes it impossible to specify the difference between

image phasor points in the middle of the phasor cloud with multiple species (Fig 5F). My main concern is that use of a single harmonic assume that all of the components are present in an image pixel whose phasor coordinate is within the pentagon with the vertices occupied by the five components. It may be a very small contribution based on the fractional intensity – but it is there. This may or may not be true depending on pixel size, presence of components and basically the type of the sample. This argument needs to be resolved as I feel this is a fundamental approach to phasor and its limitation when only one harmonic is calculated.

ANSWER: We thank the reviewer for the insightful comments and for providing us an opportunity to clarify. In regard to the major concern, the use of a single harmonic, this hybrid approach uses the phasor as an aggregator of similar spectra while maintaining the wavelength dimension of the original data.

Spectral similarity unmixing. The unmixing is performed over the spectral dimension (in this case 32 channels), by aggregating similar spectra of the original data, as explained in Supplementary Material “Hybrid Unmixing - Linear Unmixing” and further demonstrated in the pseudo-code in “HyU Algorithm”. The spectra are not calculated from phasor, rather aggregated from the original spectral cube dataset. Our strategy is less phasor-esque than the traditional geometry based phasor approach (PMID: 32235070 and 22714302) where the approach uses the phasor geometry to unmix the components. The geometrical phasor approach only uses the coordinates G, S at a specific harmonic, omitting the wavelength dimension in the final unmixing process. As such, like the reviewer correctly states, it is a limiting strategy that makes it impossible to specify the difference between image phasor points in the middle of the phasor cloud with multiple species, requiring multiple harmonics.

With HyU, we utilized the phasor as an encoder, to aggregate similar spectra, because of our familiarity with the approach, but, in principle, other encoding strategies could be utilized. The relative positions and geometry of phasor bin coordinates, from the unmixing algorithm perspective, do not directly matter, as the unmixing is performed with a 32 channel endmember over a 32 channel experimental spectrum (“Linear Unmixing” in Supp. Material).

We improved the text in the last paragraph of the Introduction to clarify this aspect:

Line 13 “HyU utilizes phasor processing as an encoder to aggregate similar spectra and applies unmixing algorithms, such as LU, on them to provide unsupervised analysis of the HFI data, simplifying the data processing and removing user subjectivity.”

Information loss and noise. We are aware that the phasor transform is a lossy encoder that in principle carries a reduced percentage of the information compared to the original “pure” data. This is evident in the scenario of very high quality signals, but in the case of fluorescent signals, where signal to noise often decreases to lower digits, the encoding loss is less relevant compared to the noise of the fluorescent signals. This fundamental advantage of increasing SNR in noisy data makes phasor a valuable tool for fluorescence microscopy (FLIM/spectral alike); this point is reported by multiple groups using phasors (Gratton:

<https://www.pnas.org/doi/full/10.1073/pnas.1108161108>,

<https://escholarship.org/content/qt5g279175/qt5g279175.pdf>, Vicidomini:

<https://www.nature.com/articles/ncomms7701>, Gerritsen:

<https://pubmed.ncbi.nlm.nih.gov/22714302/>, Fraser:

<https://pubmed.ncbi.nlm.nih.gov/28068315/>), and more recently nicely described in the work of Scipioni et al (<https://www.nature.com/articles/s41592-021-01108-4>) “However,

microscopy data are affected by a number of other detrimental factors, [...] which results in decreased signal-to-background ratio (SBR). [...] the phasor approach shows increased precision (Fig. 1f,i), decreased bias (Fig. 1e,h) and a three orders of magnitude lower execution time (Fig. 1g,j) with respect to the least mean square (LMS) fitting procedure”.

To support the validity of this hybrid unmixing approach, we have assembled a complex simulation matrix representing the performance of HyU in unmixing 2 to 8 labels as a function of the spatial overlap in the sample. This simulation matrix is built on top of the complex simulation we designed (further described below in our answer to this reviewer’s question 14), which is soon to be published in a separate manuscript. This simulation accounts for a multitude of real-world noises in experimental samples that are regularly imaged (stochasticity of fluorescence spectral emission, poisson, readout noise, electronics transfer noise, detector sensitivity at different wavelength). The results of applying our approach on an array of simulations under different conditions of SNR, number of filters applied, in comparison to standard Linear Unmixing are now reported in Supplementary Figures 16, 17, 18, and 19. We further describe how multiple components are affected by our hybrid unmixing approach in our answer to question 17 for this reviewer.

Other comments and concerns follow below:

1. The authors provide a spectra.txt files for the unmixing. How was that obtained. How to calculate and save the data from samples having only one fluorophores?

ANSWER: The “Independent Spectral Signatures” subsection of the Image analysis section of the supplementary text now provides an expanded description on this topic. Briefly, we obtain spectra from samples or pure solutions and validate with spectra reported in literature. The edited section “Independent Spectral Signatures” now states:

Independent spectral fingerprints can be obtained through samples, solutions, literatures, or spectral viewer websites (Thermo fisher, BD spectral viewer, Spectra analyzer). Fluorescent signals used in this paper were obtained by imaging single labeled samples in areas morphologically and physiologically known to express the specific fluorescence, see Supplementary Figure 21. For each dataset a phasor plot was computed. The 32-channel spectral fingerprint was extracted from the phasor-bin at the counts-weighted average position of the phasor cluster. Those fingerprints were compared with literature fingerprints and manually corrected to reduce noise. Further descriptions for how to identify new components can be found in Supplementary Note 1 and Supplementary Figure 11, 27.

To further clarify, we have added a Supplementary Figure 27 that shows a step-by-step example on how to obtain single spectral endmember from samples using the attached software:

Supplementary Figure 27. Endmember Spectrum selection process

(A) Phasor map shows the spectral distribution of the data for a single fluorescently labeled sample, in this case an 18 hpf transgenic *Tg(ubiq:lyn-tdTomato)* zebrafish. **(B)** The average spectrum corresponding to a phasor bin selection (red point in A) can be visualized using our software, which plots **(C)** the corresponding average spectrum with relative (top) and absolute (bottom) intensity. The save button allows exporting of spectral data as a txt file that can be re-loaded for unmixing other data. **(D)** Unmixing result. More step-by-step information is available in the README file associated with the software in this publication.

We then expanded Supplementary Note 1 to further clarify:

In our experience, obtaining fluorescence spectra from experimental samples has some advantages compared to utilizing spectra from an existing library, as they account for a multitude of experimental and instrumental settings. Imaging settings such as different types of lenses or optical filters (Sup. Figure 4, C and D) together with factors within the microenvironment of samples, such as pH or temperature have the potential to alter the fluorescence spectral emissions¹⁵. In the presence of unexpected fluorescent signals, spectra can also be selected and visualized directly from the phasor. Phasors facilitate the identification of unexpected independent components and their distinction from the multiple system noises. A noise-free spectrum will appear as a single point on the phasor plot, while a spectrum affected by instrument and electronic noises will mainly appear as a gaussian distribution, centered on the original spectral signal⁸. Conversely, a randomized noise across the multiple spectral channels will not produce a clustered aggregate of spectra on the phasor. A constant spectral noise, with a distinct spectrum (e.g. a constant light leakage into the system), would produce a distinct phasor cluster and could be selected for unmixing. The phasor plot representation is a 2D-histogram and provides insights into the frequency of

occurrence for these signals. These unexpected independent components in samples often appear as “tails” on the phasor distributions (Sup. Figure 11, C). In our HyU graphical interface, clicking on the phasor visualizes the spectra within a small area (9x9 bins by default, with size adjustable from the interface) of the phasor histogram (Figure 1 D).

Pre-identified phasor locations can be displayed in the software; we have also recreated their positions here in Supplementary Figure 21:

Supplementary Figure 21. Pre-identified positions for common fluorophores on the phasor map

(A) Pre-identified extrinsic label positions (g,s) are denoted on the phasor plot for the first harmonic and (B) second harmonic. (C) Intrinsic label locations are further added on the phasor plot for the first harmonic and (D) second harmonic. Second harmonic generally covers a larger portion of the phasor space compared to the first harmonic. However, in the case of intrinsic signals, the locations of the pure autofluorescence spectra are on average more separated when utilizing the first harmonic. Details on the source of the pure spectra for these locations are reported in Methods – Independent Spectral Signatures.

We have compiled a database of 32-channel spectra which have been gathered from literature, retrieved from commercially available resources (spectral viewer for fluorophores such as Alexas), and measured from both pure solutions and fusion proteins with our confocal

microscopes. The spectra.txt files are organized as space delimited files with columns denoting independent spectral signature and rows denoting the channel. A spectra.txt file containing a single column (and so, the spectra for a single fluorophore) can be utilized as shown in the demo for single fluorophore samples (Supplementary Figure 27); this would maximize the contribution of that single spectrum. Exploration of the phasor plot can provide visual indication of unexpected contributing spectral endmembers. Such spectra can be included as a component, as we denoted in Supplementary Note 1. Further instructions on how to build/format and utilize the spectra.txt file have been added to the README.docx.

Step 2 of README.docx:

* *spectra.txt* is a text file with the input spectra needed for unmixing with HyU. The file describes a 2D numpy array outputted using numpy's savetxt function. The file is formatted as a space delimited text file with the rows denoting the channels and the columns denoting the independent spectral signatures. Each row must have the same number of entries, and the values in each entry should be formatted as either integers, floats, or floats in scientific notation ($\#e\pm\#$). The values for the channels of each spectra represent the values corresponding to each spectral bin of uniform size within the detection wavelength range. The spectra.txt file linked in this README serves as an example for this formatting and contains 5 spectra of 32 channels, respectively mko2, tdtomato, mruby, citrine, and an unknown spectral signature chosen for *HyU_demo-02-fishtail.lsm*.

2. I did not see any mention of collagen fluorescence when excited at 740 nm. That should be a component in the autofluorescence category.

ANSWER: We thank the reviewer for the insight. From the analysis of our experimental data, we do not detect measurable collagen signals with our imaging conditions, in the context of the many other fluorescent signals in the sample. Multiple references in literature (PMC4337962 / PMID: 22402635, PMC123202 / PMID: 12177437, PMC4337962 / PMID: 22402635) report 2-photon fluorescence of collagen to be very low at 740 nm and suggest instead the use of Second Harmonic Generation (SHG). At 2-photon 740 nm, the SHG detection would be at around 375 nm, outside of our detection range.

3. What happens to FAD? That can be excited at 740 nm (PMID: 11964266).

ANSWER: The reviewer is correct that FAD can be excited at 740 nm. However, for our imaging conditions, we used low 2P excitation (3% power, 9 mW) at relatively low magnification (pixel size $0.259 \times 0.259 \sim 1.38 \times 1.38 \mu\text{m}$), and therefore, FAD was not sufficiently contributing to the autofluorescence in the datasets shown in the manuscript. We believe the reason to be related to three main factors:

First: According to Huang et al. (Huang et al., 2002 / PMC1302068 / PMID: 11964266), the lower concentration of FAD compared to NADH is known to create a “shadowing” effect of FAD in autofluorescent imaging experiments “because of the predominantly 2P-NAD(P)H fluorescence excited at 750 nm due to its much higher cellular concentration (Guezennec et al., 1991; Kunz and Gellerich, 1993)” leading to the use of 890-900nm excitation for exciting FAD to eliminate the contribution of NADH (Stringari et al, 2011; Skala et al, 2007)

Second: the magnification level in most of our data is relatively low, with pixel size 0.259~1.38 μm , contributing to an apparent reduction of the relative concentration of FAD with respect to NADH.

Third: the 2P excitation used here is intentionally maintained at a considerably low power (3- 4% which corresponds to 9-12mW) to minimize the inevitable photobleaching of the autofluorescent molecules and photodamage of the sample. This, combined with the lower magnification level, produces a low wattage/cm² that further reduces the amount of signal emitted by the low-concentrated FAD to levels we cannot detect. Our estimate of power density for the sample in Figure 5 and Figure 6 are approximately 4.7·10⁻⁶ mW/mm² compared to the zoomed-in sample in Supplementary Figure 22 that was 1.4·10⁻³ mW/mm².

To further confirm, we performed imaging of a wildtype zebrafish hindbrain, at least 11 times higher magnification (pixel size = 0.078x0.078 μm) and a slightly lower laser power at 2.8%; at which point, we were able to observe FAD. Under these imaging conditions, the amount of 2P power deployed on the sample was 10-fold higher than the images shown in the manuscript. This is in agreement with other published works (<https://www.pnas.org/content/104/49/19494.long>). We summarized the results in Supplementary Figure 22:

Supplementary Figure 22. FAD autofluorescence in high magnification brain region of zebrafish embryo

(A) Phasor analysis reveals a distinct autofluorescence spectral component (magenta dot) when utilizing 740 nm 2-photon excitation to image a 22 hpf wild type zebrafish brain with high magnification (pixel size = 0.078x0.078 μ m) and high power (Table S1). **(B)** The corresponding emission spectra from the phasor selections in **A**. The spectrum corresponding to the magenta phasor selection in **A** closely matches the spectral signal of FAD obtained from in vitro solutions (Methods – Independent Spectral Signatures) and accounting for local environment changes¹⁴. **(C)** FAD unmixing channel highlights the FAD cluster in the head region of zebrafish. **(D)** Composite image rendering of the unmixing results for the intrinsic signals: NADH bound, NADH free, Retinoid, Retinoic Acid, FAD, and Elastin.

We expanded Supplementary Note 2 to further describe these details:

“A map of the phasor position for common autofluorescence from pure solutions is reported in Sup. Figure 21 B. Imaging autofluorescent data, with regard to cell metabolism, requires accounting for complex and dynamic changes of metabolic pathways which can occur in a broad range of times, from seconds to years. These autofluorescent signals are often weak in nature and do not rapidly replenish after photobleaching. In our work we utilize reduced laser power to avoid rapid autofluorescence spectral signal bleaching, as well to reduce photo-damage. Additional factors known to affect emission spectra include pH and temperature, pixel-wise concentration of the fluorophore, excitation power, developmental stage and region of the sample imaged. An example for the latter is reported in Sup. Figure 12, where signals in the sample present strong localized differences. One example of the effects of different 2-photon excitation power and different levels of pixel-wise concentration is reported in Figure 5, Figure 6. In these images, samples at similar developmental stages are imaged utilizing different pixel size (0.259 μ m and 0.923 μ m lateral resolutions) and laser power (4% and 3% @740nm 2-photon) resulting in laser power densities of $\sim 4.7 \cdot 10^{-6}$ mW/mm². This different laser power causes some lower-concentration intrinsic fluorophores to not be excited, in this case mRuby is visible in Figure 5 but not in Figure 6. In both of these images, FAD is not excited in measurable quantity, whereas in Sup. Figure 22, where the laser power density is $1.4 \cdot 10^{-3}$ mW/mm², FAD contribution is measurable and unmixed. HyU is well posed for the analysis of intrinsically low autofluorescence owing to its ability to operate at low SNR. In Sup. Figure 12, we visualize unmixing of multiple autofluorescent signals based on spectra acquired from in vitro solutions. Sup. Figures 17, 19 present a simulated overview of the improvement of HyU over Linear Unmixing for autofluorescence data, as a function of number of labels, percentage of pixels containing mixed ratio of fluorophores, number of denoising filters applied and number of channels under different levels of Signal to Noise.”

4. About Bound NADH – does the spectra change based on the proteins it binds to?

ANSWER: We appreciate the explorative nature of this question with regard to the behavior of NADH. While we do not believe that this is in line with the purpose of this manuscript, a literature search shows interesting information about this subject.

In figure 5 of Maltas et al (<https://doi.org/10.1021/ac504386x>), the authors report spectra and spectral phasors of NADH bound to LDH and MDH. Maltas et al conclude that “protein-bound NADH conformations are spectrally distinguishable” in solution when measured with 1024 channels at 2nm resolution. From their Figure 5, it appears the spectra have differences, although these differences are not resolvable with the 32 channel PMT utilized in commercial confocal microscopes. For simplicity, we are copying the figure below with its original caption (from Maltas et al (<https://doi.org/10.1021/ac504386x>)

Figure 5 from Maltas et al (<https://doi.org/10.1021/ac504386x>): Figure 5. Spectral detection of NADH protein binding. Sequential additions of MDH, MDH, and LDH to solutions of NADH (left column) or NADPH (right column). In all plots, the color or symbol shape corresponds to times before protein is added (red, square), after adding MDH once (blue, circle), after adding MDH twice (orange, triangle), and after adding LDH (green, inverted triangle). (a) Integrated-gate spectra. The spectrum taken before protein addition was scaled to its maximum intensity, with all other spectra scaled to minimize least-squares differences. Fractional-difference spectra are calculated using the spectrum before protein addition as the reference spectrum. (b) Spectral phasor plots. Shifts due to MDH addition to the NADH solution (left) are collinear (best fit line is shown). The shift due to LDH addition is not collinear with MDH-induced shifts, evidencing that protein-bound NADH conformations are spectrally distinguishable. As a negative control, no shift is observed when proteins are added to the NADPH solution (right). (c) Spectrum shape

plots for the same data as in (b). For (b) and (c), analysis is performed over the first 512 pixels (400–530 nm wavelength range).

5. What is the distance of separation of the pure components in the phasor space that allow for successful linear unmixing. I presume at some point the S/N will make it difficult if the spectra of the components are too close.

ANSWER: In general, a good rule would be for pure spectral endmembers to be separated by at least one phasor scatter error, between center of scatter (endmember 1) to center of scatter (endmember 2), based on the definition introduced in the previous work (PMID: 28068315).

For our experimental conditions, the phasor scatter error is ~ 0.09 ; hence, that would be a threshold distance between pure endmembers for a reliable linear unmixing. However, it should be noted that multiple factors contribute to this “minimal distance”.

1. Number of histogram bins utilized: in this work, we utilize the phasor as an encoder/aggregator for the spectra in the raw dataset. We do not use the geometry of phasor for unmixing. The phasor transformed data is discretized as a 2D histogram which is then used to sort and average the raw spectra in wavelength space. Both the number of bins used to discretize the phasor plot and the actual (G,S) locations of the pure components on the phasor plot will affect the quality of the unmixing result.
2. Instrument noise: instruments are characterized by noise that differs in quantity across wavelengths. The parameter we estimate above can be used as reference for a Quasar Detector (Zeiss 710-780 series). A better sensor (e.g. the cooled version of Quasar in Zeiss 880-above series) will have lower noise than our current instrument. Any later generation emCCD or BSI-sCMOS camera based spectral detector would have considerably lower noise.

6. What about when the linear unmixing won't work? For example, a case where the individual components lie in line in the phasor space.

ANSWER: We believe that the reviewer is referring to the case where multiple independent components lie exactly on a line within the phasor plot such that multiple combinations of different fluorophores will occupy the exact same location on the phasor plot.

Example of this scenario: a sample with 3 endmembers, A, B, C, where $A = B/2 + C/2$.

Mathematically, it is true that in this extreme case, the unmixing will have unpredictable results. The likely reason behind this unpredictability is that this scenario breaks one of the general constraints of Linear Mixing Models where endmember spectral signatures need to be linearly independent, making standard linear unmixing or fully constrained linear unmixing an ill posed problem. In practice, however, in such a case, one might argue if A is really an endmember or simply a combination of two spectra present in the sample. In our experience, we have found that it is highly unlikely for this condition to occur in spectral fluorescence of biomedical samples. As demonstrated in Supplementary Figure 16, the chances of the phasor locations of different fluorophores to line up exactly on the same line is very small.

Though it may appear very close to a single line, there is sufficient non-linearity to create an offset that this case does not happen. It is much more likely for different combinations of many fluorophores (>4) to somehow occupy the same phasor location. However, we have demonstrated in our reply to the first comment and in the numerous additional simulations (Supplemental Figure 16, 17, 18, 19) that noise affects the unmixing results much more than the encoding loss of the phasor method.

7. Coming back to Q1. – Prior knowledge of the components – How are they calculated? Where are the coordinates stored? Are only the center of the phasor cloud used or the whole distribution?

ANSWER: We believe our answer to Q1 addresses the majority of this question. We have expanded details on this subject in the Image analysis “Independent Spectral Signatures” subsection of the Image analysis section of the supplementary text and our expanded Supplementary Note 1 as well as added Supplementary Figures 21, 27.

We utilized the 32-channel spectrum contained in the phasor-bin at the weighted average position of the phasor cluster following the method described in the updated “Independent Spectral Signatures” subsection:

Independent spectral fingerprints can be obtained through samples, solutions, literatures, or spectral viewer websites (Thermo fisher, BD spectral viewer, Spectra analyzer). Fluorescent signals used in this paper were obtained by imaging single labelled samples in areas morphologically and physiologically known to express the specific fluorescence, see Supplementary Figure 21. For each dataset a phasor plot was computed. The 32-channel spectral fingerprint was extracted from the phasor-bin at the counts-weighted average position of the phasor cluster. Those fingerprints were compared with literature fingerprints and manually corrected to reduce noise. Further descriptions for how to identify new components can be found in Supplementary Note 1 and Supplementary Figure 11, 27.

The phasor coordinates, obtained from the prior experiments, are saved/embedded in the software. New spectral signatures can be imported using a correctly formatted text file (see answer to Question 1 and README.docx). We added Supplementary Figure 21 to demonstrate phasor positions of common fluorophores (corresponding emission spectra can be found in Supplementary Figure 20).

8. I do like the point mentioned in lines 10-20 in page 2. Spectral imaging and deconvolution is absolutely necessary.

ANSWER: We thank the reviewer for this comment.

9. Lines 30-40, page 2 – missing references about the different noise.

ANSWER: We have included the following references to the main text and bibliography:

22. Bass, M. *Handbook of Optics, vol 3. Geometric Optics, General Principles Spherical Surfaces, 2nd ed., Optical Society of America, New York (1995).*
23. Hamamatsu Photonics, K. K. P. T. H. *PHOTOMULTIPLIER TUBES Basics and Applications FOURTH EDITION. (1994).*
24. Pawley, J. B. Confocal and two-photon microscopy: Foundations, applications and advances. *Microscopy Research and Technique* **59**, (2002).
25. Huang, F. *et al.* Video-rate nanoscopy using sCMOS camera-specific single-molecule localization algorithms. *Nature Methods* **10**, (2013).

10. The linear deconvolution of the phasor space involves fractional intensity and not the actual fraction – something that I found missing in the discussion.

ANSWER: The reviewer is correct, however we are not utilizing or calculating fractional intensity on phasor. We utilize linear unmixing on the aggregated corresponding spectra in wavelength space. We believe the initial answer to this reviewer addresses this question in completeness.

11. How do 5 photons/spectra work with Poisson statistics and associated uncertainty?

ANSWER: Utilizing 5 photons per spectra results in very noisy data. As we demonstrate in the figures below, which represent both simulated single spectrum and simulated combined multiple spectra data, there is a considerable amount of obfuscation between the ideal spectral shape and the combined signal of the stochastically emitted spectra with the background noise. Averaging of the spectra using the phasor does provide a more spectrally similar array of values, but still results in a very noisy spectra, with deteriorated unmixing results.

Simulation of tdTomato spectral fluorescent emission with max 5 photons.

Simulations are performed starting from a reference spectrum for tdTomato (black dash line), acquired from a pure solution and matched with values reported in literature. **(A)** Spectra are shown for 3 randomly selected pixels (orange, blue, green lines) within the phasor histogram bin with maximum number of counts. **(B)** Average of the spectra (blue line) belonging to all pixels inside the phasor histogram bin with maximum number of pixel counts. In both plots, the reference spectrum scaled to the max of the simulated spectra is delineated with a black dashed line.

Simulation of multiple spectral fluorescent emission with max 5 photons per spectra. Simulations are performed by combining four reference spectra (mKO2, Citrine, mRuby, tdTomato) at 5 photons per spectra. **(A)** Spectra are shown for 3 randomly selected pixels (orange, blue, green lines) within the phasor histogram bin with the maximum number of counts. **(B)** Average of the spectra belonging to all pixels inside the phasor histogram bin with

maximum number of pixel counts. In both plots, the dashed lines represent the reference spectrum, scaled to the maximum value of the simulated spectra.

12. Does the Elastin spectrum change on crosslinking in a tissue compared to the solution?

ANSWER: This is an interesting question that we have not yet addressed. While we believe this to be outside the focus of this paper, we found answers in literature. Richards-Kortum et al (DOI: 10.1146/annurev.physchem.47.1.555) state that collagen and elastin fluorescence is associated with cross-links and report differences in fluorescence emission maxima between elastin in powdered form, in skin and in load bearing structures ([https://doi.org/10.1016/0584-8539\(89\)80031-5](https://doi.org/10.1016/0584-8539(89)80031-5)). These differences were echoed by Eyre et al (10.1146/annurev.bi.53.070184.003441) and more recently by Croce et al (10.4081/ejh.2014.2461).

13. Td-Tomato (PMID 19127988) and mRuby (PMID 23459413) can be excited with a 740 nm two-photon excitation. I am curious how the authors did not observe that in Figure 6.

ANSWER: The reviewer is correct, both tdTomato and mRuby can be excited with 740 nm. Our main purpose in Figure 6 was to demonstrate the different extrinsic and intrinsic signal profiles when excited by either 1-photon or 2-photon. Since our focus for this sample is in the autofluorescence for two-photon 740 nm excitation, we purposely chose samples with reduced extrinsic fluorescence expression to prevent a suppressing effect that we experimentally found to interfere with measurement of autofluorescence signals. Although the fluorescence signals are still visible, we decided to not include the tdTomato and mRuby channels from the 740nm section of the figure owing to the large amount of signals to display. We have now added Supplementary Figure 15 with both signals from 740 nm two-photon excitation.

Note that Figure 5 and Figure 6 data come from the same sample. tdTomato expression is observed in figure 5 because of the imaging setting. The imaging area in Figure 5 is 2.59x2.59 μm versus 9.23x9.23 μm in Figure 6. The image area in Figure 6 is 12 times larger for each pixel while using 1% lower laser power. Laser power was intentionally reduced to avoid photo-bleaching and photo-damage as well as to minimize disturbance of the system during longer time series imaging, with the expectation of reduced excitation of extrinsic fluorescent signals. We added this data in Supplementary Figure 15 and observed low intensity results showing a low expression level.

Supplementary Figure 15:

Supplementary Figure 15. HyU unmixing on low concentration signals using customized independent spectra

Results from unmixing intrinsic and extrinsic signals of a quadra-transgenic zebrafish: *Gt(citca-citrine);Tg(ubiq:lyn-tdTomato; ubiq:Lifeact-mRuby;fli1:mKO2)* at a single timepoint of the dataset presented in Figure 6 provide further information and highlight the weak expression of some extrinsic signals in this dataset. **(A)** Input spectra for the intrinsic signatures were directly acquired by selection of the endmembers in the phasor plot. Input spectra for the extrinsic signatures were acquired from other datasets of samples expressing those signatures individually and excited at 740 nm with 2-photon, since these extrinsic signals are not strongly expressed within this dataset. **(B)** Renderings of unmixing results were automatically adjusted to show the best contrast. Unmixing can still be performed with spectra from weak input signatures. **(C)** Histogram counts of each unmixed independent spectral signature demonstrate the low signals of the extrinsic fluorescence signatures compared to the intrinsic ones. The median values of the mRuby and tdTomato channels are 57 and 77 Digital Levels respectively, considerably lower than those of the other signals.

14. What determines how much spatial denoising needs to be used? Spatial denoising indeed doesn't affect the intensity image – but it does affect the phasor mapped image – something that hasn't been discussed at all.

ANSWER: This question has been answered in our previous paper (Cutrale et.al., Nature Methods). It was determined that the spatial denoising plateaus after 5 applications. Still, we have added multiple simulations to better quantify the performance improvement of HyU with respect to LU for multiple denoising filters (Supplementary Figure 16 and 17) in terms of Relative Mean Squared Error (RMSE) (as defined in the Methods section, Mean Square Error). The new figures show a matrix of RMSE values for a number of fluorophores over the percentage of overlap. Each RMSE value is calculated as average from a simulation that has n fluorophores and a specific percentage of pixels containing randomized ratios of n fluorophores, where n is the number of fluorophores in the specific matrix position. For example, the RMSE value for 6 fluorophores at 50% overlap is calculated from a simulated dataset with 6 fluorophores where 50% of the pixels contain a randomized combination of the 6 fluorophores, while the remaining pixels contain a single fluorophore. Further details on these newer simulations have been added to the Methods section Hyperspectral Fluorescence Image Simulation.

Hyperspectral Fluorescence Image Simulation

The model simulates spectral fluorescent emission by generating a stochastic distribution of photons with profile equivalent to the pure reference spectra (as described in Sup. Note 1). The effect of photon starvation, commonly observed on microscopes, is synthetically obtained by manually reducing the number of photons in this stochastic distribution.

Detection, Poisson and signal transfer noises are then added to produce 32-channel fluorescence emission spectra that closely resemble those acquired on microscopes. The simulations include accurate integration of dichroic mirrors and imaging settings.

Simulation Types:

Biologically comparable simulations

To quantify the performance of HyU vs LU for microscopy data acquired experimentally, we generated synthetic data where each input spectra was organized with intensity distributions taken from experimental data of fluorescently labeled biological samples.

We calibrated the analog (Digital Levels) to photon counting rate based on existing literature^{5,6}. Experimental data was discretized to photons to produce biologically relevant photon masks with distributions of signals highly resembling those of the samples. This provided intensities and ratios which closely resemble those acquired from a confocal microscope while allowing control over the effects of photon starvation.

Spatially and spectrally overlapping simulations

We also included simulations to quantify the performance of HyU vs LU with respect to the number of spectral combinations and of endmembers. The results are summarized in Supplementary Figures 16-19 in the form of matrices of spectral overlap (0 to 100%, steps of 10%, X-axis) by number of endmembers (2 to 8 endmembers, Y-axis) representing the Relative Mean Squared Error (RMSE) (Supplementary Methods, Performance quantification). Each RMSE value reported in a matrix is the average of analysis of a 1024x1024 pixels image simulation with a spectral dimension of 32- channels matching the spectral range and bandwidth of the detectors in commercial confocal microscopes (LSM 780, Carl Zeiss). These simulations were created with artificial intensity distributions so that a simulation with X% overlap and n fluorophores would have X percentage of pixels with a randomized ratio of n input spectra. As an example, for a simulation with 6 fluorophores and 50% overlap, the simulated dataset would have 50% of the pixels contain a randomized combination of the 6 fluorophores, while the remaining pixels contain a single fluorophore. This allowed us to investigate the effects of an increasing number of spectral combinations on the compressive nature of the phasor method for HyU.

Supplementary Figure 16 is simulated utilizing only extrinsic fluorophores and Supplementary Figure 17 is simulated using a combination of intrinsic and extrinsic fluorophores:

Supplementary Figure 16. RMSE improvement for simulated fluorescent spectral combinations highlights increased HyU performance across multiple denoising filters.

Twelve matrices demonstrate the RMSE improvement of HyU with respect to LU when unmixing a collection of synthetic data with 2 to 8 extrinsic labels (Y axis of each matrix) as a function of the spatial overlap of these labels in a sample (X axis of each matrix). In the matrix, 0% overlap denotes simulations with spatially distinct fluorophores, where each pixel corresponds to a single fluorophore, while simulations with 100% overlap contain, in every pixel, a randomized ratio of the n fluorophores. Each one of the values reported in a matrix is the average of a 1024x1024x32 pixels simulation and shows the RMSE improvement of HyU to LU. Different columns in the figure report the RMSE improvement matrices with different numbers of denoising filters (0x, 1x, 3x, 5x) applied with a total number of photons per pixel at (A) 16 (B) 32 (C) 48. In the absence of denoising filters, the improvement of HyU overall is less than 8%. Denoising filters improve RMSE by over 80%. Spectra utilized for this simulation are reported in Sup.

Figure 20A.

Supplementary Figure 17. RMSE improvement for simulated fluorescent and autofluorescent spectral combinations highlights increased HyU performance across multiple denoising filters

Twelve matrices demonstrate the RMSE improvement of HyU with respect to LU when unmixing a collection of synthetic data with 2 to 8 extrinsic and intrinsic labels (Y axis of each matrix) as a function of the spatial overlap of these labels in a sample (X axis of each matrix). In the matrix, 0% overlap denotes simulations with spatially distinct fluorophores, where each pixel

corresponds to a single fluorophore, while simulations with 100% overlap have, in every pixel, a randomized ratio of the n extrinsic and intrinsic fluorophores. Each one of the values reported in a matrix is the average of a 1024x1024x32 pixels simulation and shows the RMSE improvement of HyU to LU. Different columns in the figure report the RMSE improvement matrices with different numbers of denoising filters (0x, 1x, 3x, 5x) applied with a total number of photons per pixel at (A) 16 (B) 32 (C) 48. In the absence of denoising filters, the improvement of HyU overall is less than 25%. Denoising filters improve RMSE by over 100%. Spectra utilized for this simulation are reported in Sup. Figure 20B.

15. Figure 1 D-E – this is strictly not true. Once you transfer to phasor – the information remaining for the spectra is the FWHM and the peak/center – so how does the proper spectra being calculated in figure E?

ANSWER: The reviewer is correct that full spectra cannot be calculated from the first harmonic components without major loss of information. We are not calculating the spectra from the phasor; instead, we are preserving them in wavelength space from the original input data (32 channel) that was used to calculate the phasor components. The unmixing is performed over the spectral dimension (in this case, 32 channels) on aggregated similar spectra from the original data (in 32 channels), as explained in Supplementary Material “Hybrid Unmixing - Linear Unmixing” and further demonstrated in the pseudo-code in “HyU Algorithm”.

We improved Figure 1 D-E, changing the connecting arrow to be “Phasor Aggregation”.

16. The reduction of data from 10^7 to 10^4 . How much of that is related to spectral denoising and how much is related to the transformation to phasor?

ANSWER: The majority of the data reduction occurs when transforming the raw data to the phasor. The initial transformation reduces the data by approximately two orders of magnitude, from 10^7 to 10^5 . Successive spectral denoising filters further reduce the data in much smaller increments, staying within 10^4 elements.

17. Page 4 line 19 – after two-components – what happens with three/four and their possible combinations?

ANSWER: Line 19 is meant to provide an overview of the standard phasor approach, with a simple example of one of the advantages of phasors. HyU does not utilize the geometry of phasor for performing the unmixing, circumventing in most cases the challenges related to phasor ratiometric unmixing of more than 3 components.

The unmixing is performed over the spectral dimension (in this case, 32 channels), by aggregating similar spectra of the original data, as explained in Supplementary Material “Hybrid Unmixing - Linear Unmixing” and further demonstrated in the pseudo-code in “HyU Algorithm”. The spectra are not calculated from phasor; rather, they are aggregated from the original spectral cube dataset. HyU strategy is less phasor-esque than the traditional geometry based phasor approach (PMID: 32235070 and 22714302) where the approach uses the phasor geometry to unmix the components. The phasor geometrical approach only uses the coordinates G, S at a specific harmonic, omitting the wavelength dimension in the final unmixing process. As such, like the reviewer correctly states, it is a limiting strategy that makes it impossible to specify the difference between image phasor points in the middle of the phasor cloud with multiple species, requiring multiple harmonics.

With HyU, we utilized the phasor as an encoder, to aggregate similar spectra, because of our familiarity with the approach, but, in principle, other encoding strategies could be utilized. The relative positions and geometry of phasor bin coordinates, from the unmixing algorithm perspective, do not directly matter, as the unmixing is performed with a 32 channel endmember over a 32 channel experimental spectrum (“Linear Unmixing” in Supp. Material).

The phasor transform is a lossy encoder that in principle carries a reduced percentage of the information compared to the original data. This is evident in the scenario of very strong signals (e.g. in hyperspectral reflectance), but in the case of fluorescent signals, where signal to noise often decreases to lower digits, the encoding loss is less impactful on the results compared to the noise of the fluorescent signals. This fundamental advantage of phasor in increasing SNR in noisy data makes phasor a valuable tool for fluorescence microscopy (FLIM/spectral alike); this point is reported by multiple groups using phasors:

Gratton: <https://www.pnas.org/doi/full/10.1073/pnas.1108161108>,
<https://escholarship.org/content/qt5g279175/qt5g279175.pdf>,

Vicidomini: <https://www.nature.com/articles/ncomms7701>, Gerritsen:

<https://pubmed.ncbi.nlm.nih.gov/22714302/>,

Fraser: <https://pubmed.ncbi.nlm.nih.gov/28068315/>

and more recently nicely described in the work of Scipioni et al (<https://www.nature.com/articles/s41592-021-01108-4>) “*However, microscopy data are affected by a number of other detrimental factors, [...] which results in decreased signal-to-background ratio (SBR). [...] the phasor approach shows increased precision (Fig. 1f,i), decreased bias (Fig. 1e,h) and a three orders of magnitude lower execution time (Fig. 1g,j) with respect to the least mean square (LMS) fitting procedure*”.

We characterize the encoding loss of the Hybrid Unmixing approach owing to phasor and show that HyU still outperforms the standard LU. We performed simulations following the description provided in answer 14 to this reviewer, assembling a matrix of performance improvement with respect to Linear Unmixing by utilizing Relative Mean Squared Error (also described in Supplementary Methods) across the number of fluorophores and percentage of pixels with randomized overlap. This complex simulation matrix represents the performance of HyU in unmixing 2 to 8 labels as a function of the spatial overlap in the sample. This simulation matrix is built from a complex simulation soon to be published in a separate manuscript that accounts for a multitude of real-world noises that occur in imaging of experimental samples (stochasticity of fluorescence spectral emission, poisson, readout noise, electronics transfer noise, detector sensitivity at different wavelength).

In Supplementary Figures 16-19, we explore this matrix of fluorophores/overlap with the following parameters:

- Number of denoising filters
- Different values of SNR
- Fluorescent or Autofluorescent labels
- Number of channels detected

Supplementary Figures 16 and 17 are reported above in question 14 of this reviewer. Supplementary Figures 18 and 19:

Supplementary Figure 18. RMSE improvement for simulated fluorescent spectral combinations highlights decreasing overall performance across decreased number of spectral channels

Fifteen matrices demonstrate the RMSE improvement of HyU with respect to LU when unmixing a collection of synthetic data with 2 to 8 extrinsic labels (Y axis of each matrix) as a function of the spatial overlap of these labels in a sample (X axis of each matrix). In the matrix, 0% overlap denotes simulations with spatially distinct fluorophores, where each pixel corresponds to a single fluorophore, while simulations with 100% overlap contain, in every pixel, a randomized ratio of the n fluorophores. Each one of the values reported in a matrix is the average of a 1024x1024x32 pixels simulation and shows the RMSE improvement of HyU to LU with 3x denoising filters. Columns in the figure represent RMSE improvement matrices across an increasingly binned number of spectral channels (32, 16, 8, 6, 4) applied with a total number of photons per pixel at (A) 16 (B) 32 (C) 48. When utilizing 32 spectral channels data, RMSE improvements reach above the previously reported 80% for highly overlapping fluorophores. Successively increasing the binning across the wavelength dimension (and therefore decreasing the number of spectral channels) shows a slow downward trend of the RMSE improvement until the 4 spectral channels matrices, where the RMSE improvement drops drastically down to below 8%, especially for more than 6 labels. Spectra utilized for this simulation are reported in Sup. Figure 20A.

Supplementary Figure 19. RMSE improvement for simulated fluorescent and autofluorescent spectral combinations highlights decreasing overall performance across decreased number of spectral channels

Fifteen matrices demonstrate the RMSE improvement of HyU with respect to LU when unmixing a collection of synthetic data with 2 to 8 extrinsic and intrinsic labels (Y axis of each matrix) as a function of the spatial overlap of these labels in a sample (X axis of each matrix). In the matrix, 0% overlap denotes simulations with spatially distinct fluorophores, where each pixel corresponds to a single fluorophore, while simulations with 100% overlap contain, in every pixel, a randomized ratio of the n extrinsic and intrinsic fluorophores. Each one of the values reported in a matrix is the average of a 1024x1024x32 pixels simulation and shows the RMSE improvement of HyU to LU. Columns in the figure represent RMSE improvement matrices with 3x denoising filters across an increasingly binned number of spectral channels (32, 16, 8, 6, 4) applied with a total number of photons per pixel at (A) 16 (B) 32 (C) 48. When utilizing 32 spectral channel data, RMSE improvements reach up to the previously reported 100% for highly overlapping fluorophores. Successively increasing the binning across the wavelength dimension (and therefore decreasing the number of channels) shows a slow downward trend of the RMSE improvement until the 4 spectral channel matrices, where the RMSE improvement drops drastically down to below 25%, especially for more than 3 labels. Spectra utilized for this simulation are reported in Sup. Figure 20B.

With regard to the simulations, the Supplementary Methods section “Hyperspectral Fluorescence Image Simulation” has been updated to include descriptions of the new simulations as reported in our reply to question 14 of this reviewer.

With regard to the encoding, the Discussion in the main text was updated to state:

Due to the initial computational overhead for encoding spectra in phasors, there is a 2- fold speed reduction for HyU in comparison to standard LU. However, this may be improved with further optimizations of the HyU implementation or by implementing different types of encoding.

We added a new Supplementary Note 4 that details aspects of phasors and encoding loss. The note reads:

Supplementary Note 4: Improvements of HyU over the standard phasor analysis

Linearity of combinations is the general assumption for most of the spectral analysis algorithms in Hyperspectral Fluorescence Imaging (HFI). Each pixel is assumed to contain a linear combination of the independent spectral signatures, or endmembers, contained in the sample. This assumption requires knowledge, or identification, of the independent spectra within the sample. In standard linear unmixing algorithms, the extraction of relative amounts of spectra (ratios) is conducted on a pixel-by-pixel basis, at the expense of computational costs. Disrupted experimental signals, in the case of lower Signal to Noise Ratio (SNR) spectra, complicate the detection of spectral endmembers and reduce the accuracy of ratio determination. These standard unmixing algorithms, however, have the advantage of being unsupervised with the possibility of automating the analysis process.

The phasor approach has become a popular dimensionality reduction approach for the analysis of both fluorescence lifetime and spectral image analysis¹³⁻¹⁵. Phasors provide key advantages, including spectral compression, denoising, and computational reduction for both pre-processing³ and unmixing^{8,16,17} of HFI datasets. Phasor analysis overcomes the challenge of low SNR data analysis that limits standard unmixing algorithms, providing a multiplexing solution to a need. The phasor transform is a lossy encoder that in principle carries a reduced percentage of the information compared to the original clean data¹⁸. In the imaging of fluorescent signals, where signal to noise often decreases to lower digits, the encoding loss is less relevant compared to the noise of the fluorescent signals. This fundamental advantage of increasing SNR in noisy data has made the phasor method a valuable tool for fluorescence microscopy, both for Lifetime and Spectral Fluorescence Microscopy. This point is reported by multiple groups using phasors¹⁸⁻²¹ and, more recently, nicely described in the work of Scipioni et al²². Standard Phasor analysis²³⁻²⁵ is fully supervised and requires a manual selection of

regions or points on a graphical representation of the transformed spectra, called the phasor plot. Each selection of a region in the phasor plot associates pixels containing similar spectra to the same fluorophore, forming an output channel that contains wavelength integral of intensities with unitary ratiometric value. This “winner takes all” approach is suitable when fluorophores for each single excitation light are spectrally overlapping and spatially disperse (Sup. Figure 24), but requires separate acquisition of different excitation wavelengths for demultiplexing spatially and spectrally overlapping fluorophores (Sup. Figure 25).

HyU uses the phasor transform to group pixels with similar spectral shape within each phasor histogram bin. This approach maintains the advantage of compressing, denoising and simplifying identification of clean endmember fluorescent spectra. However, HyU improves on the robustness of the analysis. The denoised signals are maintained in a hybrid phasor and wavelength domain, and therefore can be unmixed with a multitude of standard unmixing algorithms (Sup. Figure 13), such as Linear Unmixing or Fully Constrained Least Squares. These standard unmixing approaches can operate without supervision and provide for each pixel the ratios for a set of spectral signals, overcoming some of the limitations of phasor, but generally do not perform well in experimental conditions with reduced and compromised signals, such as in fluorescence, and require extensive computational time for high spectral-count datasets. HyU provides wavelength-based denoised spectra that enable these standard algorithms to outperform

their pixel-by-pixel typical application, both in quality of the results (Sup. Figures 16-19), owing to cleaner and better defined fluorescent spectra in each phasor bin, and, generally, in speed, owing to the phasor dimensionality reduction. HyU performs well for single excitation light when fluorophores are spectrally overlapping both when they are spatially disperse or co-localized, providing a ratio for each independent spectrum currently unmixed. Our data suggests HyU has reasonable performance for up to 8 different fluorophores per dataset, for each single excitation wavelength. In an experiment with a carefully chosen palette of labels, where octuples of fluorophores can be excited by a single wavelength, with an instrument capable of spectral acquisition with 5 standard and sufficiently spectrally separated excitation wavelengths in 5 sequential acquisitions (one for each excitation light), HyU could, in principle, unmix 40 signals. This performance however decreases with the number of channels (Sup. Figures 18, 19) showing a small deterioration at 8 channels and limitations at 4.

18. One of the uses of HyU is for low light level and long term imaging. What happens to the deconvolution if there is bleaching? This is a minor concern.

ANSWER: The effects of phasor denoising with respect to photobleaching have been characterized in our previous work (<https://www.nature.com/articles/s41467-020-14486-8>) in:

- Supplementary Figure 26 (https://static-content.springer.com/esm/art%3A10.1038%2Fs41467-020-14486-8/MediaObjects/41467_2020_14486_MOESM1_ESM.pdf) and
- Supplementary Movie 3 (https://static-content.springer.com/esm/art%3A10.1038%2Fs41467-020-14486-8/MediaObjects/41467_2020_14486_MOESM6_ESM.mov)

For simplicity, we copy paste Supplementary Figure 26 from Shi et al 2020:

from Shi et al 2020:

“Supplementary Figure 26. Visualization of photobleaching with SEER. Photo-bleaching experiments were performed on a 24 hpf zebrafish embryo Gt(cltca-citrine); Tg(fli1:mKO2); Tg(ubiq:memTdTomato), labeling clathrin, pan-endothelial and membrane respectively. The experiments were performed utilizing the “bleaching” modality in the Zeiss Zen 780 inverted confocal, where single z positions were acquired in lambda mode. Frames are acquired every 13.7 sec, with 5 intermediate bleaching frames (not acquired) at high laser power until image intensity reached 90% bleaching. The SEER RGB mask represents the values of colors associated to each pixel, independent from the intensity values. The map used here is Radial map in Center of Mass mode. In this modality the map will adjust its position on the shifting center of mass of the phasor clusters, visually compensating for the decrease in intensity. (a) In the initial frame the cltca-citrine is associated to a magenta color, membrane to cerulean, pan-endothelial is not in frame and background to yellow. (b) Frame 10 shows consistent colors with the initial bleaching; the colors are maintained (c) at frame 40 and (d) frame 70 where most of the signal has bleached and most colors have switched to yellow (here, background). (e) Final frame shows the 90% bleached sample. The Alpha Color rendering adds the information of intensity to the image visualization. Here we show for comparison (f) frame 1, (g) frame 10, (h) frame 40 and (i) frame 70. Scale bar 10um. (j) Average total intensity plot as a function of frame, calculated from the sum of 32 channels, shows evident bleaching in the sample. Further visualization is provided in Supplementary Movie 3

19. How to create the spectral libraries in the software provided by the authors (page 15, line 29-30)?

ANSWER: We refer the reviewer to the answer to Q1, where we explain how to obtain the spectra from samples or literature and how to format them to fit the spectra.txt library in our example.

Correspondingly, we updated the manuscript in the section “Independent Spectral Signatures” under Image analysis in Supplementary Information, added Supplementary Figures 21 and 27, and edited the README document.

20. I do feel the references can be expanded for the phasor analysis of the multicomponent systems from other labs.

ANSWER: We expanded the reference list with respect to phasor analysis, preprocessing hyperspectral data, and unmixing hyperspectral data by including the following:

26. Digman, M. A., Caiolfa, V. R., Zamai, M. & Gratton, E. The Phasor Approach to Fluorescence Lifetime Imaging Analysis. *Biophysical Journal* **94**, L14–L16 (2008).
27. Fereidouni, F., Bader, A. N., Colonna, A. & Gerritsen, H. C. Phasor analysis of multiphoton spectral images distinguishes autofluorescence components of in vivo human skin. *Journal of Biophotonics* **7**, 589–596 (2014).
28. Scipioni, L., Rossetta, A., Tedeschi, G. & Gratton, E. Phasor S-FLIM: a new paradigm for fast and robust spectral fluorescence lifetime imaging. *Nature Methods* **18**, 542–550 (2021).
29. Ranjit, S., Malacrida, L., Jameson, D. M. & Gratton, E. Fit-free analysis of fluorescence lifetime imaging data using the phasor approach. *Nature Protocols* **13**, 1979–2004 (2018).
30. Shi, W. *et al.* Pre-processing visualization of hyperspectral fluorescent data with Spectrally Encoded Enhanced Representations. *Nature Communications* **11**, 1–15 (2020).
31. Keshava, N. & Mustard, J. F. Spectral unmixing. *IEEE Signal Processing Magazine* **19**, 44–57 (2002).
32. Dobigeon, N., Altmann, Y., Brun, N. & Moussaoui, S. Linear and Nonlinear Unmixing in Hyperspectral Imaging. in *Data Handling in Science and Technology* vol. 30 (2016).
33. Zeiss, C. & Online, M. Introduction to Spectral Imaging and Linear Unmixing. *Imaging* **1**, 1–13 (2010).
34. Hedde, P. N., Cinco, R., Malacrida, L., Kamaid, A. & Gratton, E. Phasor-based hyperspectral snapshot microscopy allows fast imaging of live, three-dimensional tissues for biomedical applications. *Communications Biology* **4**, (2021).
35. Cutrale, F. *et al.* Hyperspectral phasor analysis enables multi-plexed 5D in vivo imaging. *Nature Publishing Group* (2017) doi:10.1038/nmeth.4134.

36. Stringari, C. *et al.* Metabolic trajectory of cellular differentiation in small intestine by Phasor Fluorescence Lifetime Microscopy of NADH. *Scientific Reports* **2**, (2012).
37. Ranjit, S., Datta, R., Dvornikov, A. & Gratton, E. Multicomponent Analysis of Phasor Plot in a Single Pixel to Calculate Changes of Metabolic Trajectory in Biological Systems. *Journal of Physical Chemistry A* **123**, (2019).
38. Jeong, S. *et al.* Time-resolved fluorescence microscopy with phasor analysis for visualizing multicomponent topical drug distribution within human skin. *Scientific Reports* **10**, (2020).
39. Haas, K. T., Fries, M. W., Venkitaraman, A. R. & Esposito, A. Single-Cell Biochemical Multiplexing by Multidimensional Phasor Demixing and Spectral Fluorescence Lifetime Imaging Microscopy. *Frontiers in Physics* **9**, (2021).
40. Lanzaò, L. *et al.* Encoding and decoding spatio-temporal information for super-resolution microscopy. *Nature Communications* **6**, (2015).
41. Yao, Z. *et al.* Multiplexed bioluminescence microscopy via phasor analysis. *Nature Methods* **2022 19:7 19**, 893–898 (2022).
42. Depasquale, J. A. Actin Microridges. *Anat Rec (Hoboken)* **301**, 2037–2050 (2018).
43. Okuda, K. S., Hogan, B. M., Cantelmo, A. R. & Hogan, B. M. Endothelial Cell Dynamics in Vascular Development : Insights From Live-Imaging in Zebrafish. **11**, (2020).
44. Isogai, S., Lawson, N. D., Torrealday, S., Horiguchi, M. & Weinstein, B. M. Angiogenic network formation in the developing vertebrate trunk. (2003) doi:10.1242/dev.00733.
45. Denk, W., Strickler, J. H. & Webb, W. W. Two-photon laser scanning fluorescence microscopy. *Science (1979)* **248**, (1990).
46. Zipfel, W. R. *et al.* Live tissue intrinsic emission microscopy using multiphoton- excited native fluorescence and second harmonic generation. *Proceedings of the National Academy of Sciences* **100**, 7075–7080 (2003).
47. Datta, R. *et al.* Interactions with stromal cells promote a more oxidized cancer cell redox state in pancreatic tumors. *Science Advances* **8**, (2022).
48. Ma, N. *et al.* Label-free assessment of pre-implantation embryo quality by the Fluorescence Lifetime Imaging Microscopy (FLIM)-phasor approach. *Scientific Reports* **9**, (2019).
49. Zipfel, W. R. *et al.* Live tissue intrinsic emission microscopy using multiphoton- excited native fluorescence and second harmonic generation. *Proc Natl Acad Sci U S A* **100**, 7075–7080 (2003).
50. Bird, D. K. *et al.* Metabolic mapping of MCF10A human breast cells via multiphoton fluorescence lifetime imaging of the coenzyme NADH. *Cancer Research* **65**, (2005).

51. Lakowicz, J. R., Szmacinski, H., Nowaczyk, K. & Johnson, M. L. Fluorescence lifetime imaging of free and protein-bound NADH. *Proc Natl Acad Sci U S A* **89**, (1992).
52. Skala, M. C. *et al.* In vivo multiphoton microscopy of NADH and FAD redox states, fluorescence lifetimes, and cellular morphology in precancerous epithelia. *Proc Natl Acad Sci U S A* **104**, 19494–19499 (2007).
53. Sharick, J. T. *et al.* Protein-bound NAD(P)H Lifetime is Sensitive to Multiple Fates of Glucose Carbon. *Scientific Reports* **8**, (2018).
54. Stringari, C. *et al.* Phasor approach to fluorescence lifetime microscopy distinguishes different metabolic states of germ cells in a live tissue. *Proc Natl Acad Sci U S A* **108**, (2011).
55. Wagnieres, G. A., Star, W. M. & Wilson, B. C. Invited Review In Vivo Fluorescence Spectroscopy and Imaging for Oncological Applications. **68**, 603–632 (1998).
56. Févotte, C. & Dobigeon, N. Nonlinear hyperspectral unmixing with robust nonnegative matrix factorization. *IEEE Transactions on Image Processing* **24**, (2015).
57. Heslop, D., von Döbeneck, T. & Höcker, M. Using non-negative matrix factorization in the “unmixing” of diffuse reflectance spectra. *Marine Geology* **241**, 63–78 (2007).
58. Paddock, S. W. Confocal Microscopy, Methods and Protocols, Second Edition. *Humana Press* **1075**, 388 (2014).

The supplementary file References have been updated:

1. M., W. *The zebrafish book*. (University of Oregon Press, 1994).
2. Trinh, L. A. *et al.* A versatile gene trap to visualize and interrogate the function of the vertebrate proteome. *Genes and Development* **25**, 2306–2320 (2011).
3. Shi, W. *et al.* Pre-processing visualization of hyperspectral fluorescent data with Spectrally Encoded Enhanced Representations. *Nature Communications* **11**, 1–15 (2020).
4. Parichy, D. M., Ransom, D. G., Paw, B., Zon, L. I. & Johnson, S. L. An orthologue of the kit-related gene *fms* is required for development of neural crest-derived xanthophores and a subpopulation of adult melanocytes in the zebrafish, *Danio rerio*. *Development* (2000).
5. Digman, M. A., Dalal, R., Horwitz, A. F. & Gratton, E. Mapping the number of molecules and brightness in the laser scanning microscope. *Biophys J* **94**, 2320–2332 (2008).
6. Dalal, R. B., Digman, M. A., Horwitz, A. F., Vetri, V. & Gratton, E. Determination of particle number and brightness using a laser scanning confocal microscope operating in the analog mode. *Microsc Res Tech* **71**, 69–81 (2008).
7. Wagnieres, G. A., Star, W. M. & Wilson, B. C. Invited Review In Vivo Fluorescence Spectroscopy and Imaging for Oncological Applications. **68**, 603–632 (1998).

8. Cutrale, F. *et al.* Hyperspectral phasor analysis enables multi-plexed 5D in vivo imaging. *Nature Publishing Group* (2017) doi:10.1038/nmeth.4134.
9. Taylor, R. C. Experiments in physical chemistry (Shoemaker, David P.; Garland, Carl W.). *Journal of Chemical Education* **45**, (1968).
10. F evotte, C. & Dobigeon, N. Nonlinear hyperspectral unmixing with robust nonnegative matrix factorization. *IEEE Transactions on Image Processing* **24**, (2015).
11. Parslow, A., Cardona, A. & Bryson-Richardson, R. J. Sample drift correction following 4D confocal time-lapse Imaging. *Journal of Visualized Experiments* (2014) doi:10.3791/51086.
12. Schindelin, J. *et al.* Fiji: An open-source platform for biological-image analysis. *Nature Methods* vol. 9 Preprint at <https://doi.org/10.1038/nmeth.2019> (2012).
13. Shimozono, S., Imura, T., Kitaguchi, T., Higashijima, S. I. & Miyawaki, A. Visualization of an endogenous retinoic acid gradient across embryonic development. *Nature* **496**, 363–366 (2013).
14. Islam, M. S., Honma, M., Nakabayashi, T., Kinjo, M. & Ohta, N. pH dependence of the fluorescence lifetime of FAD in solution and in cells. *International Journal of Molecular Sciences* **14**, (2013).
15. Islam, M. S., Honma, M., Nakabayashi, T., Kinjo, M. & Ohta, N. pH dependence of the fluorescence lifetime of FAD in solution and in cells. *International Journal of Molecular Sciences* **14**, (2013).
16. Andrews, L. M., Jones, M. R., Digman, M. A. & Gratton, E. Spectral phasor analysis of Pyronin Y labeled RNA microenvironments in living cells. *Biomedical Optics Express* **4**, (2013).
17. Fereidouni, F., Bader, A. N. & Gerritsen, H. C. Spectral phasor analysis allows rapid and reliable unmixing of fluorescence microscopy spectral images. *Optics Express* **20**, (2012).
18. Cutrale, F. *et al.* Hyperspectral phasor analysis enables multiplexed 5D in vivo imaging. *Nature Methods* **14**, (2017).
19. Stringari, C. *et al.* Phasor approach to fluorescence lifetime microscopy distinguishes different metabolic states of germ cells in a live tissue. *Proc Natl Acad Sci U S A* **108**, 13582–7 (2011).
20. Lanza n , L. *et al.* Encoding and decoding spatio-temporal information for super-resolution microscopy. *Nature Communications* **6**, (2015).
21. Fereidouni, F., Bader, A. N. & Gerritsen, H. C. Spectral phasor analysis allows rapid and reliable unmixing of fluorescence microscopy spectral images. *Opt Express* **20**, 12729–12741 (2012).
22. Scipioni, L., Rossetta, A., Tedeschi, G. & Gratton, E. Phasor S-FLIM: a new paradigm for fast and robust spectral fluorescence lifetime imaging. *Nature Methods* **18**, (2021).

23. Malacrida, L., Ranjit, S., Jameson, D. M. & Gratton, E. The Phasor Plot: A Universal Circle to Advance Fluorescence Lifetime Analysis and Interpretation. *Annual Review of Biophysics* vol. 50 Preprint at <https://doi.org/10.1146/annurev-biophys-062920-063631> (2021).
24. Ranjit, S., Malacrida, L., Jameson, D. M. & Gratton, E. Fit-free analysis of fluorescence lifetime imaging data using the phasor approach. *Nature Protocols* **13**, 1979–2004 (2018).
25. Digman, M. A., Caiolfa, V. R., Zamai, M. & Gratton, E. The Phasor Approach to Fluorescence Lifetime Imaging Analysis. *Biophysical Journal* **94**, L14–L16 (2008).

Reviewer #1 (Remarks to the Author: Impact):

The paper will influence the community - but the discrepancies need to be cleared and explained.

ANSWER: We thank the reviewer for this comment and all the constructive criticism. We believe we addressed all of this reviewer's comments in the answers above and the edits in the manuscript considerably improved the quality of the work.

Reviewer #1 (Remarks to the Author: Strength of the claims):

The main concern is the linear additivity of phasor space and their implementation in this paper. Use of a single harmonic should not be enough for anything more than three components.

ANSWER: We believe we addressed this reviewer's specific comment in the answer to question 17 above and in the introductory answer.

Reviewer #1 (Remarks to the Author: Reproducibility):

I do think the data is reproducible as the imaging is done using a commercial microscope and the authors provide an software to do so. There are details that is missing that need to be provided for the use. This includes calculation and storage of single components for the analysis.

ANSWER: We thank the reviewer for helping us improve this aspect of the manuscript. We believe we addressed this specific comment in answers 1 and 19 above.

Reviewer #2 (Remarks to the Author: Overall significance):

The manuscript of Hsiao Ju Chang et al (from the lab of Prof Cultrale) deals with dynamic (time-lapse) multiplexed imaging and offers a global-based solution for spectral unmixing of hyperspectral imaging data. Therefore, the authors improve the previously published algorithm HySP (Cultrale et al, Nat. Meth. 2017), which uses dimensionality reduction via the phasor approach (normalized discrete Fourier transformation of the hyperspectral 4D fluorescence data). They achieve this improvement by integrating in HySP a linear unmixing of the expected spectral signatures in the phase domain (HyU) - including both extrinsic signals (fluorescent proteins) and intrinsic signals (NAD(P)H, retinol, elastin, etc.). The dimensionality reduction of the phasor approach implies also a global analysis of the spectra (i.e. appreciates similarities of the spectra per voxel) and by that better deals with low signals. A thorough characterization of the laser, detector, read background noise and of their distribution type (Poisson, Gaussian, etc.) and implementation for denoising and additional reference-based preprocessing (SEER, Shi et al, Nat. Commun. 2020) improves not only the image quality but also the success of the hyperspectral unmixing of 8 or 9 emission (intrinsic and extrinsic) signals, at high computation speeds, as impressively demonstrated on simulated data and on time-lapse imaging data of multiple-reporter zebrafish larvae. While being of great interest for the live imaging community, in my opinion, the manuscript needs additional experimental, algorithmic and background (citation of previous work) information to unfold the full potential, as described in detail in the following.

ANSWER: We thank the reviewer for the constructive comments and for providing us an opportunity to considerably improve this work.

Reviewer #2 (Remarks to the Author: Impact):

The relevance of the question/need for simultaneous spectrally multiplexed fluorescent microscopy to allow dynamic (time-lapse) multi-color imaging is tremendous, however, certainly going far beyond the field of developmental biology and zebrafish larvae imaging. This need has been previously recognized in the frame of intravital multi-photon imaging (not hyperspectral), with impact for cancer research (Entenberg et al, 2011), immunology and neurosciences/neuroimaging, just to mention a few examples. Specifically, there have been solutions proposed and demonstrated for dynamic in vivo fluorescence imaging, including unmixing algorithms apart of the state-of-the-art linear unmixing (Rakhymzhan et al, Sci Rep 2017), in which up to 8 extrinsic and intrinsic signals are simultaneously distinguished, while dealing with low SNRs of multi-photon microscopy still remained a challenge. Including this information in the introduction is key, in order to demonstrate the potential general relevance of the present work and to awake a real interest for a broad readership.

ANSWER: This is a very good point. We have edited the introduction to broaden the impact of this work. The introduction now reads:

Standard fluorescence microscopes collect multiple images sequentially, employing different excitation and detection bandpass filters for each label. Recently developed techniques allow for massive multiplexing by utilizing sequential labeling of fixed samples but are not suitable for *in vivo* imaging.^{12,13} Unfortunately, these approaches are ill-suited to separating overlapping fluorescence emission signals, and the narrow bandpass optical filters used to increase selectivity, decrease the photon efficiency of the imaging. (Figs. S1, S2) These limitations have restricted the number of imaged fluorophores per sample (usually 3-4) and risks exposing the specimen to damaging levels of exciting light. This has been a significant obstacle for the dynamic imaging, preventing *in vivo* and intravital imaging from reaching its full potential, with broad impact on research, from developmental biology¹⁴, cancer research¹⁵ and immunology² to neuroimaging¹⁶.

Hyperspectral Fluorescent Imaging (HFI) potentially overcomes the limitations of overlapping emissions by expanding signal detection into the spectral domain.¹⁴ HFI captures a spectral profile from each pixel, resulting in a hyperspectral cube (x,y, wavelength) of data, that can be processed to deduce the labels present in that pixel. Linear unmixing (LU) has been widely utilized to analyze HFI data, and has performed well with bright samples emitting strong signals from fully-characterized, extrinsic fluorophores such as fluorescent proteins and dyes¹⁵⁻¹⁷. However, *in vivo* fluorescence microscopy is almost always limited in the number of photons collected per pixel (due to the expression levels, the bio-physical fluorescent properties, and the sensitivity of the detection system), which reduces the quality of the spectra acquired. While solutions beyond the standard LU have been proposed²¹, the challenge of analyzing low intensity spectral signals remains.

In line with this, it is crucial to demonstrate the power of the presented algorithm for unmixing also intravital multi-photon imaging data in optically more challenging tissues and organisms, which need to deal with much lower signals and SNR values, especially due to massive scattering and wave-front distortions in mammal tissue.

ANSWER: We understand the reviewer's concern on challenging tissues with respect to scattering and wavefront distortions. However, we believe that the underlying physics for combining fluorescent signals and the mathematical requirements for applying our unmixing method should be valid independently from the type of tissue and the depth of the imaging.

With regard to the reviewer's concern for HyU to "deal with much lower signals and SNR values", the now updated plots for the unmixing performance covers a wide range of photon counts and SNR, which will likely to cover a wide range of biological applications.

With regard to "to massive scattering and wave-front distortions in mammal tissue", the reviewer is correct; in principle, highly scattering tissues might affect the fluorescent spectra. However, it should be noted that, in most cases, such massive distortions would require particularly deep imaging. Generally, fluorescent signals decrease greatly in intensity and are lost (scattering/absorption) before reaching such depth, long before these massive wavefront distortions are visible. Undoubtedly, there have to be some spectral distortions caused by the light traveling across a non-uniform biological medium; however, at the relatively shallow sample depths reached before losing signal, these distortions were not measurable at the spectral sampling resolution available on the most common confocal spectral fluorescent microscopes (32 channels, each with 8.9nm bandwidth).

From our previous experience in working with hyperspectral images and utilizing the phasor method (<https://www.nature.com/articles/nmeth.4134> , <https://www.nature.com/articles/s41467-020-14486-8>) in both deep and scattering tissue, we have demonstrated that as long as the signal is not completely indistinguishable from the noise (for $SNR \geq 2$), the phasor method proves reliable in categorizing signals (shown in Supplementary Figures 21, 22, 23, 24 of <https://www.nature.com/articles/s41467-020-14486-8> available at Nature Comm. link https://static-content.springer.com/esm/art%3A10.1038%2Fs41467-020-14486-8/MediaObjects/41467_2020_14486_MOESM1_ESM.pdf).

This correct phasor categorization enables HyU to perform averaging and denoising of spectral signals for unmixing. When working with scattered signals, in our experience, fluorescent signals drop below $SNR=2$ before sufficient scattering and spectral distortions affect the unmixing method. This physical limitation of diffusion of fluorescent signal across scattering and absorbing tissues, at least for the biological samples we imaged (cells, zebrafish, mouse), prevented us from encountering sufficiently distorted spectra to induce major errors in the phasor transform or in the unmixing.

- An example of this is shown in Supplementary Figure 28 of <https://www.nature.com/articles/s41467-020-14486-8> available at Nature Comm. link https://static-content.springer.com/esm/art%3A10.1038%2Fs41467-020-14486-8/MediaObjects/41467_2020_14486_MOESM1_ESM.pdf .
- A very similar discussion with additional figures, experiments, and descriptions is reported in the Peer Review File (pgs 47-49) of <https://www.nature.com/articles/s41467-020-14486-8> available at Nature Comm. link https://static-content.springer.com/esm/art%3A10.1038%2Fs41467-020-14486-8/MediaObjects/41467_2020_14486_MOESM2_ESM.pdf. The image in this discussion shows how stable the signal position on the phasor remains when acquiring fluorescent spectra at deeper parts of the sample.

However, we understand that providing an example of our method in more scattering mammal tissue would facilitate any adopter in their decision making. We have performed the following additional changes to address this point:

1. We have added experiments performed on both freshly excised and fixed mouse tissue. (Supplementary Figures 29, 30)
2. We have included unmixing results from deep imaging performed on zebrafish embryos for further comparison. (Supplementary Figure 23)
3. We expanded the Supplementary Information with details on mouse lines, fluorescent silica beads characterization, mouse tissue samples preparation, and imaging.

1. For mouse experiments, we have included two examples of application for the Hybrid Unmixing method: fresh tissue autofluorescence metabolic imaging and fixed tissue fluorescence imaging.

The first experiment involved unmixing autofluorescent signals in freshly excised mouse kidney tissue. The unmixed signals include NADH bound, NADH free, Retinol, Retinoic acid, and Elastin, in correspondence to the intrinsic signals previously unmixed in the zebrafish experiments. In Supplementary Figure 29, we demonstrate that the same independent autofluorescent signatures used in the zebrafish to perform HyU can be utilized in the fresh mouse tissue. Even as the intensities of the autofluorescent signatures drop rapidly within the first 75 μm of tissue, we demonstrate that HyU performs as previously shown in zebrafish. The unmixed autofluorescent signals within the mouse tissue yield spatial patterns (Sup. Fig 29 A, B, C, E) corresponding to those described in literature (<https://doi.org/10.1117/1.JBO.19.2.020901>, <https://doi.org/10.1681/ASN.2016101153>, <https://doi.org/10.1021/cr900343z>) and to those previously shown in zebrafish. Cross-section visualizations (Sup. Fig 29 B and E) show consistency in the unmixed autofluorescence patterns along the depth of the sample (z-direction) until signal drops to noise level. For these notoriously weak autofluorescent signals, intensities fade out at approximately 75 μm depth.

In the presence of massive wavefront distortions in the fluorescent spectra, the expected and most likely behavior of the HyU would be a deterioration of the unmixing quality. This generally results in a non-organized, more uniform re-assignment of intensities across the unmixed channels. When this scenario happens, severe cross-talk is visible across channels. If one were to visualize these results in a 2-D image, it would result in a “white” color where multiple signals are shown together in the same image. Supplementary Figure 29 panels B and E are single slices of the volume, representing a 1-pixel wide cross-section of the volume, along the XZ and YZ planes respectively. In these cross sections, there is no visible confounding of signals or “white” color, delineating the overlap of multiple fluorescent signals. This suggests that the algorithm, under reasonable imaging conditions, performs as characterized (Sup. Figures

16,17,18,19). This is mainly because fluorescent signals fade to dark before any appreciable distortion affects the system.

Supplementary Figure 29:

Supplementary Figure 29. Intrinsic fluorescent signatures in fresh mouse tissue

Intrinsic fluorescent signatures in fresh kidney tissue of a 7 months Balb-c mouse imaged with 2-photon excitation at 740 nm in a 150 µm deep volume. Despite the increasing scattering effect of this mammal tissue with increasing depth, HyU can perform unmixing of intrinsic fluorescent signals. (A) Volumetric rendering of the unmixing results of five intrinsic fluorescent signatures shows results consistent with literature, as visible in the (B-E) orthogonal views of (C) an unmixed (x,y) cross-section of the volume at 30 µm

depth in the sample and its corresponding **(B)** (x,z) and **(E)** (y,z) projections. **(D)** Averaged autofluorescent signals for each acquired spectral (x,y) section over the 150 μm depth of the volume show a sharp decrease of intensities after 75 μm depth as visible in **E**, the corresponding (y,z) projection.

The second experiment involved unmixing a known fluorescent signal among the background signals within the highly scattering mouse tissue by imaging Cy3 beads injected (described in the new Mouse Sample Preparation section below) into fixed mouse tissue. The ground truth spectra of the Cy3 beads is known, measured in pure bead solution (described in the new Fluorescent silica bead characterization section). With this known constant Cy3 fluorescent spectrum, we can demonstrate:

- a. The efficacy of HyU even when fluorescent signatures are buried deep within highly scattering tissue.
- b. The effect of tissue scattering on the known Cy3 fluorescent spectrum.

Supplementary Figure 30 below shows that the Cy3 beads (magenta) are unmixed with the proper spatial profiles (spherical shape) from the autofluorescent (yellow), background (green), and structural signatures (light blue) of the fixed mouse tissue.

Similarly to Supplementary Figure 29 B,C,E, there is no evident confounding overlap of signals when imaging deeper in the sample. In this case, with stronger autofluorescent (fixative-related) and fluorescent (beads) signals, signals started fading at 110 μm depth in the sample.

In support of our observation that fluorescent spectra, as measured in our commercial instrument, are not excessively distorted in their shape, we plot the spectra corresponding to the Cy3 beads at multiple depths (Supplementary Figure 30 F, G), up to 130 μm deep in the sample, in absolute (Sup. Fig 30 F) and relative (Sup. Fig 30 G) intensity. Absolute peak intensity for some of the deeper spectra is around ~ 1300 Digital Levels, corresponding to 2% of the full 16 bit dynamic range of the detector, close to the noise level of the instrument. This translates to visibly noisy spectra for deeper z-planes of the sample, as shown in the relative intensity plots (panel G), but does not show massive spectral distortions.

In regards to the performance of HyU, the unmixed fluorescent signatures remain consistent with expected spatial patterns within the scattering mouse tissue even as the overall intensity drops. Within the imaging depth we could reach in our instrument, the highly scattering nature of the mouse tissue does not affect the performance of HyU, similar to what we previously demonstrated in zebrafish.

Supplementary Figure 30:

Supplementary Figure 30. Extrinsic fluorescent signatures in fixed mouse tissue

We evaluate the performance of HyU in imaging fluorescent signals in a highly scattering fixed kidney tissue of a 7-month-old Balb-c mouse with embedded Cy3 fluorescent beads (Methods) imaged with 2-photon 850nm excitation up to 150 μm deep. (A) Volumetric rendering of the unmixed results of the signals from fixative autofluorescence (autoFL), Cy3 beads, background, and Second Harmonic Generation (SHG). (B-E) Orthogonal views of the same volume for (C) single (x,y) plane of the volume at a depth of 90 μm with cross-sections (yellow hairlines in C) (B) (z,x) of 18 μm and (E) (y,z) of 4 μm respectively showing sections of the unmixed volume containing Cy3 beads at different depths up to 140 μm. (D) Average intensity value for each acquired (x,y) spectral slice as a function of depth reveals considerable loss of fluorescent signal deeper than 110 μm. (F) Average spectra for each z-plane containing pixels with Cy3 beads signal plotted with absolute intensity (Digital Levels, DL) show decreasing intensity with depth as demonstrated by the area under each spectrum. (G) The same averaged spectra are normalized and plotted with Relative Intensity to show the consistency in the spectral shape as a function of depth in reference to Cy3 beads in solution (dashed line).

2. For completion, we added an example of deep imaging in zebrafish, to demonstrate the similarities in signal attenuation that occurs in this model when compared to highly scattering mouse tissue (Supplementary Figure 23). The phasor plot of the signals provides a consistent distribution, centered at the same coordinates, even for images acquired deeper into the tissue where the detected spectra decrease in intensity and increase in noise. The signal attenuation and noise increase match those seen for the spectra detected in both mouse examples (Sup. Fig 29 and 30).

Supplementary Figure 23. Phasor analysis on signal distortion in deep tissue

Images at different Z-positions of a 3D (x,y,z) dataset of 19 hpf *Tg(ubiq:lyn-tdTomato)* zebrafish acquired from 0 μm to 80 μm (relative to the object) depth displaying at every 13 z-slices. (A-D) Phasors calculated from the single slices at 0 μm , 26 μm , 52 μm , and 78 μm depth. (E-H) Corresponding average intensity images (across the 32 spectral channels) for each z-slice show an expected decrease in fluorescence intensity with depth. (I) Average spectra of the pixels linked to the phasor position bin for the tdTomato fluorescent signature for each of the four presented z-slices (0, 26, 52, 78 μm) show a decrease in the spectral area without change in spectral shape as shown by (J)

normalizing the average spectra shown in I to each spectrum's maximum value. This demonstrates the spectral shape does not change across different depths (z-planes), while the overall intensity decreases. **(K)** Randomly selected spectra from the raw spectral image at 0 μm (blue) and 26 μm (red), five spectra for each image and similarly **(L)** for 52 μm (green) and 78 μm (yellow). Two yellow and three green spectra are not clearly visible because of low signal intensity.

3. We expanded the supplementary information to include details on mouse lines, fluorescent silica beads characterization, mouse tissue sample preparation, and imaging:

Mouse lines

Mice imaging was approved by the Institutional Animal Care and Use Committee (IACUC) of University of Southern California, Protocol #21311. Experimental research on vertebrates complied with institutional, national and international ethical guidelines. Animals were kept on a 13:11 hours light:dark cycle. Animals were breathing double filtered air, temperature in the room was kept at 68–73 F, and cage bedding was changed weekly. All these factors contributed to minimize intra- and inter-experiment variability. Adult Balb-c mice were euthanized via overdose of isoflurane followed by cardiac puncture. Kidneys were quickly harvested from the mouse, washed in PBS, and cut longitudinally alongside the midsection in order to expose the inner part of the organ. The two halves of the organ were arranged onto a microscope slide for imaging.

Fluorescent silica beads characterization

One fluorescent silica beads solution (Nanocs, Inc.) labeled with Cy3 (Si500-S3-1, 0.5mL, 0.5 μm , 1% solid, lot# 1608BRX5) was characterized in its spectral fluorescence emission and physical size.

10x dilution in PBS of the beads was placed on a no. 1.5 imaging coverglass and spectrally characterized using spectral mode on a Zeiss LSM 780 laser confocal scanning microscope equipped with a 32-channel detector using 40x/1.1 W LD C-Apochromat Korr UV-VIS-IR lens utilizing a 2-photon laser at 740 nm to excite fluorescence from the beads, using a 690 nm lowpass filter to separate excitation and fluorescence. Spectra obtained from multiple beads with the same label were averaged, producing the reference spectrum reported in Sup. Figure 30 G (dashed line). Fluorescent silica bead size and concentration were determined via nanoparticle tracking analysis (NTA) on the Nanosight NS300 (Malvern Panalytical). Samples were run 5 times and results averaged for final size and concentration values reported.

Mouse Sample preparation

For autofluorescent measurements, mouse organ samples were collected from Balb-c mice. Following euthanasia, organs were resected and washed in Phosphate Buffered Saline (PBS) to remove residual blood and kept in PBS until imaging preparation. Organs were sectioned in order to image the internal architecture and mounted on a glass imaging dish with sufficient PBS to avoid dehydration of the sample. Following imaging, all samples were fixed in a 10% Neutral Buffered Formalin solution at 4-^oC.

For ex vivo bead characterization in tissue, mouse organ samples were collected from Balb-c mice. Following euthanasia, organs were resected and washed in PBS followed by incubation for at least 24 hours in 10% buffered formalin. The kidney was then removed from the fixative and sectioned into smaller ~5x5x5mm pieces for imaging. A fluorescent silica beads stock (Nanocs, Inc.) labeled with Cy3 (Si500-S3-1, 0.5mL, 0.5um, 1% solid, lot# 1608BRX5) and previously characterized was prepared using a 10x dilution of the fluorescent beads from their stock concentration. Beads were injected in the sample using 50ul of these solutions loaded into a 0.5mL syringe with a 28g needle. The kidney sections were then placed in imaging dishes with a small volume of PBS to keep the samples hydrated prior to imaging.

Image Acquisition.

Images were acquired on a Zeiss LSM 780 laser confocal scanning microscope equipped with a 32-channel detector using 40x/1.1 W LD C-Apochromat Korr UV-VIS-IR lens at 28-^oC.

Samples of *Gt(cItca-Citrine)*, *Tg(ubiq:lyn-tdTomato)*, *Tg(fli1::mKO2)*, and *Tg(ubiq:Lifeact-mRuby)*, were simultaneously imaged with 488 nm and 561 nm laser excitation, for citrine, tdTomato, mKO2, and mRuby. A narrow 488 nm/561 nm dichroic mirror was used to separate excitation and fluorescence emission. Samples were imaged with a 2-photon laser at 740 nm to excite autofluorescence, using a 690 nm lowpass filter to separate excitation and fluorescence.

Samples of mouse kidney tissue were imaged in 2-photon exciting at 740 nm or 850 nm with a 690+ nm lowpass filter, at 37C incubation.

For all samples, detection was performed at the full available range (410.5-694.9nm) with 8.9nm spectral binning.

Referring to the algorithm itself and to its characterization, the evolution from hyperspectral multiplexed imaging using the phasor approach HySP (Cultrale et al, 2017, Nat Meth), enhanced by preprocessing the data to account for various experimental noise via SEER (2020, Nat Commun) and finally by applying linear unmixing in the hyperspectral phase space, bringing additional significant accuracy to the unmixing capacity of the data is currently not clear in the manuscript and needs to be elaborated in the introduction, to emphasize the novelty of the present work.

ANSWER: We have improved the introduction to emphasize the novelty of HyU with respect to previously published work:

We have developed Hybrid Unmixing (HyU) as an answer to the challenges that have limited the wider acceptance of HFI for *in vivo* imaging. HyU employs the phasor approach²⁶ merged with traditional unmixing algorithms to untangle the fluorescent signals more rapidly and more accurately from multiple exogenous and endogenous labels. The phasor approach²⁶, a popular dimensionality reduction approach for the analysis of both fluorescence lifetime and spectral image analysis^{27–29} has been shown to provide key advantages to HyU, including spectral compression, denoising, and computational reduction for both pre-processing³⁰ and unmixing^{31–33} of HFI datasets.

Standard Phasor analysis^{26,27,34–41} is fully supervised and requires a manual selection of regions or points on a graphical representation of the transformed spectra, called the phasor plot. HyU utilizes phasor processing as an encoder to aggregate similar spectra and applies unmixing algorithms, such as LU, on them to provide unsupervised analysis of the HFI data, simplifying the data processing and removing user subjectivity. Our results show that HyU offers three key advantages: (1) improved unmixing over conventional LU, especially for low intensity images, down to 5 photons per spectra; (2) simplified identification of independent spectral components; (3) dramatically faster processing of large datasets, overcoming the typical unmixing bottleneck for *in vivo* fluorescence microscopy.

We further clarified the advantages of HyU over our previous work HySP in a Supplementary Note 4.

Supplementary Note 4: Improvements of HyU over the standard phasor analysis

Linearity of combinations is the general assumption for most of the spectral analysis algorithms in Hyperspectral Fluorescence Imaging (HFI). Each pixel is assumed to contain a linear combination of the independent spectral signatures, or endmembers,

contained in the sample. This assumption requires knowledge, or identification, of the independent spectra within the sample. In standard linear unmixing algorithms, the extraction of relative amounts of spectra (ratios) is conducted on a pixel-by-pixel basis, at the expense of computational costs. Disrupted experimental signals, in the case of lower Signal to Noise Ratio (SNR) spectra, complicate the detection of spectral endmembers and reduce the accuracy of ratio determination. These standard unmixing algorithms, however, have the advantage of being unsupervised with the possibility of automating the analysis process.

The phasor approach has become a popular dimensionality reduction approach for the analysis of both fluorescence lifetime and spectral image analysis¹³⁻¹⁵. Phasors provide key advantages, including spectral compression, denoising, and computational reduction for both pre-processing³ and unmixing^{8,16,17} of HFI datasets. Phasor analysis overcomes the challenge of low SNR data analysis that limits standard unmixing algorithms, providing a multiplexing solution to a need. The phasor transform is a lossy encoder that in principle carries a reduced percentage of the information compared to the original clean data¹⁸. In the imaging of fluorescent signals, where signal to noise often decreases to lower digits, the encoding loss is less relevant compared to the noise of the fluorescent signals. This fundamental advantage of increasing SNR in noisy data has made the phasor method a valuable tool for fluorescence microscopy, both for Lifetime and Spectral Fluorescence Microscopy. This point is reported by multiple groups using phasors¹⁸⁻²¹ and, more recently, nicely described in the work of Scipioni et al²². Standard Phasor analysis²³⁻²⁵ is fully supervised and requires a manual selection of regions or points on a graphical representation of the transformed spectra, called the phasor plot. Each selection of a region in the phasor plot associates pixels containing similar spectra to the same fluorophore, forming an output channel that contains wavelength integral of intensities with unitary ratiometric value. This “winner takes all” approach is suitable when fluorophores for each single excitation light are spectrally overlapping and spatially disperse (Sup. Figure 24), but requires separate acquisition of different excitation wavelengths for demultiplexing spatially and spectrally overlapping fluorophores (Sup. Figure 25).

HyU uses the phasor transform to group pixels with similar spectral shape within each phasor histogram bin. This approach maintains the advantage of compressing, denoising and simplifying identification of clean endmember fluorescent spectra. However, HyU improves on the robustness of the analysis. The denoised signals are maintained in a hybrid phasor and wavelength domain, and therefore can be unmixed with a multitude of standard unmixing algorithms (Sup. Figure 13), such as Linear Unmixing or Fully Constrained Least Squares. These standard unmixing approaches can operate without supervision and provide for each pixel the ratios for a set of spectral signals, overcoming some of the limitations of phasor, but generally do not perform well in experimental

conditions with reduced and compromised signals, such as in fluorescence, and require extensive computational time for high spectral-count datasets. HyU provides wavelength-based denoised spectra that enable these standard algorithms to outperform their pixel-by-pixel typical application, both in quality of the results (Sup. Figures 16-19), owing to cleaner and better defined fluorescent spectra in each phasor bin, and, generally, in speed, owing to the phasor dimensionality reduction. HyU performs well for single excitation light when fluorophores are spectrally overlapping both when they are spatially disperse or co-localized, providing a ratio for each independent spectrum currently unmixed. Our data suggests HyU has reasonable performance for up to 8 different fluorophores per dataset, for each single excitation wavelength. In an experiment with a carefully chosen palette of labels, where octuples of fluorophores can be excited by a single wavelength, with an instrument capable of spectral acquisition with 5 standard and sufficiently spectrally separated excitation wavelengths in 5 sequential acquisitions (one for each excitation light), HyU could, in principle, unmix 40 signals. This performance however decreases with the number of channels (Sup. Figures 18, 19) showing a small deterioration at 8 channels and limitations at 4.

Reviewer #2 (Remarks to the Author: Strength of the claims):

A. Referring to the broad applicability of the algorithm and the interest for a large community:

A.1. As previously mentioned, in order to prove the value of the approach presented in this manuscript, multiplexed time-lapse imaging in a mammal (adult mouse or rat or human) tissue is key and experimental data on this need to be added to the manuscript. I believe, one 4D (3D + time) imaging example showing 8-9 distinct emission signals would be absolutely convincing.

ANSWER: We agree with this reviewer that a multiplexed time-lapse imaging in a mammal (adult mouse or rat or human) tissue showing 8-9 distinct emission signals would absolutely be convincing. However, such an experiment would require a specialized team with expertise in mouse/rat multi-color labeling, a dedicated mouse/rat colony, specialized equipment for both anesthesia and imaging of the samples, and a complex survival imaging animal protocol. We currently do not have such expertise in our group, which mainly uses zebrafish, lack the compliant instrumentation for performing mammal timelapse imaging under anesthesia as regulated by IACUC of USC, and do not have active mouse protocols that would allow such imaging. Performing such a complex experiment would require 2-3 years of work, after securing the necessary funding for additional personnel and instrumentation, while, arguably, there likely is a very limited number of laboratories with the capability and equipment of routinely performing 4D imaging of mouse/rat samples with 8-9 labels. For these reasons we believe such an experiment would, itself, be worthy of an independent publication. However, it would not further the demonstration of the method in this manuscript.

As such, we believe this experiment to be beyond the scope of this manuscript, which presents a fluorescence unmixing method for the broader audience. We agree with the reviewer that example applications in mouse should be part of this manuscript (as we stated above) with respect to the concerns of distortions of spectra in scattering tissues. We have included additional mouse tissue experiments in this revision that we believe address the performance of the method when applied to mammalian samples. We kindly refer to the above answer to this reviewer (Remarks to the Author: Impact) for details of the additional experiments.

B. Referring to the unmixing approach:

B.1. In order to judge the added value of the integration of linear unmixing and of reference extrinsic and intrinsic spectral signatures on the performance of unmixing, a thorough comparison with the previously available HySP (Cultrale et al, 2017) needs to be provided, additionally to the comparison to state-of-the-art linear unmixing algorithms already included in the manuscript.

ANSWER: To address this point:

1. We have improved the introduction to emphasize the novelty of HyU with respect to previously published work (see answer above for in-line text).
2. We included a full comparison with different types of samples in Supplementary Figures 24 and 25.
3. We added an extensive discussion that outlines the advantages of HyU over the previously available HySP in the new Supplementary Note 4.

1. The introduction now refers to our previous work and overviews the improvements of HyU over it. The excerpt is copied for reference in the last answer to this reviewer's "Impact" section.
2. We included comparisons of HyU vs HySP with different sample types, where for each laser excitation, the sample contains spectrally overlapping spatially disperse labels or both spectrally and spatially overlapping fluorophores.

Supplementary Figure 24:

Supplementary Figure 24. Comparison of HyU vs HySP results from a spectrally overlapping and spatially disperse sample

Results are presented for unmixing using HyU and Hyperspectral Phasors (HySP) on a spectrally overlapping spatially disperse dataset collected from a tri-labeled transgenic zebrafish embryo obtained by injecting mRNA-encoding H2B–cerulean (cyan) in double transgenic embryos *Gt(desm-citrine) ct122a/+;Tg(kdrl:eGFP)* (magenta and yellow, respectively) (A-F) HyU unmixing results and (G-L) HySP unmixing results renderings for the dataset. Line profiles of (F) HySP (L) HyU analysis results (B, H dashed line) show the similarity in signal between the two methods for all channels within a non-overlapping sample. (A,F) Volumetric images show a similarity between the HyU and HySP results. This is further demonstrated for the results in a (B,H) single z-slice, for just the (C,I) Citrine channel, the (D,J) Cerulean channel, and the (E,K) mCherry channel. (F,L) Line profiles for the lines shown in B and H, respectively, also demonstrate the similar results of HyU and HySP for spatially non-overlapping samples.

Supplementary Figure 25:

Supplementary Figure 25. Comparison of HyU vs HySP results from a spectrally overlapping and spatially non-disperse sample

Results for unmixing using HyU and HySP on a spectrally overlapping and spatially non-disperse dataset collected from a 5 dpf dual-labeled transgenic zebrafish embryo: *Gt(citca-citrine);Tg(fli1:mKO2)*, presenting frequent combinations of signals in pixels across the dataset. **(A-E)** HyU unmixing results and **(F-K)** HySP unmixing results for the dataset. **(A,F)** Volumetric images show the expected signal overlaps between channels for the HyU result and a more distinct separation in the HySP result. This is further demonstrated for the results in a **(B,G)** single z-slice, for just the **(C,H)** mKO2 channel, and the **(D,I)** Citrine channel. **(E,J)** Line profiles for the lines shown in **B** and **G**, respectively, demonstrate the fractional nature of HyU results compared to the winner-takes-all analysis of HySP.

3. Supplementary Note 4, replicated in our first answer to this reviewer, furthers the discussion.

B.2. A central advantage of the here presented approach is the capacity of dealing even with low signals, i.e. unmixing even low endogenous signals, such as NAD(P)H, even free and bound – having extremely similar emission spectra (one reason why their fluorescence lifetime has been used to resolve the two states). The authors show the improvement referring to number of photons per spectrum, however, in order to judge the true improvement brought by the algorithm for real imaging data (which includes background with diverse types of noise distributions), the unmixing quality needs to be related to the signal-to-noise (SNR) ratio per voxel. While mentioning SNR in the text, no values or comparison are provided in this sense – it is important to mention how the SNR as such (not only the number of photons per spectrum) impacts on the spectral resolution, i.e. how similar can be two spectra at a certain SNR to be able to still resolve them?

ANSWER: We thank the reviewer for this very good point. To address this comment we prepared:

1. A plot correlating the SNR as a function of spectrum-type and photons/spectrum
 2. A definition of the Spectral SNR
1. The plot represents how photons per spectra translate to a more standardized definition of SNR, utilizing the fluorophores' spectra from this manuscript. We provide these plots to show the direct relationship between photons per spectrum and the SNR of a single spectra. The reason for demonstrating the relationship between photons per spectra and SNR instead of directly calculating and presenting the SNR values is because the exact description and quantification of SNR becomes highly convoluted and unintuitive for

combinations of multiple spectra. It is important to notice the differences in Spectral SNR between fluorophores as the result of the different spectra shapes which cover different numbers of channels in our instrument-simulating algorithm (410-692nm spectral detection range, 32 bands each with 8.8nm bandwidth).

Supplementary Figure 28:

Supplementary Figure 28. Relationship between Spectral SNR and Photon/Spectrum

The direct relationship between SNR and photons per spectrum is shown here using the calculation of Spectral SNR for varying levels of photons per spectrum. The spectral SNR has a general trend of increased values with increasing photons per spectrum, but it is not a truly monotonic function. This non-monotonicity demonstrates the limitations of SNR when analyzing spectral images. **(A)** Absolute Spectral SNR and **(B)** Relative Spectral SNR follow the same trends of higher values with increasing photons per spectrum. However, the Relative Spectral SNR better differentiates the effects of the differing spectral shapes on the SNR. Citrine, mKO2, mRuby, and tdTomato each have easily distinguished values for the slope of the regression in ascending order. tdTomato has a spectral shape which provides the best SNR while Citrine provides the worst SNR, even with the same number of photons per spectrum.

2. For this simulated hyperspectral (multispectral/multichannel) fluorescent data, the term of Spectral SNR is calculated inclusively of the spectral dimension. We define the criteria used for calculating Spectral SNR in a new section in Supplementary Methods. Briefly, these Spectral SNR calculations are designed to also include the noise resulting from the stochastic emission of fluorescent photons which disrupts the shape of spectra, therefore compromising the unmixing analysis. The additional section in Supplementary Methods reads:

Spectral Signal to Noise Ratio

Since each synthetic dataset has a ground truth, the SNR can be calculated by comparing the simulated image to the ground truth. Since these are hyperspectral images, we extend the definition of SNR to the wavelength dimension of the data and use the term Spectral SNR. We define two types of Spectral SNR, Absolute Spectral SNR and Relative Spectral SNR.

Spectral SNR is calculated as follows for each single spectrum simulation. First, for each pixel and channel, the absolute value of the difference is taken between the ground truth intensity and the simulated intensity. Then the mean is calculated over all of the pixels for each channel. Finally, the sum is taken over all of the channels and divided by either 32 for the absolute SNR, or the number of channels with signal for the relative SNR. The number of channels with signal is calculated by checking if there is a statistically significant number of pixels in a single channel with a pixel SNR value greater than zero.

$$Absolute\ SNR = \frac{\sum_{c=1}^{32} \frac{\sum_{n=1}^P |i_{sim} - i_{gnd}|}{P}}{32} \quad (23)$$

$$Relative\ SNR = \frac{\sum_{c=1}^C \frac{\sum_{n=1}^P |i_{sim} - i_{gnd}|}{P}}{C} \quad (24)$$

Where i_{gnd} is the intensity per pixel per channel for the ground truth data, i_{sim} is the intensity per pixel per channel for the simulated (noisy) data, P is the total number of pixels, and C is the number of channels with signal.

B.3. A corner stone in acquiring better unmixing is the availability of appropriate reference spectral signatures. Whereas the current software provides the spectra necessary for the data shown in the manuscript and gives the opportunity for the users to identify signatures in their own data, the manuscript remains elusive of how the user can differentiate between a real spectral signature and different types of optical or electronical background and interferences – as well known from the use of the phasor approach in fluorescence lifetime imaging, a major challenge when dealing with experimental noisy imaging data in the frequency (phase) domain. The manuscript would benefit from including such a guide to validate the capacity for external use of the algorithm.

ANSWER: We have expanded Supplementary Note 1 titled “Identification of spectra and new components with HyU” with further clarifications, insights and references on how to distinguish

independent spectral signatures from various types of noise. Supplementary Note 1 now reads:

Supplementary Note 1: Identification of spectra and new components with HyU

Identification of independent spectral components has been an adversity for unmixing hyperspectral data. First, the collected spectra may be distorted by reduced SNR. Secondly, excitation of intrinsic signals causes uncertainty of biological sample. Favorably, HyU simplifies this process by adapting Phasor approach and achieving semi- or full-automation process for spectra identification and selection. In HyU spectra can be loaded from an existing library, virtually automating the analysis process. Pre-identified cursors are generated from common fluorophores such as mKO2, tdTomato, mRuby, Citrine. In our experience, obtaining fluorescence spectra from experimental samples has some advantages compared to utilizing spectra from existing library, as they account for a multitude of experimental and instrumental settings. Imaging settings such as different types of lenses or optical filters (Sup. Figure 4, C and D) together with factors within the microenvironment of samples, such as pH or temperature have the potential to alter the fluorescence spectral emissions¹². In the presence of unexpected fluorescent signals, spectra can also be selected and visualized directly from the phasor. Phasors facilitate the identification of unexpected independent components and their distinction from the multiple system noises. A noise-free spectrum will appear as a single point on the phasor plot, while a spectrum affected by instrument and electronic noises will mainly appear as a gaussian distribution, centered on the original spectral signal⁶. Conversely, a randomized noise across the multiple spectral channels will not produce a clustered aggregate of spectra on the phasor. A constant spectral noise, with a distinct spectrum (e.g. a constant light leakage into the system), would produce a distinct phasor cluster and could be selected for unmixing. The phasor plot representation is a 2D-histogram and provides insights into the frequency of occurrence for these signals. These unexpected independent components in samples often appear as “tails” on the phasor distributions (Sup. Figure 11, C). In our HyU graphical interface, clicking on the phasor visualizes the spectra within a small area (9x9 bins by default, with size adjustable from the interface) of the phasor histogram (Figure 1 D). In the example in Sup. Figure 9-11 we identify 5 distinct endmembers on the Phasor (Sup. Figure 10, C), visualize their spectra identifying Citrine, mRuby, Td-Tomato, mKO2, and one strong autofluorescence signature. The use of Residual Phasor Map (Sup. Figure 11, B) allows for identification of areas in the phasor with high amount of residuals, likely corresponding to a missing endmember in the unmixing. Residual Image Maps (Sup. Figure 11, C) provide a rapid overview of residuals in the image data, for identification of location in the dataset of the missing endmember.

B.4. Finally, fully agreeing with the authors that the number of detectors may be varied, depending on the imaged sample type and on the excitation strategy, in order to acquire an emission signal at all, an analysis of how the number of detectors (channels) impacts on the resolution between different signatures (spectra) is needed also for less than 32 detectors (4 to 6 channels being the reality in many labs due to truly low fluorescence signals in deep tissue, e.g. of mice or of humans).

ANSWER: To characterize how the number of channels affects the hybrid unmixing analysis, we have assembled a complex simulation matrix representing the performance of HyU in unmixing 2 to 8 labels as a function of the spatial overlap in the sample. This simulation matrix is built on top of the complex simulation we designed (further expanded in the Hyperspectral Fluorescence Image Simulation section of the supplementary methods), which is soon to be published in a separate manuscript. This simulation accounts for a multitude of real-world noises in experimental samples that are regularly imaged (stochasticity of fluorescence spectral emission, poisson, readout noise, electronics transfer noise, detector sensitivity at different wavelength).

We replicate the expanded Hyperspectral Fluorescence Image Simulation section:

Hyperspectral Fluorescence Image Simulation

The model simulates spectral fluorescent emission by generating a stochastic distribution of photons with profile equivalent to the pure reference spectra (as described in Sup. Note 1). The effect of photon starvation, commonly observed on microscopes, is synthetically obtained by manually reducing the number of photons in this stochastic distribution.

Detection, Poisson and signal transfer noises are then added to produce 32-channel fluorescence emission spectra that closely resemble those acquired on microscopes. The simulations include accurate integration of dichroic mirrors and imaging settings.

Simulation Types:

Biologically comparable simulations

To quantify the performance of HyU vs LU for microscopy data acquired experimentally, we generated synthetic data where each input spectra was organized with intensity distributions taken from experimental data of fluorescently labeled biological samples.

We calibrated the analog (Digital Levels) to photon counting rate based on existing literature^{5,6}. Experimental data was discretized to photons to produce biologically relevant photon masks with distributions of signals highly resembling those of the samples. This provided intensities and ratios which closely resemble those acquired from a confocal microscope while allowing control over the effects of photon starvation.

Spatially and spectrally overlapping simulations

We also included simulations to quantify the performance of HyU vs LU with respect to the number of spectral combinations and of endmembers. The results are summarized in Supplementary Figures 16-19 in the form of matrices of spectral overlap (0 to 100%, steps of 10%, X-axis) by number of endmembers (2 to 8 endmembers, Y-axis) representing the Relative Mean Squared Error (RMSE) (Supplementary Methods, Performance quantification). Each RMSE value reported in a matrix is the average of analysis of a 1024x1024 pixels image simulation with a spectral dimension of 32 channels matching the spectral range and bandwidth of the detectors in commercial confocal microscopes (LSM 780, Carl Zeiss). These simulations were created with artificial intensity distributions so that a simulation with X% overlap and n fluorophores would have X percentage of pixels with a randomized ratio of n input spectra. As an example, for a simulation with 6 fluorophores and 50% overlap, the simulated dataset would have 50% of the pixels contain a randomized combination of the 6 fluorophores, while the remaining pixels contain a single fluorophore. This allowed us to investigate the effects of an increasing number of spectral combinations on the compressive nature of the phasor method for HyU.

The results are reported in Supplementary Figure 19 and described in Supplementary Note 4 (reported above):

Supplementary Figure 19. RMSE improvement for simulated fluorescent and autofluorescent spectral combinations highlights decreasing overall performance across decreased number of spectral channels

Fifteen matrices demonstrate the RMSE improvement of HyU with respect to LU when unmixing a collection of synthetic data with 2 to 8 extrinsic and intrinsic labels (Y axis of each matrix) as a function of the spatial overlap of these labels in a sample (X axis of each matrix). In the matrix, 0% overlap denotes simulations with spatially distinct fluorophores, where each pixel corresponds to a single fluorophore, while simulations with 100% overlap contain, in every pixel, a randomized ratio of the n extrinsic and intrinsic fluorophores. Each one of the values reported in a matrix is the average of a 1024x1024x32 pixels simulation and shows the RMSE improvement of HyU to LU. Columns in the figure represent RMSE improvement matrices with 3x denoising filters across an increasingly binned number of spectral channels (32, 16, 8, 6, 4) applied with a total number of photons per pixel at (A) 16 (B) 32 (C) 48. When utilizing 32 spectral channel data, RMSE improvements reach up to the previously reported 100% for highly overlapping fluorophores. Successively increasing the binning across the wavelength dimension (and therefore decreasing the number of channels) shows a slow downward trend of the RMSE improvement until the 4 spectral channel matrices, where the RMSE improvement drops drastically down to below 25%, especially for more than 3 labels. Spectra utilized for this simulation are reported in Sup. Figure 20B.

The excerpt of Supplementary Note 4 clarifying the performance / number of channels:

In an experiment with a carefully chosen palette of labels, where octuples of fluorophores can be excited by a single wavelength, with an instrument capable of spectral acquisition with 5 standard and sufficiently spectrally separated excitation wavelengths in 5 sequential acquisitions (one for each excitation light), HyU could in principle unmix 40 signals. This performance however decreases with the number of channels (Sup. Figure 25) showing a small deterioration at 8 channels and limitations at 4.

Reviewer #2 (Remarks to the Author: Reproducibility):

The current version of the HySP platform was easy to use and the provided sample data delivered similar results as those shown in the manuscript.

ANSWER: We thank the reviewer for the constructive feedback and for helping us improve the quality of the work presented here.

Reviewer #3 (Remarks to the Author: Overall significance):

In this report, Chiang and co-workers presented the Hybrid Unmixing (HyU) method for the efficient and robust analysis of multiple fluorescent signals. The authors employ the spectral phasor method for reducing spectral data dimension and denoising noises in the imaging system. The superiority of the proposed method has been demonstrated compared to the conventional linear unmixing method by exploiting computer simulation and experimental results. This article seems to be timely the report as increasing the biomedical applications using hyperspectral imaging methods. However, I found that there are some confusing points to be addressed clearly to publish this manuscript in Nature Portfolio.

ANSWER: We thank the reviewer for the valuable comments on the manuscript. Following the reviewer's comments we have:

1. improved the explanation of the advantages of HyU by modifying our introduction and adding a new Supplementary Note 4
2. extended the description of the simulations with more detail, providing our reasoning and multiple figures to demonstrate why they provide a realistic replication of standard experimental conditions and are reliable as standards to compare our results with our unmixing methods
3. expanded our Supplementary Note 1 with further clarifications, insights, and references on how to distinguish both known and unknown independent spectral signatures within the phasor plot
4. demonstrated our unmixing method in both scattering and non-optically clear tissue by performing hyperspectral fluorescence experiments in both freshly excised and fixed mouse tissue, summarizing the HyU unmixing results in two new supplementary figures as well as demonstrating the unmixing method for deep imaging in zebrafish.

Comments:

1) Hyperspectral phasor compresses spectral dimension by exploiting real and imaginary parts of Fourier transformation. Moreover, there were reports that hyperspectral phasor could be applied for multiplexed fluorescence imaging. If there are any advantages of combining phasor and spectral unmixing methods, please describe them clearly in the Introduction.

ANSWER: To address this comment we have:

1. modified the introduction to clarify the advantages of combining phasor and spectral unmixing
2. added a Supplementary Note 4 to further the clarification of the advantages for HyU compared to previous reports of hyperspectral phasors

1. The introduction now reads:

HyU employs the phasor approach²⁶ merged with traditional unmixing algorithms to untangle the fluorescent signals more rapidly and more accurately from multiple exogenous and endogenous labels. The phasor approach²⁶, a popular dimensionality reduction approach for the analysis of both fluorescence lifetime and spectral image analysis²⁷⁻²⁹ has been shown to provide key advantages to HyU, including spectral compression, denoising, and computational reduction for both pre-processing³⁰ and unmixing³¹⁻³³ of HFI datasets. Standard Phasor analysis^{26,27,34-41} is fully supervised and requires a manual selection of regions or points on a graphical representation of the transformed spectra, called the phasor plot. HyU utilizes phasor processing as an encoder to aggregate similar spectra and applies unmixing algorithms, such as LU, on them to provide unsupervised analysis of the HFI data, simplifying the data processing and removing user subjectivity. Our results show that HyU offers three key advantages: (1) improved unmixing over conventional LU, especially for low intensity images, down to 5 photons per spectra; (2) simplified identification of independent spectral components; (3) dramatically faster processing of large datasets, overcoming the typical unmixing bottleneck for *in vivo* fluorescence microscopy.

2. We have added Supplementary Note 4 to directly address the improvements of HyU over previously published works on spectral phasors, particularly in this portion of the note:

HyU uses the phasor transform to group pixels with similar spectral shape within each phasor histogram bin. This approach maintains the advantage of compressing, denoising and simplifying identification of clean endmember fluorescent spectra. However, HyU improves on the robustness of the analysis. The denoised signals are maintained in a hybrid phasor and wavelength domain, and therefore can be unmixed with a multitude of standard unmixing algorithms (Sup. Figure 13), such as Linear Unmixing or Fully Constrained Least Squares. These standard unmixing approaches can operate without supervision and provide for each pixel the ratios for a set of spectral signals, overcoming some of the limitations of phasor, but generally do not perform well in experimental conditions with reduced and compromised signals, such as in fluorescence, and require extensive computational time for high spectral-count datasets. HyU provides wavelength-based denoised spectra that enable these standard algorithms to outperform their pixel-by-pixel typical application, both in quality of the results (Sup. Figures 16-19), owing to cleaner and better defined fluorescent spectra in each phasor bin, and, generally, in speed, owing to the phasor dimensionality reduction. HyU performs well for single excitation light when fluorophores are spectrally overlapping both when they are spatially disperse or co-localized, providing a ratio for each independent spectrum currently unmixed.

2) If I understood correctly, numbers of photons (For instance, 5 photons per spectral in the last paragraph in Introduction) were calculated from the computer simulation. If so, this quantitative value is significantly affected by the noise levels used in the simulation. Therefore, it would be good to add these values were obtained from the simulation for clarity.

ANSWER:

We have edited the Section “Hyperspectral Fluorescence Image Simulation” in the Supplementary Materials to include more details on the simulations:

Hyperspectral Fluorescence Image Simulation

The model simulates spectral fluorescent emission by generating a stochastic distribution of photons with profile equivalent to the pure reference spectra (as described in Sup. Note 1). The effect of photon starvation, commonly observed on microscopes, is synthetically obtained by manually reducing the number of photons in this stochastic distribution.

Detection, Poisson and signal transfer noises are then added to produce 32-channel fluorescence emission spectra that closely resemble those acquired on microscopes. The simulations include accurate integration of dichroic mirrors and imaging settings.

Simulation Types:

Biologically comparable simulations

To quantify the performance of HyU vs LU for microscopy data acquired experimentally, we generated synthetic data where each input spectra was organized with intensity distributions taken from experimental data of fluorescently labeled biological samples.

We calibrated the analog (Digital Levels) to photon counting rate based on existing literature^{5,6}. Experimental data was discretized to photons to produce biologically relevant photon masks with distributions of signals highly resembling those of the samples. This provided intensities and ratios which closely resemble those acquired from a confocal microscope while allowing control over the effects of photon starvation.

Spatially and spectrally overlapping simulations

We also included simulations to quantify the performance of HyU vs LU with respect to the number of spectral combinations and of endmembers. The results are summarized in Supplementary Figures 16-19 in the form of matrices of spectral overlap (0 to 100%, steps of 10%, X-axis) by number of endmembers (2 to 8 endmembers, Y-axis) representing the Relative Mean Squared Error (RMSE) (Supplementary Methods, Performance quantification). Each RMSE value reported in a matrix is the average of analysis of a 1024x1024 pixels image simulation with a spectral dimension of 32- channels matching the spectral range and bandwidth of the detectors in commercial confocal microscopes (LSM 780, Carl Zeiss). These simulations were created with artificial intensity distributions so that a simulation with X% overlap and n fluorophores would have X percentage of pixels with a randomized ratio of n input spectra. As an example, for a simulation with 6 fluorophores and 50% overlap, the simulated dataset

would have 50% of the pixels contain a randomized combination of the 6 fluorophores, while the remaining pixels contain a single fluorophore. This allowed us to investigate the effects of an increasing number of spectral combinations on the compressive nature of the phasor method for HyU.

The simulation algorithm is soon to be published in a separate manuscript. It includes extensive characterization of instrument noise and produces distributions of signals that closely resemble the corresponding images acquired from the fluorescent microscope with analog mode detection. We provide the reviewer with a link to the GitHub repository with the version of the simulation used for the paper to demonstrate the complexity of the simulation.

<https://github.com/TranslationalImagingCenter/fluoroSim-HyU>

Here, we show some images that will be part of the separate publication with an example of a simulated uniform signal (a Chroma Slide) recreated using our simulation framework after acquiring the experimental signal using a confocal fluorescent microscope. The distribution of intensities between the experimental and simulated signals (A, B) present similar characteristics, even accounting for the stochasticity of photon emission and multiple noise contributions. The average spectrum of each dataset (C, D) shows close resemblance in both shape and intensity.

Comparison of microscope measured uniform signal and corresponding simulation: In this example, a yellow Chroma Slide is imaged using a 780 Zeiss Inverted in spectral mode (32 channels). **(A,C)** The top row shows images, histogram intensities, and the average spectrum for experimental data, focusing on two significant channels (468 nm and 548 nm). **(B, D)** The bottom row shows the corresponding results created by the simulation.

We further characterized the performance of the hyperspectral fluorescent simulation with respect to noise, detectors, and fluorescence characteristics. In the following image, we show the plots of mean vs variance of intensity for a constant fluorescent signal (in this case a Chroma Slide) deriving from experimental (blue lines) and simulated (orange lines) data. The mean/variance plot is expected to be linear when the detector gain is within the linear response range. The plots below show only 6 different channels, but for all 32 channels the data was acquired at a gain of 740, well within the linear gain response range. Each data point in all plots is the (mean, variance) of a 1024x1024 pixels image for that specific channel. We acquired 7 datasets, each with an increasing laser power excitation (constant gain). As expected, the experimental data shows linearity in the mean/variance (blue lines). Similarly,

simulated data where we increased the number of stochastically emitted fluorescence photons to mimic the increase in laser power, also provides a linear mean/variance (orange lines). The plots show consistent similarity between simulations and experimentally acquired data at both high and low intensities, suggesting a good replication of the multiple system noises.

High similarity of mean vs variance plots between experimental and simulated images.

Mean vs Variance plots are provided for experimental and simulated data in analog mode at low photon emissions for a constant fluorescent signal. Linearity in mean/variance is expected within the gain's linear response range. In this image, experimental data was acquired sequentially at increasing laser power, while simulated data was created by increasing the average statistically emitted fluorescent photons, effectively simulating an "increasing laser power". Results show a linear mean/variance relation for experimental data and a similar linear relation for simulated data. Slope values are reported in each plot's legend.

Finally, we show the simulation's ability to reconstruct a biologically relevant hyperspectral image, replicating the spatial characteristics and most typical noises of fluorescent signals acquired through a confocal microscope lens and detectors. A comparison between the individual channel images of the (A) experimental hyperspectral dataset and (B) its synthetic counterpart demonstrates the realism of the computationally generated hyperspectral image. In (C), we compare the intensity profiles of a section of each pair of images (yellow line in A and B), showing, within the stochasticity of noise, how similar the synthetic data pattern (C, orange line) is to the experimental (C, blue line).

Simulations recapitulate spatial intensity distributions across all channels

(A) 2D intensity images of the experimental data for channels 18 (566 nm), 25 (629 nm), and 30 (673 nm) each with 8.9nm bandwidth, provide a baseline comparison for the simulations. **(B)** Corresponding images of the simulated data for channels 18 (566 nm), 25 (629 nm), and 30 (673 nm) show visually similar images for each channel, considering the stochasticity of noise. **(C)** Line profiles for experimental (blue) and simulated (orange) data corresponding to the yellow lines in **A** and **B** provide a comparison for the spatial distribution of intensity values and present a high degree of similarity for the distributions across multiple channels even accounting for stochastic noise.

3) The authors addressed that the HyU method is more computationally efficient than the linear spectral unmixing method. This is true as the spectral dimension was reduced in Hyperspectral Phasors and histogram binning. However, these spectral compression and denoising also require computational power. Does the proposed method is more efficient when the entire process is considered?

ANSWER: This comment was addressed in Supplementary Note 3 and Supplementary Figure

13. We noted that due to the highly efficient and optimized algorithms for matrix inversion, LU is strictly faster than hybrid unmixing overall, but our comment of the greatly decreased computational costs of hybrid unmixing are demonstrated with the usage of fitting algorithms instead of LU. We have edited the main text for clarity, which now reads:

Speed tests with iterative fitting unmixing algorithms demonstrate a speed increase of up to 500-fold when the HyU compressive strategy is applied. (Fig. S13, Supplementary Note 3). Due to the initial computational overhead for encoding spectra in phasors, there is a 2-fold speed reduction for HyU in comparison to standard LU. However, this may be improved with further optimizations of the HyU implementation **or by implementing different types of encoding.**

4) For spectral unmixing, it seems to use the reference signals obtained from pure fluorophores. What happens if there are unknown fluorescence signals? Can the proposed method be applied for blind spectral separation?

ANSWER: In principle, yes, HyU could be implemented with blind spectral separation. However, in this current iteration of the approach, applying HyU to unknown fluorescence signals requires manual inspection of the phasor plot and is not automatic. The biggest advantage of HyU is the capability to visualize the spectra of endmembers on phasor.

- The example reported in Figure 5 B shows how we identified the spectrum from an unidentified component by observing a separate cluster on the phasor. After a literature search, we determined that its spectral shape and anatomical location suggest that the signature is from blood cells.
- Supplementary Figure 11 shows an example of utilizing residual maps as a strategy for identifying unexpected spectral contributions using the phasor by color-coding phasor bins with low unmixing performance, which provide a quantitative clue to which phasor areas may contain an unexpected independent spectral signature.
- Phasor-based blind spectral separation such as those described in the work of Scipioni et al (<https://www.nature.com/articles/s41592-021-01108-4>) could be directly applied to HyU.

We have expanded Supplementary Note 1 titled “Identification of spectra and new components with HyU” with further clarifications, insights, and references on how to distinguish independent spectral signatures from various types of noise. Supplementary Note 1 now reads:

Supplementary Note 1: Identification of spectra and new components with HyU

Identification of independent spectral components has been an adversity for unmixing hyperspectral data. First, the collected spectra may be distorted by reduced SNR. Secondly, excitation of intrinsic signals causes uncertainty of biological sample. Favorably, HyU simplifies this process by adapting Phasor approach and achieving semi- or full-automation process for spectra identification and selection. In HyU spectra can be loaded from an existing library, virtually automating the analysis process. Pre-identified cursors are generated from common fluorophores such as mKO2, tdTomato, mRuby, Citrine. In our experience, obtaining fluorescence spectra from experimental samples has some advantages compared to utilizing spectra from an existing library, as they account for a multitude of experimental and instrumental settings. Imaging settings such as different types of lenses or optical filters (Sup. Figure 4, C and D) together with factors within the microenvironment of samples, such as pH or temperature have the potential to alter the fluorescence spectral emissions¹². In the presence of unexpected fluorescent signals, spectra can also be selected and visualized directly from the phasor. Phasors facilitate the identification of unexpected independent components and their distinction from the multiple system noises. A noise-free spectrum will appear as a single point on the phasor plot, while a spectrum affected by instrument and electronic noises will mainly appear as a gaussian distribution, centered on the original spectral signal⁶. Conversely, a randomized noise across the multiple spectral channels will not produce a clustered aggregate of spectra on the phasor. A constant spectral noise, with a distinct spectrum (e.g. a constant light leakage into the system), would produce a distinct phasor cluster and could be selected for unmixing. The phasor plot representation is a 2D-histogram and provides insights into the frequency of occurrence for these signals. These unexpected independent components in samples often appear as “tails” on the phasor distributions (Sup. Figure 11, C). In our HyU graphical interface, clicking on the phasor visualizes the spectra within a small area (9x9 bins by default, with size adjustable from the interface) of the phasor histogram (Figure 1 D). In the example in Sup. Figure 9-11 we identify 5 distinct endmembers on the Phasor (Sup. Figure 10, C), visualize their spectra identifying Citrine, mRuby, Td-Tomato, mKO2, and one strong autofluorescence signature. The use of Residual Phasor Map (Sup. Figure 11, B) allows for identification of areas in the phasor with high amount of residuals, likely corresponding to a missing endmember in the unmixing. Residual Image Maps (Sup. Figure 11, C) provide a rapid overview of residuals in the image data, for identification of location in the dataset of the missing endmember.

5) Following the previous question, I wonder about the effect of light scattering on the accuracy of the proposed method. In fig4, the proposed method can be applied for volumetric imaging. I wonder there are consistent fluorescence signals over the depth of tissue. Fluorescence signals occurred in deep tissue regions experience more light scattering, which might occur in spectral distortions.

ANSWER: We have performed the following additional changes to address this point:

1. We have added unmixing results from deep imaging performed on zebrafish embryos. (Supplementary Figure 23)
 2. We have also added experiments performed on both freshly excised and fixed mouse tissue for further comparison to more highly scattering tissue. (Supplementary Figures 29, 30)
 3. We expanded the Supplementary Information with details on mouse lines, fluorescent silica beads characterization, mouse tissue samples preparation, and imaging.
-
1. To address the question of light scattering in deep tissue, we added an example of deep imaging in zebrafish, to demonstrate that scattering distortions due to increased depth do not affect the signal and therefore the unmixing, even with signal attenuation (Supplementary Figure 23). As seen in Supplementary Figure 23, the phasor plots of the signals demonstrate a consistent distribution, centered at the same coordinates, even for images acquired deeper into the tissue where the detected spectra decrease in intensity and increase in noise. Following the trend of the absolute intensity average spectra (I) within the center bin shows the attenuation of the signal over depth as the area under the curve decreases. Even with that attenuation, (J) the normalized forms of the average spectra display no difference in shape. The consistency of the average spectra is a strength of the phasor method, which allows for grouping of the similar spectra even with the signal attenuation and noise increase seen in the (K,L) random individual spectra of the pixels across different depths.

Supplementary Figure 23. Phasor analysis on signal distortion in deep tissue

Images at different Z-positions of a 3D (x,y,z) dataset of 19 hpf *Tg(ubiq:lyn-tdTomato)* zebrafish acquired from 0 μm to 80 μm (relative to the dataset) depth displaying at every 13 z-slices. (A-D) Phasors calculated from the single slices at 0 μm, 26 μm, 52 μm, and 78 μm depth. (E-H) Corresponding average intensity images (across the 32 spectral channels) for each z-slice show an expected decrease in fluorescence intensity with depth. (I) Average spectra of the pixels linked to the phasor position bin for the tdTomato fluorescent signature for each of the four presented z-slices (0, 26, 52, 78 μm) show a decrease in the spectral area without change in spectral shape as shown by (J) normalizing the average spectra shown in I to each spectrum's maximum value. This demonstrates the spectral shape does not change across different depths (z-planes), while the overall intensity decreases. (K) Randomly selected spectra from the raw spectral image at 0 μm (blue) and 26 μm (red), five spectra for each image and similarly (L) for 52 μm (green) and 78 μm (yellow). Two yellow and three green spectra are not clearly visible because of low signal intensity.

2. Mouse experiments: we have included two examples of application for the Hybrid Unmixing method: fresh tissue autofluorescence metabolic imaging and fixed tissue fluorescence imaging. These experiments provide support for the reliability of HyU even in tissue that are even more scattering than that of the zebrafish.

The first experiment involved unmixing autofluorescent signals in freshly excised mouse kidney tissue. The unmixed signals include NADH bound, NADH free, Retinol, Retinoic acid, and Elastin, in correspondence to the intrinsic signals previously unmixed in the zebrafish experiments. In Supplementary Figure 29, we demonstrate that the same independent autofluorescent signatures used in the zebrafish to perform HyU can be utilized in the fresh mouse tissue. Even as the intensities of the autofluorescent signatures drop rapidly within the first 75 μm of tissue, we demonstrate that HyU performs as previously shown in zebrafish. The unmixed autofluorescent signals within the mouse tissue yield spatial patterns (Sup. Fig 29 A, B, C, E) corresponding to those described in literature (<https://doi.org/10.1117/1.JBO.19.2.020901>, <https://doi.org/10.1681/ASN.2016101153>, <https://doi.org/10.1021/cr900343z>) and to those previously shown in zebrafish. Cross-section visualizations (Sup. Fig 29 B and E) show consistency in the unmixed autofluorescence patterns along the depth of the sample (z-direction) until signal drops to noise level. For these notoriously weak autofluorescent signals, intensities fade out at approximately 75 μm depth.

In the presence of wavefront distortions in the fluorescent spectra, the expected and most likely behavior of the HyU would be a deterioration of the unmixing quality. This deterioration generally results in a non-organized, more uniform re-assignment of intensities across the unmixed channels. When this scenario happens, severe cross-talk is visible across channels. If one were to visualize these results in a 2-D image, it would result in a “white” color where multiple signals are shown together in the same image, due to how multiple colors are combined in the rendering algorithms used for visualizing fluorescence microscopy data. Supplementary Figure 29 panels B and E are single slices of the volume, representing a 1-pixel wide cross-section of the volume, along the XZ and YZ planes respectively. In these cross sections, there is no visible confounding of signals or “white” color, delineating the overlap of multiple fluorescent signals. This suggests that the algorithm, under reasonable imaging conditions, performs as characterized (Sup. Figures 16,17,18,19). This is mainly because fluorescent signals fade to dark before any appreciable distortion affects the system.

Supplementary Figure 29:

Supplementary Figure 29. Intrinsic fluorescent signatures in fresh mouse tissue

Intrinsic fluorescent signatures in fresh kidney tissue of a 7 months Balb-c mouse imaged with 2-photon excitation at 740nm in a 150 µm deep volume. Despite the increasing scattering effect of this mammal tissue with increasing depth, HyU can perform unmixing of intrinsic fluorescent signals. (A) Volumetric rendering of the unmixing results of five intrinsic fluorescent signatures shows results consistent with literature, as visible in the (B-E) orthogonal views of (C) an unmixed (x,y) cross-section of the volume at 30 µm depth in the sample and its corresponding (B) (x,z) and (E) (y,z) projections. (D) Averaged autofluorescent signals for each acquired spectral (x,y) section over the 150 µm depth of the volume show a sharp decrease of intensities after 75 µm depth as visible in E, the corresponding (y,z) projection.

The second experiment involved unmixing a known fluorescent signal among the background signals within the highly scattering mouse tissue by imaging Cy3 beads injected (described in the new Mouse Sample Preparation section below) into fixed mouse tissue. The ground truth spectra of the Cy3 beads is known, measured in pure bead solution (described in the new Fluorescent silica bead characterization section). With this known constant Cy3 fluorescent spectrum, we can demonstrate:

- a. The efficacy of HyU even when fluorescent signatures are buried deep within highly scattering tissue.
- b. The effect of tissue scattering on the known Cy3 fluorescent spectrum.

Supplementary Figure 30 below shows that the Cy3 beads (magenta) are unmixed with the proper spatial profiles (spherical shape) from the autofluorescent (yellow), background (green), and structural signatures (light blue) of the fixed mouse tissue.

Similarly to Supplementary Figure 29 B,C,E, there is no evident confounding overlap of signals when imaging deeper in the sample. In this case, with stronger autofluorescent (fixative-related) and fluorescent (beads) signals, signals started fading at 110 μm depth in the sample.

In support of our observation that fluorescent spectra, as measured in our commercial instrument, for scattering mammalian samples, are not excessively distorted in their shape, we plot the spectra corresponding to the Cy3 beads at multiple depths (Supplementary Figure 30 F, G), up to 130 μm deep in the sample, in absolute (Sup. Fig 30 F) and relative (Sup. Fig 30 G) intensity. Absolute peak intensity for some of the deeper spectra is around ~ 1300 Digital Levels, corresponding to 2% of the full 16 bit dynamic range of the detector, close to the noise level of the instrument. This translates to visibly noisy spectra for deeper z-planes of the sample, as shown in the relative intensity plots (panel G), but does not show evident spectral distortions.

In regards to the performance of HyU, the unmixed fluorescent signatures remain consistent with expected spatial patterns within the scattering mouse tissue even as the overall intensity drops. Within the imaging depth we could reach in our instrument, the highly scattering nature of the mouse tissue does not affect the performance of HyU, similar to what we previously demonstrated in zebrafish.

Supplementary Figure 30:

Supplementary Figure 30. Extrinsic fluorescent signatures in fixed mouse tissue

We evaluate the performance of HyU in imaging fluorescent signals in a highly scattering fixed kidney tissue of a 7-month-old Balb-c mouse with embedded Cy3 fluorescent beads (Methods) imaged with 2-photon 850nm excitation up to 150 μm deep. (A) Volumetric rendering of the unmixing results of the signals from fixative autofluorescence (autoFL), Cy3 beads, background, and Second Harmonic Generation (SHG). (B-E) Orthogonal views of the same volume for (C) single (x,y) plane of the volume at a depth of 90 μm with cross-sections (yellow hairlines in C) (B) (z,x) of 18 μm and (E) (y,z) of 4 μm respectively showing sections of the unmixed volume containing Cy3 beads at different depths up to 140 μm . (D) Average intensity value for each acquired (x,y) spectral image slice as a function of depth reveals considerable loss of fluorescent signal deeper than 110 μm . (F) Average spectra for each z-plane containing pixels with Cy3 beads signal plotted with absolute intensity (Digital Levels, DL) show decreasing intensity with depth as visible by the area under spectrum. (G) The same averaged spectra are normalized and plotted with Relative Intensity to show the consistency in the spectral shape as a function of depth in reference to Cy3 beads in solution (dashed line).

3. We expanded the supplementary information to include details on mouse lines, fluorescent silica beads characterization, mouse tissue sample preparation, and imaging:

Mouse lines

Mice imaging was approved by the Institutional Animal Care and Use Committee (IACUC) of University of Southern California, Protocol #21311. Experimental research on vertebrates complied with institutional, national and international ethical guidelines. Animals were kept on a 13:11 hours light:dark cycle. Animals were breathing double filtered air, temperature in the room was kept at 68–73 F, and cage bedding was changed weekly. All these factors contributed to minimize intra- and inter-experiment variability. Adult Balb-c mice were euthanized via overdose of isoflurane followed by cardiac puncture. Kidneys were quickly harvested from the mouse, washed in PBS, and cut longitudinally alongside the midsection in order to expose the inner part of the organ. The two halves of the organ were arranged onto a microscope slide for imaging.

Fluorescent silica beads characterization

One fluorescent silica beads solution (Nanocs, Inc.) labeled with Cy3 (Si500-S3-1, 0.5mL, 0.5um, 1% solid, lot# 1608BRX5) was characterized in its spectral fluorescence emission and physical size.

10x dilution in PBS of the beads was placed on a no. 1.5 imaging coverglass and spectrally characterized using spectral mode on a Zeiss LSM 780 laser confocal scanning microscope equipped with a 32-channel detector using 40x/1.1 W LD C-Apochromat Korr UV-VIS-IR lens utilizing a 2-photon laser at 740 nm to excite fluorescence from the beads, using a 690nm lowpass filter to separate excitation and fluorescence. Spectra obtained from multiple beads with the same label were averaged, producing the reference spectrum reported in Sup. Figure 30 G (dashed line). Fluorescent silica bead size and concentration were determined via nanoparticle tracking analysis (NTA) on the Nanosight NS300 (Malvern Panalytical). Samples were run 5 times and results averaged for final size and concentration values reported.

Mouse Sample preparation

For autofluorescent measurements, mouse organ samples were collected from Balb-c mice. Following euthanasia, organs were resected and washed in Phosphate Buffered Saline (PBS) to remove residual blood and kept in PBS until imaging preparation. Organs were sectioned in order to image the internal architecture and mounted on a glass imaging dish with sufficient PBS to avoid dehydration of the sample. Following imaging, all samples were fixed in a 10% Neutral Buffered Formalin solution at 4-°C.

For ex vivo bead characterization in tissue, mouse organ samples were collected from Balb-c mice. Following euthanasia, organs were resected and washed in PBS followed by incubation for

at least 24 hours in 10% buffered formalin. The kidney was then removed from the fixative and sectioned into smaller ~5x5x5mm pieces for imaging. A fluorescent silica beads stock (Nanocs, Inc.) labeled with Cy3 (Si500-S3-1, 0.5mL, 0.5um, 1% solid, lot# 1608BRX5) and previously characterized was prepared using a 10x dilution of the fluorescent beads from their stock concentration. Beads were injected in the sample using 50ul of these solutions loaded into a 0.5mL syringe with a 28g needle. The kidney sections were then placed in imaging dishes with a small volume of PBS to keep the samples hydrated prior to imaging.

Image Acquisition.

Images were acquired on a Zeiss LSM 780 laser confocal scanning microscope equipped with a 32-channel detector using 40x/1.1 W LD C-Apochromat Korr UV-VIS-IR lens at 28-^oC.

Samples of *Gt(cltca-Citrine)*, *Tg(ubiq:lyn-tdTomato)*, *Tg(fli1::mKO2)*, and *Tg(ubiq:Lifeact-mRuby)*, were simultaneously imaged with 488 nm and 561 nm laser excitation, for citrine, tdTomato, mKO2, and mRuby. A narrow 488 nm/561 nm dichroic mirror was used to separate excitation and fluorescence emission. Samples were imaged with a 2-photon laser at 740 nm to excite autofluorescence, using a 690nm lowpass filter to separate excitation and fluorescence.

Samples of mouse kidney tissue were imaged with 2-photon excitation at 740 nm or 850 nm with a 690+ nm lowpass filter, at 37-^oC incubation.

For all samples, detection was performed at the full available range (410.5-694.9nm) with 8.9nm spectral binning.

Reviewer #3 (Remarks to the Author: Strength of the claims):

This work demonstrates the superiority of the proposed method using computer simulation and experimental data. The authors clearly claim that the proposed method is more efficient and robust than conventional linear spectral unmixing methods.

ANSWER: We thank this reviewer for the constructive comments and helping us improve the manuscript.

Reviewer #3 (Remarks to the Author: Reproducibility):

The authors provide the code and data used in the manuscript. This allows other people to reproduce these results. And the dataset used in this work is appropriate for the purpose of the study.

ANSWER: We thank this reviewer for the insightful remarks.

Reviewers comments:

Reviewer #1:

Remarks to the Author: Overall significance:

The results are significant and as a new method of analysis of hyperspectral data - it adds to the community. The authors have replied to most of my questions. I do think they need to add a little more to explain how this method is different from the phasor and linear combination method. I am a bit conflicted as to usual readers – the difference between phasor as analysis method and phasor as encoder as to decrease the data from 10^7 to 10^4 may not be clear. The authors comment about collage autofluorescence – while SHG is certainly true for collagen 1 it is not for other types of collagens – some of them can be highly fluorescent. About question 6 – While I agree that mathematically a point on the line between the position of two other phasor points may be unlikely. That is true for the center of the distribution. It is definitely not true when S/N creates a distribution. In that case the points can be along that line. If the authors propose only using the center of the distribution – that needs to be mentioned clearly. About FAD – I can see why laser power can be a valid argument. I do not agree with the effect of zooming. Lets say an area has a particular NADH/FAD distribution. Why would it matter if we zoom into a smaller part of that or a larger area? Unless zooming in separates the areas of high NADH and high FAD spatially and they appear at different phasor positions. That brings out a separate problem. If HyU is zoom dependent – then how will anyone know if they are getting the proper distribution or not? Finally I do like the new figures.

Remarks to the Author: Impact:

I would suggest Nature communications or Scientific reports. The authors use phasor- but not to it's full capacity. The paper will indeed influence the field. However, I do feel publication in Nature methods can influence it in a way that may not be the best for the field.

Remarks to the Author: Strength of the claims:

The work is convincing.

Remarks to the Author: Reproducibility:

The paper is very detailed and can be reproducible.

Reviewer #2:

Remarks to the Author: Overall significance:

In their revised manuscript "HyU: Hybrid Unmixing for longitudinal in vivo imaging of low signal to noise fluorescence", the authors convincingly highlight the achievements/unique features of the further-developed

algorithm for multiplexed imaging of hyperspectral data, expanding its use to low signal-to-noise ratio of real data, moving towards two-photon microscopy. Especially, they made particular efforts in demonstrating the power and the limitations of the algorithm when applied in highly-scattering tissue or when the number of detection channels is limited, providing an excellent and useful benchmarking for a broad life sciences audience interested in imaging and microscopy.

Although, I think, of high interest and definitely an important next step, I do agree with the authors that intravital imaging of highly scattering organs is a major challenge and can be the subject of future work. I fully recommend the present manuscript for publication in its present form.

Remarks to the Author: Impact:

As mentioned above, with the new added data, the presented algorithm will find broad applicability for all sorts of data, which typically need to deal with low signals.

Remarks to the Author: Strength of the claims:

In the current form, the revised manuscript provides all necessary evidence for the unique power of the HyU algorithm.

Remarks to the Author: Reproducibility:

Beyond of statistics, I find the broad range of applications, excitation schemes and microscopy types particularly important to emphasize the general validity of the algorithm.

Reviewer #3:

Remarks to the Author: Overall significance:

In this report, Chiang and co-workers presented the Hybrid Unmixing (HyU) method for the efficient and robust analysis of multiple fluorescent signals. The authors employ the spectral phasor method for reducing spectral data dimension and denoising noises in the imaging system. The superiority of the proposed method has been demonstrated compared to the conventional linear unmixing method by exploiting computer simulation and experimental results. This article seems to be timely the report as increasing the biomedical applications using hyperspectral imaging methods. And the authors appropriately discussed the proposed method by comparing previous techniques.

Remarks to the Author: Strength of the claims:

The revised manuscript has clearly addressed all of my previous comments and concerns.

Reviewers comments:

We would like to thank all Reviewers for their constructive criticism that helped considerably to improve the quality of this manuscript.

Reviewer #1 (Remarks to the Author: Overall significance):

The results are significant and as a new method of analysis of hyperspectral data - it adds to the community.

We thank the reviewer for this comment.

The authors have replied to most of my questions. I do think they need to add a little more to explain how this method is different from the phasor and linear combination method. I am a bit conflicted as to

usual readers – the difference between phasor as analysis method and phasor as encoder as to decrease the data from 10^7 to 10^4 may not be clear.

We have refined our previous explanation (yellow highlight) of the differences between the Standard Phasor analysis method and the Hybrid Unmixing method in the introduction. We have now expanded this (green highlighted portions) to address the specific concern about the difference between the phasor method as an analysis and as an encoder:

We have developed Hybrid Unmixing (HyU) as an answer to the challenges that have limited the wider acceptance of HFI for *in vivo* imaging. HyU combines our previous phasor hyperspectral approach²⁶ with traditional unmixing algorithms to untangle the fluorescent signals more rapidly and more accurately from multiple exogenous and endogenous labels. The phasor approach²⁶ is a popular dimensionality reduction approach for the analysis of both fluorescence lifetime and spectral image analysis^{27–29}. In HyU, the phasor approach provides spectral compression, denoising, and computational reduction that simplifies both pre-processing³⁰ and unmixing^{31–33} of HFI datasets. Standard Phasor analysis^{26,27,34–41} is fully supervised and requires a manual selection of regions or points on the phasor plot, a graphical representation of the transformed spectra. In contrast, HyU utilizes phasor processing as an encoder to aggregate similar spectra onto the phasor plot, reducing even the largest volumetric datasets so that unmixing algorithms, such as LU, can be applied on a far smaller number of elements (the number of pixels on the phasor plot). Furthermore HyU provides unsupervised analysis of the HFI data, simplifying the data processing and removing user subjectivity. Our results show that HyU offers three key advantages: (1) improved unmixing over conventional LU, especially for low intensity images, down to 5 photons per spectra; (2) simplified identification of independent spectral components; (3) dramatically faster processing of large datasets, overcoming the typical unmixing bottleneck for *in vivo* fluorescence microscopy.

We believe this further extended description, obeying the limited word count of the manuscript sufficiently clarifies the difference between Standard Phasor analysis and HyU. The graphical representation in Figure 1 clarifies this: notice the absence of any typical selection on the phasor plot, either in the form of Region of Interest processing or in the form of linear geometric unmixing, both factors that are essential for Standard Phasor analysis.

This combination of description and Fig 1 (and absence of Standard Phasor indicators) should be sufficient to avoid any confusion in the reader.

The authors comment about collagen autofluorescence – while SHG is certainly true for collagen 1 it is not for other types of collagens – some of them can be highly fluorescent.

We thank the reviewer for this insight. While we agree that some types of collagen can be highly fluorescent, the references we cited in our previous answer (duplicated below) demonstrate that depending on the imaging conditions, collagen may not be visible. We did not observe any such signal within the regions of the specific samples imaged in this work, under our imaging conditions (reported in Table S1 supplementary).

Multiple references in literature (PMC4337962 / PMID: 22402635, PMC123202 / PMID: 12177437, PMC4337962 / PMID: 22402635) report 2-photon fluorescence of collagen to be very low at 740 nm...

About question 6 – While I agree that mathematically a point on the line between the position of two other phasor points may be unlikely. That is true for the center of the distribution. It is definitely not true when S/N creates a distribution. In that case the points can be along that line. If the authors propose only using the center of the distribution – that needs to be mentioned clearly.

Owing to the extended length of the Response to Referees, we tried summarizing the common points to the questions of each reviewer, where possible, in the first comment. In this case, the noise aspect in question was expansively described in the initial comment “Information loss and noise”.

We do not propose only using the center of the distribution and we account for noise as explained when presenting the ~3.3 billion simulations used to assemble Supplemental Figures 16, 17, 18, 19.

Our answer to question 6 covers two main subjects:

- i) acknowledging that there are specific cases where mathematically, linear unmixing would not work
- ii) acknowledging the likelihood of many fluorophores and their combinations to occupy the same location (whether or not we are referring to the center or the full distribution corresponding to a fluorescent signature). In this scenario, where multiple components may be present in the exact same phasor location, we comment that the amount of error that may arise from this situation (encoding loss) is most likely overwhelmed by the error introduced by the noise instead.

We report here the second part of question 6:

However, we have demonstrated in our reply to the first comment and in the numerous additional simulations (Supplemental Figure 16, 17, 18, 19) that noise affects the unmixing results much more than the encoding loss of the phasor method.

The first comment here referenced states:

Information loss and noise. We are aware that the phasor transform is a lossy encoder that in principle carries a reduced percentage of the information compared to the original “pure” data. This is evident in the scenario of very high-quality signals, but in the case of fluorescent signals, where signal to noise often decreases to lower digits, the encoding loss is less relevant compared to the noise of the fluorescent signals.

This fundamental advantage of increasing SNR in noisy data makes phasor a valuable tool for fluorescence microscopy (both FLIM and hyperspectral); this point is reported by multiple groups using phasors (Gratton:

<https://www.pnas.org/doi/full/10.1073/pnas.1108161108>,

<https://escholarship.org/content/qt5g279175/qt5g279175.pdf>, Vicidomini:

<https://www.nature.com/articles/ncomms7701> , Gerritsen:

<https://pubmed.ncbi.nlm.nih.gov/22714302/>, Fraser:

<https://pubmed.ncbi.nlm.nih.gov/28068315/>), and more recently nicely described in the work of Scipioni et al (<https://www.nature.com/articles/s41592-021-01108-4>) *“However, microscopy data are affected by a number of other detrimental factors, [...] which results in decreased signal-to-background ratio (SBR). [...] the phasor approach shows increased precision (Fig. 1f,i), decreased bias (Fig. 1e,h) and a three orders of magnitude lower execution time (Fig. 1g,j) with respect to the least mean square (LMS) fitting procedure”* .

To support the validity of this hybrid unmixing approach, we have assembled a complex simulation matrix representing the performance of HyU in unmixing 2 to 8 labels as a function of the spatial overlap in the sample. This simulation matrix is built on top of the complex simulation we designed (further described below in our answer to this reviewer’s question 14), which is soon to be published in a separate manuscript. This simulation accounts for a multitude of real-world noises in experimental samples that are regularly imaged (stochasticity of fluorescence spectral emission, poisson, readout noise, electronics transfer noise, detector sensitivity at different wavelength). The results of applying our approach on an array of simulations under different conditions of SNR, number of filters applied, in comparison to standard Linear Unmixing are now reported in Supplementary Figures 16, 17, 18, and 19. We further describe how multiple components are affected by our hybrid unmixing approach in our answer to question 17 for this reviewer.

About FAD – I can see why laser power can be a valid argument. I do not agree with the effect of zooming. Lets say an area has a particular NADH/FAD distribution. Why would it matter if we zoom into a smaller part of that or a larger area? Unless zooming in separates the areas of high NADH and high FAD spatially and they appear at different phasor positions. That brings out a separate problem. If HyU is zoom dependent – then how will anyone know if they are getting the proper distribution or not?

We worry that the reviewer is misunderstanding our intended meaning of “zooming in to image a smaller region”. It is not merely cropping the large image to display a smaller field of view (FoV), in which case the reviewer would be correct. Instead, in the optics of confocal laser scanning microscope, “zooming” by using a higher magnification objective or by scanning a smaller region of the specimen delivers more exciting light to the FoV, increasing it by the square of the zoom or magnification of the objective lens. Thus, zooming in by two- fold would increase the fluorescence excitation by four-fold for one-photon excitation, and by 16-fold for two-photon excitation.

We hope that this clarifies the issue, but if the reviewer would like to explore this further, we direct them to the excellent book, Principles of Fluorescence Spectroscopy (Lakowicz), section 18.2

“The physical origin of the 2PE cross-sections can be understood by some simple considerations. For one-photon absorption the number of photons absorbed per second (NA_1) is given by

$$NA_1 \text{ (photon/s)} = \sigma_1 \text{ (cm}^2\text{)} I \text{ (photon/cm}^2\text{s)} \quad (18.1)$$

where I is the intensity and σ_1 is the cross-section for one-photon absorption. The units are given within the parentheses. The cross-section in cm^2 is multiplied by the number of photons passing near the molecule per second to yield the number of photons absorbed per second. To obtain NA_1 in photons per second the cross-section must be in units of cm^2 .

Now consider two-photon absorption. The number of photons absorbed per second by 2PE (NA_2) is given by

$$NA_2 \text{ (photons/s)} = \sigma_2 I^2 \text{ (photons/cm}^2\text{s)}^2 \quad (18.2)$$

In order for the units to match on both sides of eq. 18.2 the units of σ_2 must be $\text{cm}^4\text{s/photon}$. Similarly, for 3PE,

$$NA_3 \text{ (photons/s)} = \sigma_3 I^3 \text{ (photons/cm}^2\text{s)}^3 \quad (18.3)$$

and the units of a three-photon cross-section are $\text{cm}^6 \text{ s}^2/\text{photon}^2$.”

We quote the Principles of Fluorescence Spectroscopy (Lakowicz), section 1.7: “The steady-state intensity (of the fluorophore) is given by

$$I_{SS} = \int_0^{\infty} I_0 e^{-t/\tau} dt = I_0 \tau \quad (1.16)$$

The value of I_0 can be considered to be a parameter that depends on the fluorophore concentration and a number of instrumental parameters.”

While it is true that the **true spectral signature** should not change depending on the zoom or power, the **amount of signal** depends not only on the concentration of the fluorophores, and the excitation wavelength used, but also on the zoom and excitation power.

These factors are major contributors to the measured intensity of the fluorophore, which can alter both the SNR and could alter the resulting analysis. The chief advantage of our method is to overcome the low signal that occurs during these types of acquisitions. If signal cannot be detected clearly (signal approaches the noise), it is extremely challenging if not impossible to perform any kind of analysis.

The reviewer goes on to ask:

That brings out a separate problem. If HyU is zoom dependent – then how will anyone know if they are getting the proper distribution or not?

HyU, like any fluorescent microscopy imaging experiment, requires proper experimental settings based on the scientific/experimental scope and probes utilized. The absence of a signal cannot be taken as the absence of an analyte in HyU or any other analysis approach.

To obtain more meaningful results, in fluorescence microscopy, imaging area (zoom), pixel dwell time and laser power will need to be properly set. There are multiple strategies for validating if experimental settings are proper. In the manuscript, we show the pipeline for using phasors to identify presence of fluorophores with practical examples to assist the readership in proper experimental design (see Supplementary Figure 11 and more extensively in Supplementary Note 1: Identification of spectra and new components with HyU).

Finally I do like the new figures.

We thank the reviewer for this comment.

Reviewer #1 (Remarks to the Author: Impact):

I would suggest Nature communications or Scientific reports. The authors use phasor- but not to its full capacity. The paper will indeed influence the field. However, I do feel publication in Nature methods can influence it in a way that may not be the best for the field.

We thank the reviewer for this comment.

Reviewer #1 (Remarks to the Author: Strength of the claims): The

work is convincing.

We thank the reviewer for this comment.

Reviewer #1 (Remarks to the Author: Reproducibility): The

paper is very detailed and can be reproducible.

We thank the reviewer for this comment.

Reviewer #2 (Remarks to the Author: Overall significance):

In their revised manuscript "HyU: Hybrid Unmixing for longitudinal in vivo imaging of low signal to noise fluorescence", the authors convincingly highlight the achievements/unique features of the further-developed algorithm for multiplexed imaging of hyperspectral data, expanding its use to low signal-to-noise ratio of real data, moving towards two-photon microscopy. Especially, they made particular efforts in demonstrating the power and the limitations of the algorithm when applied in highly-scattering tissue or when the number of detection channels is limited, providing an excellent and useful benchmarking for a broad life sciences audience interested in imaging and microscopy.

Although, I think, of high interest and definitely an important next step, I do agree with the authors that intravital imaging of highly scattering organs is a major challenge and can be the subject of future work. I fully recommend the present manuscript for publication in its present form.

We thank this reviewer for this comment. Reviewer

#2 (Remarks to the Author: Impact):

As mentioned above, with the new added data, the presented algorithm will find broad applicability for all sorts of data, which typically need to deal with low signals.

Reviewer #2 (Remarks to the Author: Strength of the claims):

In the current form, the revised manuscript provides all necessary evidence for the unique power of the HyU algorithm.

Reviewer #2 (Remarks to the Author: Reproducibility):

Beyond of statistics, I find the broad range of applications, excitation schemes and microscopy types particularly important to emphasize the general validity of the algorithm.

Reviewer #3 (Remarks to the Author: Overall significance):

In this report, Chiang and co-workers presented the Hybrid Unmixing (HyU) method for the efficient and robust analysis of multiple fluorescent signals. The authors employ the spectral phasor method for reducing spectral data dimension and denoising noises in the imaging system. The superiority of the proposed method has been demonstrated compared to the

conventional linear unmixing method by exploiting computer simulation and experimental results. This article seems to be timely the report as increasing the biomedical applications using hyperspectral imaging methods. And the authors appropriately discussed the proposed method by comparing previous techniques.

We thank the reviewer for this comment.

Reviewer #3 (Remarks to the Author: Strength of the claims):

The revised manuscript has clearly addressed all of my previous comments and concerns.

We thank the reviewer for this comment.